# Optimizing over Multiple Distributions under Generalized Quasar-Convexity Condition

**Shihong Ding**[1]
dingshihong@stu.pku.edu.cn

**Long Yang**[1]
YANGLONG001@pku.edu.cn

**Luo Luo**[2,4]
luoluo@fudan.edu.cn

**Cong Fang**[1,3][†]
fangcong@pku.edu.cn

[1] State Key Lab of General AI, School of Intelligence Science and Technology, Peking University
[2] School of Data Science, Fudan University
[3] Institute for Artificial Intelligence, Peking University
[4] Shanghai Key Laboratory for Contemporary Applied Mathematics

## Abstract

We study a typical optimization model where the optimization variable is composed of multiple probability distributions. Though the model appears frequently in practice, such as for policy problems, it lacks specific analysis in the general setting. For this optimization problem, we propose a new structural condition/landscape description named generalized quasar-convexity (GQC) beyond the realms of convexity. In contrast to original quasar-convexity [24], GQC allows an individual quasar-convex parameter $\gamma_i$ for each variable block $i$ and the smaller of $\gamma_i$ implies less block-convexity. To minimize the objective function, we consider a generalized oracle termed as the internal function that includes the standard gradient oracle as a special case. We provide optimistic mirror descent (OMD) for multiple distributions and prove that the algorithm can achieve an adaptive $\tilde{\mathcal{O}}((\sum_{i=1}^{d} 1/\gamma_i)\varepsilon^{-1})$ iteration complexity to find an $\varepsilon$-suboptimal global solution without pre-known the exact values of $\gamma_i$ when the objective admits "polynomial-like" structural. Notably, it achieves iteration complexity that does not explicitly depend on the number of distributions and strictly faster ($\sum_{i=1}^{d} 1/\gamma_i$ v.s. $d \max_{i \in [1:d]} 1/\gamma_i$) than mirror decent methods. We also extend GQC to the minimax optimization problem proposing the generalized quasar-convexity-concavity (GQCC) condition and a decentralized variant of OMD with regularization. Finally, we show the applications of our algorithmic framework on discounted Markov Decision Processes problem and Markov games, which bring new insights on the landscape analysis of reinforcement learning.

## 1 Introduction

We study a common class of generic minimization problem

$$\min_{\boldsymbol{x} \in \mathcal{X}} f(\boldsymbol{x}), \tag{1}$$

where the optimization variable $\boldsymbol{x}$ is composed of $d$ probability distributions $\{\boldsymbol{x}_i\}_{i=1}^{d}$ and $\mathcal{X}$ denotes the product space of the $d$ probability simplexes. Problem (1) meets widespread applications in

---

[†]Corresponding author.

38th Conference on Neural Information Processing Systems (NeurIPS 2024).

reinforcement learning optimization [62, 2, 35], multi-class classification [53] and model selection type aggregation [29]. In this paper, we are particularly interested in the case where $d$ is reasonably large and we manage to obtain complexities dependent of $d$ non-explicitly.

When $f$ is convex with respect to $\boldsymbol{x}$, many efficient algorithms can be powerful tools for solving Problem (1). One well-known algorithm is mirror descent (MD) [5] which is based on Bregman divergence. The wide choices of Bregman divergence enable the algorithm to iterate and converge under specifically constrained region [34]. In particular, if one applies the usual Euclidean distance, the algorithm reduces to project gradient descent [37]. One common and more sophisticated selection is the Kullback-Leibler (KL) divergence, the algorithm thereby becoming the variant of multiplicative weights update (MWU) [41] over probability distribution.

Turning to the non-convex world, specific analysis for Problem (1) is rare. In general, finding an approximate global solution suffers from the curse of dimensionality [51, 46]. And one interesting direction is to consider suitable relaxations for the desired solutions, such as an approximate local stationary point of smooth functions [31, 19]. However, for many cases, local solutions may not be sufficient. Moreover, the algorithms often converge much faster in practice than the theoretic lower bounds in non-convex optimization suggest. This observed discrepancy can be attributed to the fairly weak assumptions underpinning these generic bounds. For example, many generic non-convex optimization theories, e.g. Carmon et al. [7, 8] only focus on the consideration of Lipschitz continuity of the gradient and some higher-order derivatives. In practice, the objective is often more "structured". For example, the recent progress in neural networks shows that systems of neural networks approximate convex kernel systems when the model is overparameterized [28]. As pointed out by Hinder et al. [24], much more research is needed to characterize structured sets of functions for which minimizers can be efficiently found; It was also noted by Yurii Nesterov [47] that lots of functions are essentially convex; Our work follows this research line.

We propose generalized quasar-convexity (GQC) for the class of "structure". The original quasar-convex functions [22] is parameterized by a constant $\gamma \in (0, 1]$ and requires $f(\boldsymbol{x}) - f(\boldsymbol{x}^*) \leq \frac{1}{\gamma}\langle \nabla f(\boldsymbol{x}), \boldsymbol{x} - \boldsymbol{x}^* \rangle$. These functions are unimodal on all lines that pass through a global minimizer and so all critical points are minimizers. We extend quasar-convexity by introducing individual quasar-convex parameter $\gamma_i$ for each distribution $\boldsymbol{x}_i$. Therefore GQC is parameterized by $d$ constants $\{\gamma_i\}_{i=1}^d$ and implies quasar-convexity in the case $d = 1$. The main intuition of the generalization is the observation that $d/\min_{i\in[1:d]}\gamma_i$ often depends on the number of distributions $d$ in real problems, whereas, $\sum_{i=1}^d 1/\gamma_i$ may not. That is to say, the hardness for distribution $i$ diverges according to the magnitude of $\gamma_i$. The larger of $\gamma_i$ implies more convexity and the simpler to solve $\boldsymbol{x}_i$. In general, one always have $\sum_{i=1}^d 1/\gamma_i \leq d\max_{i\in[1:d]}1/\gamma_i$. In the worst case, $\sum_{i=1}^d 1/\gamma_i$ can be $d$ times smaller than $d\max_{i\in[1:d]}1/\gamma_i$ (see discussions in Section 3.3), which motivates us to study the GQC condition.

We then study designing efficient algorithms to solve (1). One simple case is when $\{\gamma_i\}_{i=1}^m$ is pre-known by the algorithms. The possible direction is to impose a $\gamma_i$-dependent update rule, such as by non-uniform sampling. However, in general cases, $\{\gamma_i\}_{i=1}^m$ is not known and determining $\{\gamma_i\}_{i=1}^m$ require non-negligible costs.

In this paper, we consider a generalized oracle, which we refer to as the internal function. Here the standard gradient oracle can be viewed as a special case of the internal function. We provide the optimistic mirror descent algorithm for multiple distributions, which makes sure that each probability distribution is updated according to its own internal function. We first establish an $\mathcal{O}((d\gamma_{\max})^{1/2}(\sum_{i=1}^d \gamma_i^{-1})^{3/2}L\varepsilon^{-1}\log(N))$ complexity with $N = \max_{i\in[1:d]}n_i$ and $\gamma_{\max} = \max_{i\in[1:d]}\gamma_i$ when $\gamma_{\max} < \infty$. However, such an complexity depends on $d\gamma_{\max}$ and requires the step size rely on pre-known $\gamma_{\max}\sum_{i=1}^d \gamma_i^{-1}$. We then consider $f$ satisfies "polynomial-like" structural (see Assumption 3.3). We show the assumption can be achieved in a variety of function classes and important machine learning problems. Under the assumption, we show the algorithm can adapt to the values of $\{\gamma_i\}_{i=1}^m$ and guarantees an reduced iteration complexity $\mathcal{O}((\sum_{i=1}^d 1/\gamma_i)\varepsilon^{-1}\log(N)\log^{4.5}(\varepsilon^{-1}))$. In the following, the $\widetilde{\mathcal{O}}(\cdot)$ notation hides factors that are polynomial in $\log(\varepsilon^{-1})$ and $\log(N)$.

We also extend our framework to the minimax optimization

$$\min_{\boldsymbol{x}\in\mathcal{X}} \max_{\boldsymbol{y}\in\mathcal{Y}} f(\boldsymbol{x}, \boldsymbol{y}), \tag{2}$$

| Solution type | Related work | Iteration complexity | Single loop |
|---|---|---|---|
| $\varepsilon$-approximate NE | **Cen et al. [9]** **Chen et al. [12]** | $\widetilde{\mathcal{O}}\left(\frac{1}{(1-\theta)^2\varepsilon}\right)$ | ✗ |
| | **Wei et al. [67]** | $\widetilde{\mathcal{O}}\left(\frac{\lvert\mathcal{S}\rvert^3}{(1-\theta)^8\varepsilon^2}\right)$ | ✓ |
| | **Cen et al. [10]** | $\widetilde{\mathcal{O}}\left(\frac{\lvert\mathcal{S}\rvert}{(1-\theta)^4\varepsilon}\right)$ | ✓ |
| | **This Work** | $\widetilde{\mathcal{O}}\left(\frac{1}{(1-\theta)^{2.5}\varepsilon}\right)$ | ✓ |

Table 1: Comparison of policy optimization methods for finding an $\varepsilon$-approximate NE of infinite horizon two-player zero-sum Markov games in terms of the max-min gap (see Eq. (4)). Since the iteration complexity of several research works (such as Zhao et al. [75], Alacaoglu et al. [3] and Zeng et al. [72]) involve concentrability coefficient and initial distribution mismatch coefficient, we will not delve into them here.

where both $x$ and $y$ are composed of $d$ probability distributions, and $\mathcal{Z} = \mathcal{X} \times \mathcal{Y}$ is a joint region. In the general non-convex and non-concave setting, it is known that finding even an approximated local solution for (2) is computationally intractable [16]. We introduce the generalized quasar-convexity-concavity (GQCC) condition analogous to GQC and demonstrate the feasibility of obtaining an $\varepsilon$-approximate Nash equilibrium with $\mathcal{O}((1 - \theta)^{-2.5} \max_{\boldsymbol{z} \in \mathcal{Z}}(\sum_{i=1}^{d} \psi_i(\boldsymbol{z}))\varepsilon^{-1} \log(M) \log(\varepsilon^{-1}))$ iteration complexities, where $\max_{\boldsymbol{z} \in \mathcal{Z}}(\sum_{i=1}^{d} \psi_i(\boldsymbol{z}))$ is analogous to $(\sum_{i=1}^{d} 1/\gamma_i)$ with $\psi_i(\boldsymbol{z})$ defined in the GQCC condition; $\theta$ is the discount parameter; $M = \max_{i \in [1:d]}\{m_i + n_i\}$. Intuitively, the GQCC condition can be viewed as the generalization of convexity-concavity condition. Similarly, the $\widetilde{\mathcal{O}}(\cdot)$ notation hides factors that are polynomial in $\log(\varepsilon^{-1})$ and $\log(M)$.

Finally, we demonstrate the applications of our framework. For problem (1), we consider both infinite horizon discounted and finite horizon MDPs problem. For problem (2), we study the infinite horizon two-player zero-sum Markov games. We prove the learning objectives admit the GQC and GQCC conditions, respectively. This provides new landscape description for RL problems, thereby bringing new insights. Accordingly, our algorithms achieve state-of-the-art iteration complexities up to logarithmic factors. We provide $\widetilde{\mathcal{O}}(\varepsilon^{-1})$ iteration bound for finding an $\varepsilon$-approximate Nash equilibrium of infinite horizon two-player zero-sum Markov games, which outperforms the $\widetilde{\mathcal{O}}(\lvert\mathcal{S}\rvert^3\varepsilon^{-2})$ bound of Wei et al. [67] and the $\widetilde{\mathcal{O}}(\lvert\mathcal{S}\rvert\varepsilon^{-1})$ bound of Cen et al. [10] by factors of $\lvert\mathcal{S}\rvert^3\varepsilon^{-1}$ and $\lvert\mathcal{S}\rvert$, respectively, up to a logarithmic factor.

## 1.1 Contribution

(A) We introduce new structural conditions GQC for minimization problems and GQCC for minimax problems over multiple distributions.

(B) We provide adaptive algorithm that achieves $\widetilde{\mathcal{O}}((\sum_{i=1}^{d} 1/\gamma_i)\varepsilon^{-1})$ iteration complexities to find an $\varepsilon$-suboptimal global minimum of "polynomial-like" function under GQC. We also provide an implementable minimax algorithm, given a generalized quasar-convex-concave function with proper conditions, uses $\widetilde{\mathcal{O}}((1 - \theta)^{-2.5} \max_{\boldsymbol{z} \in \mathcal{Z}}(\sum_{i=1}^{d} \psi_i(\boldsymbol{z}))\,\varepsilon^{-1})$ iterations to find an $\varepsilon$-approximate Nash equilibrium.

(C) We show that discounted MDP and infinite horizon two-player zero-sum Markov games admit the GQC and GQCC conditions, respectively, and also satisfy our mild assumptions. In addition, we provide $\widetilde{\mathcal{O}}((1 - \theta)^{-2.5}\varepsilon^{-1})$ iteration bound for finding an $\varepsilon$-approximate Nash equilibrium of infinite horizon two-player zero-sum Markov games. Detailed comparisons between our method and prior arts are provided in Table 1.

## 1.2 Related Works

**Minimization:** Convexity condition has been studied at length and plays a critical role in optimizing minimization problems [59, 44, 25, 60, 6, 49]. Several other "convexity-like" conditions have

attracted considerable attention, which provide opportunity for designing algorithmic framework to achieve global convergence. Star-convexity [47] is a typical example that relaxes convexity, showing potential in machine learning recently [32, 76]. Quasi-convexity, which admits that the highest point along any line segment is one of the endpoints, is also an important condition [6]. Following this, the concept of weak quasi-convexity is proposed by Hardt et al. [22] which is an extension of star-convexity in the differentiable case, and Hinder et al. [24] provides lower bound for the number of gradient evaluations to find an $\varepsilon$-minimizer of a quasar-convex function (a linguistically clearer redefinition of weak quasi-convex function claimed by Hinder et al. [24] ).

**Minimax Optimization:** Minimax problem attracted considerable attention in machine learning. There exist a variety of algorithms to find the approximate Nash equilibrium points [63, 43, 48, 45, 40, 33, 55, 66, 27] or stationary points [71] for convex-concave functions. Without convex-concave assumption, there exist related work considered specific structures in objective, including nonconvex-(strongly-)concave assumption [39, 73, 50], Kurdyka–Lojasiewicz condition (or specific PL condition) [68, 11, 69, 38], interaction dominant condition [21] and negative comonotonicity [17, 36].

**RL Landscape Descriptions:** For the policy gradient based model of infinite horizon reinforcement learning problems, Agarwal et al. [2] provides a convergence proof for the natural policy gradient descent, which is the same as the mirror descent-modified policy iteration algorithm [20] with negative entropy as the Bregman divergence. Subsequently, Lan [35] focuses on exploring the structural properties of infinite horizon reinforcement learning problems with convex regularizers. For two-player zero-sum Markov games [61, 42] under full information setting, there are various algorithms [26, 54, 64, 18, 42, 67, 9, 74, 70] have been proposed. Specifically, Cen et al. [9] focus on finding approximate minimax soft $Q$-function in regularized infinite horizon setting; Zhao et al. [74] focus on finding one-sided approximate Nash equilibrium in standard infinite horizon setting with $\tilde{\mathcal{O}}(\varepsilon^{-1})$ iteration bound which depends on the concentrability coefficient; Yang and Ma [70] focus on finding approximate Nash equilibrium in standard finite horizon setting with $\tilde{\mathcal{O}}(\varepsilon^{-1})$ iteration bound.

**Related Works on Optimistic Mirror Descent (OMD) and Optimistic Multiplicative Weights Update (OMWU):** The connection between online learning and game theory [58, 4, 23, 1] has since led to the discovery of broad learning algorithms such as multiplicative weights update (MWU) [41]. Rakhlin and Sridharan [57] introduces an optimistic variant of online mirror descent [56, 14]– optimistic mirror descent. Daskalakis et al. [15] shows that the external regret of each player achieves near-optimal growth in multi-player general-sum games, with all players employ the optimistic multiplicative weights update.

## 2  Preliminary

**Notation:** Let $\boldsymbol{x} = (\boldsymbol{x}_1, \cdots, \boldsymbol{x}_d) \in \mathbb{R}^{\sum_{i=1}^d n_i}$ be the joint vector variable, for every vector variable $\boldsymbol{x}_i \in \mathbb{R}^{n_i}$. Let $\boldsymbol{\alpha} = (\boldsymbol{\alpha}(1), \cdots, \boldsymbol{\alpha}(n))$ be the multi-indices, where $\boldsymbol{\alpha}(i) \in \mathbb{Z}_+$, we define $|\boldsymbol{\alpha}| = \sum_{i=1}^n \boldsymbol{\alpha}(i)$ and $\boldsymbol{\alpha}! = \boldsymbol{\alpha}(1)! \cdots \boldsymbol{\alpha}(n)!$. For any vector $\boldsymbol{u} = (\boldsymbol{u}(1), \cdots, \boldsymbol{u}(n)) \in \mathbb{R}^n$, we define $\boldsymbol{u}^{\boldsymbol{\alpha}} = \boldsymbol{u}(1)^{\alpha(1)} \cdots \boldsymbol{u}(n)^{\alpha(n)}$. Let $f : \mathbb{R}^n \to \mathbb{R}$ be a smooth function, we expand its Taylor expansion with Lagrange remainder $R_{K,\boldsymbol{w}}^f(\boldsymbol{u})$ as follows,

$$R_{K,\boldsymbol{w}}^f(\boldsymbol{u}) = f(\boldsymbol{u}) - \sum_{i=0}^K \sum_{|\boldsymbol{\alpha}|=i} \frac{D^{\boldsymbol{\alpha}} f(\boldsymbol{w})}{\boldsymbol{\alpha}!} \cdot (\boldsymbol{u} - \boldsymbol{w})^{\boldsymbol{\alpha}}. \tag{3}$$

Given matrices $\mathbf{Q}$ and $\mathbf{P}$ in $\mathbb{R}^{\ell_1 \times \ell_2}$ we claim that $\mathbf{Q} \leq \mathbf{P}$ if $[\mathbf{Q}]_{i,j} - [\mathbf{P}]_{i,j} \leq 0$ for every $i, j$. For a sequence of vector-valued functions $\{\boldsymbol{F}_i\}_{i=1}^d$, we say that $\{\boldsymbol{F}_i\}_{i=1}^d$ is uniformly $L$-Lipschitz continuous with respect to $\|\cdot\|'$ under $\|\cdot\|$ if $\|\boldsymbol{F}_i(\boldsymbol{x}_i) - \boldsymbol{F}_i(\boldsymbol{u}_i)\|' \leq L\|\boldsymbol{x}_i - \boldsymbol{u}_i\|$ for every $i \in [1:d]$ and any $\boldsymbol{x}, \boldsymbol{u} \in \mathcal{X}$. We denote by $\|\cdot\|_*$ the dual norm of $\|\cdot\|$. Let $\mathbf{P} : \mathbb{R}^{\ell_1 \times \ell_2} \to \mathbb{R}^{n_1 \times n_2}$ be a matrix function, we say that $\mathbf{P}$ is a $\theta$-contraction mapping under $\|\cdot\|$ if $\|\mathbf{P}(\mathbf{Q}_1) - \mathbf{P}(\mathbf{Q}_2)\|_\infty \leq \theta\|\mathbf{Q}_1 - \mathbf{Q}_2\|$ for any $\mathbf{Q}_1, \mathbf{Q}_2 \in \mathbb{R}^{\ell_1 \times \ell_2}$. For matrix-valued function $\mathbf{P} : \mathbb{R}^n \to \mathbb{R}^{\ell_1 \times \ell_2}$, we define $\mathbf{D}_{\mathbf{P}}(\boldsymbol{x}, \boldsymbol{x}') = \mathbf{P}(\boldsymbol{x}) - \mathbf{P}(\boldsymbol{x}')$ for any $\boldsymbol{x}, \boldsymbol{x}' \in \mathbb{R}^n$. The KL divergence $\mathrm{KL}(\boldsymbol{p}\|\boldsymbol{q}) = \sum_{j=1}^n \boldsymbol{p}(j) \cdot \log\left(\frac{\boldsymbol{p}(j)}{\boldsymbol{q}(j)}\right)$ between distributions $\boldsymbol{p}$ and $\boldsymbol{q}$ is defined on probability simplex $\Delta_n$. And the variance of $\boldsymbol{x}$ over $\boldsymbol{p}$ is defined by $\mathrm{Var}_{\boldsymbol{p}}(\boldsymbol{x}) = \sum_{j=1}^n \boldsymbol{p}(j) \cdot (\boldsymbol{x}(j) - \mathbb{E}_{j' \sim \boldsymbol{p}}[\boldsymbol{x}(j')])^2$. We define max-min gap of function

$f : \mathcal{X} \times \mathcal{Y} \to \mathbb{R}$ as follows,

$$\mathcal{G}_f(\boldsymbol{x}, \boldsymbol{y}) := \max_{\boldsymbol{y}' \in \mathcal{Y}} f(\boldsymbol{x}, \boldsymbol{y}') - \min_{\boldsymbol{x}' \in \mathcal{X}} f(\boldsymbol{x}', \boldsymbol{y}). \tag{4}$$

We claim that $(\boldsymbol{x}, \boldsymbol{y})$ is an $\varepsilon$-approximate Nash equilibrium ($\varepsilon$-approximate NE) if $\mathcal{G}_f(\boldsymbol{x}, \boldsymbol{y}) \leq \varepsilon$. When $\varepsilon = 0$, $(\boldsymbol{x}, \boldsymbol{y})$ is a Nash equilibrium.

**Infinite Horizon Discounted Markov Decision Process:** We consider the setting of an infinite horizon discounted Markov decision process (MDP), denoted by $\mathcal{M} := (\mathcal{S}, \mathcal{A}, \mathbb{P}, \sigma, \theta, \boldsymbol{\rho}_0)$. $\mathcal{S}$ is a finite state space; $\mathcal{A}$ is a finite action space; $\mathbb{P}(s|s', a')$ denotes the probability of transitioning from $s$ to $s'$ under playing action $a'$; $\sigma : \mathcal{S} \times \mathcal{A} \to [0, 1]$ is a cost function, which quantifies the cost associated with taking action $a$ in state $s$; $\theta \in [0, 1)$ is a discount factor; $\boldsymbol{\rho}_0$ is an initial state distribution over $\mathcal{S}$.

$\boldsymbol{\pi} : \mathcal{S} \to \Delta_{\mathcal{A}}$ (where $\Delta_{\mathcal{A}}$ is the probability simplex over $\mathcal{A}$) denotes a stochastic policy, i.e., the agent play actions according to $a \sim \boldsymbol{\pi}(\cdot|s)$. We use $\mathbf{Pr}_t^{\boldsymbol{\pi}}(s'|s) = \mathbf{Pr}^{\boldsymbol{\pi}}(s_t = s'|s_0 = s)$ to denote the probability of visiting the state $s'$ from the state $s$ after $t$ time steps according to policy $\boldsymbol{\pi}$. Let trajectory $\tau = \{(s_t, a_t)\}_{t=0}^{\infty}$, where $s_0 \sim \boldsymbol{\rho}_0$, and, for all subsequent time steps $t$, $a_t \sim \boldsymbol{\pi}(\cdot|s_t)$ and $s_{t+1} \sim \mathbb{P}(\cdot|s_t, a_t)$. The value function $V^{\boldsymbol{\pi}} : \mathcal{S} \to \mathbb{R}$ is defined as the discounted sum of future cost starting at state $s$ and executing $\boldsymbol{\pi}$, i.e.

$$V^{\boldsymbol{\pi}}(s) = (1 - \theta)\mathbb{E}\left[\sum_{t=0}^{\infty} \theta^t \sigma(s_t, a_t) \middle| \boldsymbol{\pi}, s_0 = s\right].$$

Moreover, we define the action-value function $Q^{\boldsymbol{\pi}} : \mathcal{S} \times \mathcal{A} \to \mathbb{R}$ and the advantage function $A^{\boldsymbol{\pi}} : \mathcal{S} \times \mathcal{A} \to \mathbb{R}$ as follows:

$$Q^{\boldsymbol{\pi}}(s, a) = (1 - \theta)\mathbb{E}\left[\sum_{t=0}^{\infty} \theta^t \sigma(s_t, a_t) \middle| \boldsymbol{\pi}, s_0 = s, a_0 = a\right], \quad A^{\boldsymbol{\pi}}(s, a) = Q^{\boldsymbol{\pi}}(s, a) - V^{\boldsymbol{\pi}}(s).$$

It's also useful to define the discounted state visitation distribution $\boldsymbol{d}_{s_0}^{\boldsymbol{\pi}}$ of a policy $\boldsymbol{\pi}$ as $\boldsymbol{d}_{s_0}^{\boldsymbol{\pi}}(s) = (1 - \theta)\sum_{t=0}^{\infty} \theta^t \mathbf{Pr}_t^{\boldsymbol{\pi}}(s|s_0)$. In order to simplify notation, we write $\boldsymbol{d}_{\boldsymbol{\rho}_0}^{\boldsymbol{\pi}}(s) = \mathbb{E}_{s_0 \sim \boldsymbol{\rho}_0}[\boldsymbol{d}_{s_0}^{\boldsymbol{\pi}}(s)]$, where $\boldsymbol{d}_{\boldsymbol{\rho}_0}^{\boldsymbol{\pi}}$ is the discounted state visitation distribution under initial distribution $\boldsymbol{\rho}_0$.

## 3 Minimization Optimization

In this section, we propose the generalized quasar-convexity (GQC) condition, and analyze a related algorithmic framework for minimization over $\mathcal{X} = \prod_{i=1}^{d} \Delta_{n_i}$, under mild assumptions.

### 3.1 Generalized Quasar-Convexity (GQC)

We provide a novel depiction of function structure–generalized quasar-convexity, which is defined as follows:

**Definition 3.1** (Generalized Quasar-Convexity (GQC)). Let $\boldsymbol{x}^* \in \mathcal{X} \subset \mathbb{R}^{\sum_{i=1}^{d} n_i}$ be a minimizer of the function $f : \mathcal{X} \to \mathbb{R}$. We say that $f$ is generalized quasar-convex on $\mathcal{X}$ with respect to $\boldsymbol{x}^*$ if for all $\boldsymbol{x} \in \mathcal{X}$, there exist a sequence of vector-valued functions $\{\boldsymbol{F}_i : \mathcal{X} \to \mathbb{R}^{n_i}\}_{i=1}^{d}$ and a sequence of positive scalars $\{\gamma_i\}_{i=1}^{d}$ such that

$$f(\boldsymbol{x}^*) \geq f(\boldsymbol{x}) + \sum_{i=1}^{d} \frac{1}{\gamma_i} \langle \boldsymbol{F}_i(\boldsymbol{x}), \boldsymbol{x}_i^* - \boldsymbol{x}_i \rangle. \tag{5}$$

If Eq. (5) holds, we say that $\boldsymbol{F} = (\boldsymbol{F}_1^\top, \cdots, \boldsymbol{F}_d^\top)^\top$ is the internal function of $f$. Given $i \in [1 : d]$ we say that $\boldsymbol{F}_i$ is the internal function of $f$ for variable block $\boldsymbol{x}_i$.

Our proposed GQC condition concerns the multi-variable generalized extension of the quasar-convexity condition. In the case $d = 1$, the GQC condition degenerates into the $\gamma$-quasar-convexity condition as studied in Hinder et al. [24] with the gradient $\nabla f(\boldsymbol{x})$ belongs to the internal functions of $f$. In the case $d > 1$, the GQC condition is instrumental in capturing the crucial characteristic of those optimization applications with each variable block has difficulty to be optimized.

**Algorithm 1** Optimistic Mirror Descent for Multi-Distributions

**Input:** $\left\{ \boldsymbol{g}_i^0 = \boldsymbol{x}_i^0 = (1/n_i, \cdots, 1/n_i) \right\}_{i=1}^d$, $\eta$ and $T$.
**Output:** Randomly pick up $t \in \{1, \cdots, T\}$ following the probability $\mathbb{P}[t] = 1/T$ and return $\boldsymbol{x}^t$.
1: **while** $t \leq T$ **do**
2:     **for all** $i \in [1 : d]$ **do**
3:         $\boldsymbol{x}_i^t = \underset{\boldsymbol{x}_i \in \Delta_{n_i}}{\operatorname{argmin}} \ \eta \left\langle \boldsymbol{F}_i(\boldsymbol{x}^{t-1}), \boldsymbol{x}_i \right\rangle + \mathrm{KL}\left( \boldsymbol{x}_i \, \| \boldsymbol{g}_i^{t-1} \right)$,
4:         $\boldsymbol{g}_i^t = \underset{\boldsymbol{g}_i \in \Delta_{n_i}}{\operatorname{argmin}} \ \eta \left\langle \boldsymbol{F}_i(\boldsymbol{x}^t), \boldsymbol{g}_i \right\rangle + \mathrm{KL}\left( \boldsymbol{g}_i \, \| \boldsymbol{g}_i^{t-1} \right)$.
5:     **end for**
6:     $t \leftarrow t + 1$.
7: **end while**

## 3.2 Main Results

Recall that GQC condition provides a perspective to bound function error $f(\boldsymbol{x}) - f(\boldsymbol{x}^*)$ based on internal function, which is different from that based on gradient oracle. We therefore aim to provide an algorithmic framework for finding an approximate suboptimal global solution using internal function. Given an objective function $f : \mathcal{X} \to \mathbb{R}$ with internal function $\boldsymbol{F}$, our algorithm (Algorithm 1) independently computes points $\boldsymbol{g}_i^t$ and $\boldsymbol{x}_i^t$ following OMD over each block. If $\max_{i \in [1:d]} \gamma_i < \infty$ and internal function $\boldsymbol{F}$ has Lipschitz continuity, we have following basic and primary convergence result of Algorithm 1,

**Theorem 3.2.** *Assuming that $\boldsymbol{F}$ is $L$-Lipschitz continuous with respect to $\| \cdot \|_*$ under $\| \cdot \|$ and $\gamma_{\max} = \max_{i \in [1:d]} \gamma_i < \infty$, and setting $\eta = (L^2 d \gamma_{\max} \sum_{i=1}^d \gamma_i^{-1})^{-1/2}/2$, we have*

$$\frac{1}{T} \sum_{t=1}^T (f(\boldsymbol{x}^t) - f(\boldsymbol{x}^*)) \leq \frac{2L \max_{i \in [1:d]} \log(n_i) \, (d\gamma_{\max})^{1/2} \left( \sum_{i=1}^d \gamma_i^{-1} \right)^{3/2}}{T}. \tag{6}$$

However, the estimation provided by Theorem 3.2 depends on $d\gamma_{\max}$. And the step size relying on $\gamma_{\max} \left( \sum_{i=1}^d \gamma_i^{-1} \right)$ might be difficult to set when $\{\gamma_i\}_{i=1}^d$ is unknown.

We then hope to propose an alternative analytical method that can adapt to unknown $\{\gamma_i\}_{i=1}^d$ and obtain complexity which does not depends on block dimension $d$ explicitly. The challenges includes: 1) The algorithm does not know the weight $1/\gamma_i$; 2) every $\boldsymbol{F}_i$ has dependence on the joint variable $\boldsymbol{x}$ instead of depending on $\boldsymbol{x}_i$. Before we present the details of convergence analysis, we need the following notations and assumptions:

Denote $P_{K,\boldsymbol{y}}^f(\boldsymbol{x})) = \sum_{i=0}^K \sum_{|\boldsymbol{\alpha}|=i} \frac{|D^{\boldsymbol{\alpha}} f(\boldsymbol{y})|}{\boldsymbol{\alpha}!} \cdot (|\boldsymbol{x}| + |\boldsymbol{y}|)^{\boldsymbol{\alpha}}$ and let $\boldsymbol{P}_{K,\boldsymbol{y}}^{\boldsymbol{\phi}}(\boldsymbol{x}) = (P_{K,\boldsymbol{y}}^{\phi^{(1)}}(\boldsymbol{x}), \cdots,$ $P_{K,\boldsymbol{y}}^{\phi^{(\ell)}}(\boldsymbol{x}))$ for any vector-valued function $\boldsymbol{\phi} : \mathbb{R}^n \to \mathbb{R}^\ell$. Recalling the definition of $R_{K,\boldsymbol{w}}^f$ in Eq. (3), we shall also define $\boldsymbol{R}_{K,\boldsymbol{y}}^{\boldsymbol{\phi}}(\boldsymbol{x}) = (R_{K,\boldsymbol{y}}^{\phi^{(1)}}(\boldsymbol{x}), \cdots, R_{K,\boldsymbol{y}}^{\phi^{(\ell)}}(\boldsymbol{x}))$.

**Assumption 3.3.** Let $\boldsymbol{F}$ be the internal function of $f$. There exists $\Theta_1, \Theta_2 > 0$, $K_0 \in \mathbb{Z}_+$, and $\theta \in [0, 1)$, and a fixed $\boldsymbol{y} \in \mathbb{R}^{\sum_{i=1}^d n_i}$ such that

**[A₁]** $\left\| \boldsymbol{R}_{K,\boldsymbol{y}}^{\boldsymbol{F}}(\boldsymbol{x}) \right\|_\infty \leq \Theta_1 \theta^K$ for any integer $K > K_0$ and $\boldsymbol{x} \in \mathcal{X}$.

**[A₂]** $\left\| \boldsymbol{P}_{K,\boldsymbol{y}}^{\boldsymbol{F}}(\boldsymbol{x}) \right\|_\infty \leq \Theta_2$ for any integer $K \in \mathbb{Z}_+$ and $\boldsymbol{x} \in \mathcal{X}$.

Assumption 3.3 is a characterization of "polynomial-like" functions. We clarify this view as follows. For a standard polynomial function $p$, it's clear that $p$ satisfies Assumption 3.3, since the Taylor expansion of $p$ after order $K_0$ is always equal to 0 (**[A₁]** in Assumption 3.3 holds) and $\mathcal{X}$ is a bounded and closed set (**[A₂]** in Assumption 3.3 holds). Assumption 3.3 is easy to achieve. Shown in Proposition B.2 and Remark B.3 in Appendix B, Assumption 3.3 can be satisfied by many smooth functions defined on bounded region $\mathcal{X}$. In addition, we introduce a simple machine learning example: learning one single neuron network over a simplex in the realizable setting.

*Example* 3.4. The objective function is written as $f(\boldsymbol{p}, \mathbf{P}) = \frac{1}{2}\mathbb{E}_{\boldsymbol{x},y}(\sum_{i=1}^m \boldsymbol{p}_i\sigma(\boldsymbol{x}^\top\mathbf{P}_i) - y)^2$, where $\boldsymbol{p} \in \Delta_m$ and $\mathbf{P} = (\mathbf{P}_1, \cdots, \mathbf{P}_m) \in \prod_{i=1}^m \Delta_d$ and the target $y$ given $\boldsymbol{x} \in [-C, C]^d$ admits $y = \sigma(\boldsymbol{x}^\top\mathbf{P}_1^*)$ for some $\mathbf{P}_1^* \in \Delta_d$. For activation function $\sigma(x) = \exp\{x\}$, $f$ satisfies GQC condition and Assumption 3.3 with the internal functions $\boldsymbol{F_p} = \{\mathbb{E}[(\sum_{j=1}^m \boldsymbol{p}_j\sigma(\boldsymbol{x}^\top\mathbf{P}_j) - y)\sigma(\boldsymbol{x}^\top\mathbf{P}_i)]\}_{i=1}^m$ for block $\boldsymbol{p}$ and $\boldsymbol{F}_{\mathbf{P}_i} = \mathbb{E}[(\sigma(\boldsymbol{x}^\top\mathbf{P}_i) - y)\boldsymbol{x}]$ for block $\mathbf{P}_i$.

Note previous work [65] studies single neuron learning by considering $\mathbf{P}_1^*$ in the sphere and assuming $\boldsymbol{x}$ follows from a Gaussian distribution. To our knowledge, there is no evidence shows that objective function of Example 3.4 has quasar-convexity. This example demonstrates the advantage of studying the GQC framework over the previous approach. The proof of Example 3.4 is in Section B.2.

**Parameter Setting** Before stating the convergence result, we set the parameters as follows:

$$\Theta = \Theta_1 + \Theta_2 + 1, \quad H = \lceil\log(T)\rceil, \quad \beta_0 = (4H)^{-1}, \quad \beta = \min\left\{\frac{\sqrt{\beta_0/8}}{H^3}, \frac{1}{2\Theta(H+3)}\right\},$$

$$\Gamma = e^2 + \mathcal{O}(\Theta_2), \quad \hat{K} = \max\left\{\frac{H\log(4\beta^{-1}) + \log(\Theta_1)}{\log(\theta^{-1})}, K_0\right\}, \quad \eta = \min\left\{\frac{\beta}{6e^3\hat{K}\Gamma\max\{\Theta, 1\}}, \frac{\beta_0^4}{\mathcal{O}(\Theta)}\right\}. \tag{7}$$

**Theorem 3.5.** *Let $f$ satisfies the GQC condition and denote $N = \max_{i\in[1:d]}\{n_i\}$. Under Assumption 3.3, the following estimation holds for Algorithm 1's output $\{\boldsymbol{x}^t\}_{t=1}^T$*

$$\frac{1}{T}\sum_{t=1}^T (f(\boldsymbol{x}^t) - f(\boldsymbol{x}^*)) \leq \left(\sum_{i=1}^d 1/\gamma_i\right)\left[\frac{1}{\eta}\log(N) + \eta\Theta^3(6 + 330240\Theta H^5)\right]T^{-1}, \tag{8}$$

Theorem 3.5 implies that for any generalized quasar-convex function $f$ satisfies Assumption 3.3, the $T$-step random solution outputted by Algorithm 1 is a $\mathcal{O}((\sum_{i=1}^d 1/\gamma_i)T^{-1}\log(N)\log^{4.5}(T))$-suboptimal solution. Ignoring the logarithmic factor, the iteration complexity of our algorithm is competitive to the state-of-the-art algorithm when applied to specific application (i.e. policy optimization of reinforcement learning [2]). Moreover, our algorithm makes iteration complexity depend on $\sum_{i=1}^d 1/\gamma_i$ linearly. In some common applications, $\sum_{i=1}^d 1/\gamma_i$ has no dependence on $d$, which is the number of variable blocks (see discussions in Section 3.3).

### 3.3 Application to Reinforcement Learning

This section reveals that GQC condition provides a novel analytical approach to reinforcement learning. We show how to leverage Algorithm 1 to find $\varepsilon$-suboptimal global solution for infinite horizon reinforcement learning problem. And in Appendix B.3.2, we show how to leverage Algorithm 1 to minimize finite horizon reinforcement learning problem.

The infinite horizon reinforcement learning is formulated as the following policy optimization problem:

$$\min_{\boldsymbol{\pi}\in\mathcal{X}} J^{\boldsymbol{\pi}}(\boldsymbol{\rho}_0), \tag{9}$$

where $J^{\boldsymbol{\pi}}(\boldsymbol{\rho}_0) = \mathbb{E}_{s_0\sim\boldsymbol{\rho}_0}[V^{\boldsymbol{\pi}}(s_0)]$ and $\mathcal{X} = \prod_{i=1}^{|\mathcal{S}|}\Delta_{\mathcal{A}}$ denotes $|\mathcal{S}|$ probability simplexes. We write $\mathcal{S} = \{s_i\}_{i=1}^{|\mathcal{S}|}$ and denote the action-value vector on state $s_i$ by $\boldsymbol{Q}^{\boldsymbol{\pi}}(s_i, \cdot)$. The next Proposition 3.6 states that $J^{\boldsymbol{\pi}}(\boldsymbol{\rho}_0)$ satisfies the GQC condition for any initial state distribution $\boldsymbol{\rho}_0$.

**Proposition 3.6.** *Let $\{\boldsymbol{\pi}^*(\cdot|s) \in \Delta_{\mathcal{A}}\}_{s\in\mathcal{S}}$ denote the optimal global solution of problem* (9). *We have that $J^{\boldsymbol{\pi}}(\boldsymbol{\rho}_0)$ satisfies the GQC condition in Eq.* (5) *with internal function $\boldsymbol{F}_i(\boldsymbol{\pi}) = \boldsymbol{Q}^{\boldsymbol{\pi}}(s_i, \cdot)$ for variable block $\boldsymbol{\pi}_i$ and $\boldsymbol{F}$ satisfies Assumption 3.3 with $\Theta_1 = \theta, \Theta_2 = 1$ and $K_0 = 1$.*

According to Theorem 3.5, if we apply Algorithm 1 to the infinite horizon reinforcement learning basing action-value vector $\mathbf{Q}^{\boldsymbol{\pi}}$ with parameter selection Eq. (7), which is actually a simple variant of natural policy gradient descent [2], then the iterations $T$ we need to find an $\varepsilon$-suboptimal global solution is upper-bounded by $\mathcal{O}(\max\{1, \log^{-1}(\theta^{-1})\}(1-\theta)^{-1}\varepsilon^{-1}\log^{4.5}(\varepsilon^{-1})\log(|\mathcal{A}|))$ under Agarwal et al. [2]'s setting. Therefore, the iteration complexity of Algorithm 1 does not depend on the size of states, since the summation of $\boldsymbol{d}_{\boldsymbol{\rho}_0}^{\boldsymbol{\pi}^*}$ over $\mathcal{S}$ ($\sum_{i=1}^{|\mathcal{S}|} 1/\gamma_i = \sum_{i=1}^{|\mathcal{S}|}\boldsymbol{d}_{\boldsymbol{\rho}_0}^{\boldsymbol{\pi}^*}(s_i) = 1$) mollifies the accumulation

of the maximum of $d_{\boldsymbol{\rho}_0}^{\boldsymbol{\pi}^*}$ over $\mathcal{S}$ with $|\mathcal{S}|$ times. Specifically, if we take into account the loosest upper bound $|\mathcal{S}| \max_{i \in [1:|\mathcal{S}|]} d_{\boldsymbol{\rho}_0}^{\boldsymbol{\pi}^*}(s_i)$, then the iteration complexity of algorithm may suffer from the linear dependence on $|\mathcal{S}|$, since $\max_{i \in [1:|\mathcal{S}|]} d_{\boldsymbol{\rho}_0}^{\boldsymbol{\pi}^*}(s_i) \geq (1 - \theta) \max_{i \in [1:|\mathcal{S}|]} \boldsymbol{\rho}_0(s_i)$. Previous research [2, Theorem 5.3] has demonstrated that utilizing the information of joint variables to separately update each variable block ensures global convergence for problem (9) with $\mathcal{O}((1 - \theta)^{-2} \varepsilon^{-1})$ iteration complexity. However, their analytical approach is carefully designed for infinite horizon reinforcement learning problems.

# 4 Minimax Optimization

In this section, we introduce the generalized quasar-convexity-concavity (GQCC) condition, which can be verified in real applications such as two-player zero-sum Markov games. We provide a related algorithm for minimax optimization (minimizing $\mathcal{G}_f(\boldsymbol{x}, \boldsymbol{y})$ has been defined in Eq. (4)) over $\mathcal{Z} = \prod_{i=1}^d \mathcal{Z}_i = \prod_{i=1}^d (\Delta_{n_i} \times \Delta_{m_i})$, under proper assumptions. We specify the divergence-generating function $v$ as $v(\boldsymbol{x}) = \mathbb{E}_{i \sim \boldsymbol{x}(\cdot)}[\log(\boldsymbol{x}(i))]$ in probability simplexes setting. We also provide a framework for minimax problem over the general compact convex regions in Appendix C.

## 4.1 Generalized Quasar-Convexity-Concavity (GQCC)

We provide a new notion called generalized quasar-convexity-concavity for nonconvex-nonconcave minimax optimization, which is defined as follows:

**Definition 4.1** (Generalized Quasar-Convexity-Concavity (GQCC)). Denote $\mathcal{Z}_i = \mathcal{X}_i \times \mathcal{Y}_i$ for any $i \in [1:d]$, and let $f : \mathcal{Z} \to \mathbb{R}$ be the objective function. We say that $f$ is generalized quasar-convex-concave on $\mathcal{Z}$ if for all $\boldsymbol{z} = (\boldsymbol{x}, \boldsymbol{y}) \in \mathcal{Z}$, there exist a sequence of functions $\{f_i : \mathbb{R}^{\ell \times d} \times \mathcal{Z}_i \to \mathbb{R}\}_{i=1}^d$, a sequence of non-negative functions $\{\psi_i : \mathcal{Z} \to \mathbb{R}_+ \cup 0\}_{i=1}^d$ and a matrix-valued function $\mathbf{P} = (\mathbf{P}_1, \cdots, \mathbf{P}_d) : \mathcal{Z} \to \mathbb{R}^{\ell \times d}$ where every $\mathbf{P}_i$ is a $\ell$-dimensional vector-valued function, such that

$$\mathcal{G}_f(\boldsymbol{x}, \boldsymbol{y}) \leq \sum_{i=1}^d \psi_i(\boldsymbol{z}) \mathcal{G}_{f_i(\mathbf{P}(\boldsymbol{z}), \cdot, \cdot)}(\boldsymbol{x}_i, \boldsymbol{y}_i), \tag{10}$$

where each $f_i(\mathbf{Q}, \cdot, \cdot)$ is convex-concave for a fixed $\mathbf{Q} = (\mathbf{Q}_1, \cdots, \mathbf{Q}_d) \in \mathbb{R}^{\ell \times d}$. We denote the internal operator of $f$ for variable block $\boldsymbol{z}_i$ by $\boldsymbol{F}_i$ where $\boldsymbol{F}_i(\mathbf{Q}, \boldsymbol{z}_i) = ((\nabla_{\boldsymbol{x}_i} f_i(\mathbf{Q}, \boldsymbol{z}_i))^\top, (-\nabla_{\boldsymbol{y}_i} f_i(\mathbf{Q}, \boldsymbol{z}_i))^\top)^\top$. Moreover, we say that $\boldsymbol{F} = (\boldsymbol{F}_1^\top, \cdots, \boldsymbol{F}_d^\top)^\top$ is the internal operator of $f$.

The GQCC condition is an extension of the GQC condition in minimax optimization setting. The specific connection between them can be found in Appendix C. The GQCC condition can be viewed as an extension of the convexity-concavity condition in multi-variable optimization; it seamlessly reduces to the convexity-concavity condition with $f_1(\mathbf{P}(\boldsymbol{z}), \boldsymbol{z}) = f(\boldsymbol{z})$ and $\psi_1(\boldsymbol{z}) \equiv 1$, in the case $d = 1$. Assuming every $\psi_i$ is bounded, $f_i(\mathbf{P}(\boldsymbol{z}), \boldsymbol{z}_i) \equiv f_i(\mathbf{0}, \boldsymbol{z}_i)$ with Lipschitz continuous gradient and is convex-concave with respect to $\boldsymbol{z}_i$, then finding the Nash equilibrium point of $f$ is reduced to finding the Nash equilibrium points of $d$ independent convex-concave minimax problems. However, how to find the approximate Nash equilibrium points in more general case has not been well-studied. Most of existing work for minimax optimization without convex-concave assumption are focused on finding the approximate stationary points.

## 4.2 Main Results

For simplicity, we denote by $\boldsymbol{F}_i^{\boldsymbol{x}}$ and $\boldsymbol{F}_i^{\boldsymbol{y}}$ the projection of $\boldsymbol{F}_i$ in the $\boldsymbol{x}_i$ and $\boldsymbol{y}_i$ directions, respectively, i.e., $\boldsymbol{F}_i^\top = ((\boldsymbol{F}_i^{\boldsymbol{x}})^\top, (\boldsymbol{F}_i^{\boldsymbol{y}})^\top)$. Given an objective function $f : \mathcal{Z} \to \mathbb{R}$ with internal operator $\boldsymbol{F}$, our algorithm (Algorithm 2) employs regularized OMD over each distribution independently basing on $\boldsymbol{F}_i$ and updates matrix $\mathbf{Q}^t$ to track the behavior of function $\mathbf{P}$ iteratively. It's worth noting that each iteration of Algorithm 2 provides explicit expressions for $\boldsymbol{x}_i^t$ and $\boldsymbol{g}_i^t$ (see the proof of Theorem 3.5 in Appendix B). Consequently, Algorithm 2 essentially operates as a single-loop algorithm.

**Assumption 4.2.** In Definition 4.1, we assume that matrix-valued function $\mathbf{P}$ has the form of $\mathbf{P}(\mathbf{Q}^{\boldsymbol{z}}, \boldsymbol{z})$ where $\mathbf{Q}^{\boldsymbol{z}} \in \mathbb{R}^{\ell \times d}$ depends on $\boldsymbol{z}$, and $\mathbf{P}$ satisfies the following properties on region $\{\mathbf{Q} \in \mathbb{R}^{\ell \times d} \mid \|\mathbf{Q}\|_\infty \leq C\} \times \mathcal{Z}$ for some constant $C > 0$:

**Algorithm 2** Optimistic Mirror Descent with Regularization for Multiple Distributions

---

**Input:** $\{z_i^0\}_{i=1}^d = \{g_i^0\}_{i=1}^d = \{(1/n_i, \cdots, 1/n_i), (1/m_i, \cdots, 1/m_i)\}_{i=1}^d, \{\alpha_t \geq 0\}_{t=1}^T$ with $\sum_{t=1}^T \alpha_t = 1$, $\{\gamma_t \geq 0\}_{t=1}^T, \{\lambda_t \geq 0\}_{t=1}^T, \eta$ and $\mathbf{Q}^0 = \mathbf{0}$.

**Output:** $\bar{z}_T = \sum_{t=1}^T \alpha_t z^t$.

1: **while** $t \leq T$ **do**
2:     $\mathbf{Q}^t = (1 - \beta_{t-1})\mathbf{Q}^{t-1} + \beta_{t-1}\mathbf{P}(\mathbf{Q}^{t-1}, z_{t-1})$.
3:     **for all** $i \in [1:d]$ **do**
4:         $x_i^t = \underset{x_i \in \mathcal{X}_i}{\operatorname{argmin}}\ \eta\langle F_i^x(\mathbf{Q}^{t-1}, z_i^{t-1}), x_i\rangle + \gamma_t \mathrm{KL}\left(x_i \left\| (g_i^x)^{t-1}\right.\right) + \lambda_t v(x_i)$,
5:         $y_i^t = \underset{y_i \in \mathcal{Y}_i}{\operatorname{argmin}}\ \eta\langle F_i^y(\mathbf{Q}^{t-1}, z_i^{t-1}), y_i\rangle + \gamma_t \mathrm{KL}\left(y_i \left\| (g_i^y)^{t-1}\right.\right) + \lambda_t v(y_i)$,
6:         $(g_i^x)^t = \underset{g_i^x \in \mathcal{X}_i}{\operatorname{argmin}}\ \eta\langle F_i^x(\mathbf{Q}^t, z_i^t), g_i^x\rangle + \gamma_t \mathrm{KL}\left(g_i^x \left\| (g_i^x)^{t-1}\right.\right) + \lambda_t v(g_i^x)$,
7:         $(g_i^y)^t = \underset{g_i^y \in \mathcal{Y}_i}{\operatorname{argmin}}\ \eta\langle F_i^y(\mathbf{Q}^t, z_i^t), g_i^y\rangle + \gamma_t \mathrm{KL}\left(g_i^y \left\| (g_i^y)^{t-1}\right.\right) + \lambda_t v(g_i^y)$.
8:     **end for**
9:     $t \leftarrow t + 1$.
10: **end while**

---

[**A₁**] There exist constants $L_1, L_2 \geq 0$ such that $F_i(\cdot, z_i)$ is uniformly $L_1$-Lipschitz continuous with respect to $\|\cdot\|_\infty$ under $\|\cdot\|_\infty$, and $F_i(\mathbf{Q}, \cdot)$ is uniformly $L_2$-Lipschitz continuous with respect to $\|\cdot\|_\infty$ under $\|\cdot\|_1$.

[**A₂**] There are a positive constant $\gamma > 0$ and a set of non-negative constant matrices $\{\mathbf{B}_i, \mathbf{C}_i\}_{i=1}^d$ satisfying $\left\|\sum_{i=1}^d(\mathbf{B}_i + \mathbf{C}_i)\right\|_\infty \leq \gamma$, such that $\mathbf{D}_{\mathbf{P}(\mathbf{Q},\cdot,y)}(x, x') \leq \sum_{i=1}^d \mathbf{C}_i\langle F_i^x(\mathbf{Q}, z_i), x_i - x_i'\rangle$ and $\mathbf{D}_{\mathbf{P}(\mathbf{Q},x,\cdot)}(y, y') \geq \sum_{i=1}^d \mathbf{B}_i\langle F_i^y(\mathbf{Q}, z_i), y_i' - y_i\rangle$.

[**A₃**] There exists $\theta \in [0, 1)$ such that $\mathbf{P}(\cdot, z)$ is a $\theta$-contraction mapping under $\|\cdot\|_\infty$, and $\|\mathbf{P}(\mathbf{Q}, z)\|_\infty \leq C$ for any $z \in \mathcal{Z}$.

We present Lemma 4.3 to demonstrate that there exist $\mathbf{Q}^* \in \mathbb{R}^{\ell \times d}, x^* \in \mathcal{X}$ and $y^* \in \mathcal{Y}$ satisfy the saddle point and fixed point conditions of function $\mathbf{P}$, i.e., Eq. (11), under proper assumptions.

**Lemma 4.3.** *Assuming that Assumption 4.2 holds, $[\mathbf{P}(\mathbf{Q}, \cdot, \cdot)]_{k,j}$ is continuous, convex with respect to $x$, concave with respect to $y$ for any $(k, j)$, and $\min_{k,j,i} \frac{\min\{[\mathbf{C}_i]_{k,j}, [\mathbf{B}_i]_{k,j}\}}{[\mathbf{C}_i]_{k,j} + [\mathbf{B}_i]_{k,j}} \geq C'$ for some $C' > 0$, then there exist $\mathbf{Q}^* \in \mathbb{R}^{\ell \times d}$ and $z^* \in \mathcal{Z}$ such that*

$$\mathbf{Q}^* = \mathbf{P}(\mathbf{Q}^*, x^*, y^*), \quad \mathbf{Q}^* \leq \mathbf{P}(\mathbf{Q}^*, x, y^*), \quad and \quad \mathbf{Q}^* \geq \mathbf{P}(\mathbf{Q}^*, x^*, y). \tag{11}$$

For Algorithm 2, we let $\beta_{T,t} = \beta_t \prod_{j=t+1}^T (1 - \beta_j)$ for any $T \geq t$ and $\beta_{T,T} = \beta_T$, and set parameters

$$c = 2(1-\theta)^{-1}, \ \eta \leq \frac{(1-\theta)^{1/2}}{16L_2((\gamma L_1)^{1/2} + 1)}, \ \beta_t = \frac{c}{c+t}, \ \alpha_t = \beta_{T,t}, \ \gamma_t = \frac{\alpha_{t-1}}{\alpha_t}, \ \lambda_t = 1 - \gamma_t. \tag{12}$$

Then we have the following convergence result by denoting $M = \max_{i \in [1:d]} \{m_i + n_i\}$.

**Theorem 4.4.** *For any generalized quasar-convex-concave function $f$ which satisfies Assumption 4.2 with $\mathbf{P} \equiv \mathbf{Q}^*$, where $\mathbf{Q}^*$ satisfies Eq. (11). Algorithm 2's output $\bar{z}_T = (\bar{x}_T, \bar{y}_T)$ satisfies*

$$\mathcal{G}_f(\bar{x}_T, \bar{y}_T) \leq 60 \max_{z \in \mathcal{Z}}\left(\sum_{i=1}^d \psi_i(z)\right)(1-\theta)^{-1}\left(\frac{2}{\eta}\log(M) + \eta L_1^2 + L_1 Y_T^\eta\right)T^{-1},$$

*where $Y_T^\eta = 8(c+1)[\frac{4\gamma}{\eta}\log(M) + 160\gamma L_2 + 2\eta\gamma L_1^2(1 + 64C^2)](\log(c+T) + 1)$.*

Similar to minimization Algorithm 1, the iteration complexity of minimax Algorithm 2 linearly depends on the upper bound of $\sum_{i=1}^d \psi_i$ over $\mathcal{Z}$. Generally, the upper bound of $\sum_{i=1}^d \psi_i$ on $\mathcal{Z}$ is related to $d$. In specific problems of multi-variable optimization (such as two-player zero-sum Markov games), one can uniformly bound $\sum_{i=1}^d \psi_i$ on $\mathcal{Z}$ by a constant.

## 4.3 Application to Infinite Horizon Two-Player Zero-Sum Markov Games

In this section, we show how to leverage Algorithm 2 to achieve accelerated rates for optimizing infinite horizon two-player zero-sum Markov games. Our algorithm use $\tilde{\mathcal{O}}(\varepsilon^{-1})$ iteration bound to find an $\varepsilon$-approximate Nash equilibrium of infinite horizon two-player zero-sum Markov games.

As similar as the definition of discounted MDP in Preliminary, we utilize $\mathcal{M} = (\mathcal{S}, \mathcal{A}, \mathcal{B}, \mathbb{P}, \sigma, \theta, \boldsymbol{\rho}_0)$ to define a infinite horizon two-player zero-sum Markov game. The difference here compared to Section 3.3 is that the cost function $\sigma$ is defined on $\mathcal{S} \times \mathcal{A} \times \mathcal{B}$ with values in $[0, 1]$, and the transition model $\mathbb{P}(s|s', a', b')$ denotes the probability of transitioning into state $s$ upon player 1 taking action $a'$ and player 2 taking action $b'$ in state $s'$. We can define the value function $V^{\boldsymbol{z}}$ and action-value function $Q^{\boldsymbol{z}}$ on the joint distribution $\boldsymbol{z} = (\boldsymbol{x}, \boldsymbol{y}) \in \mathcal{Z} = \prod_{i=1}^{|\mathcal{S}|} \Delta_{\mathcal{A}} \times \prod_{i=1}^{|\mathcal{S}|} \Delta_{\mathcal{B}}$. The infinite horizon two-player zero-sum Markov games consider the following policy optimization problem:

$$\min_{\boldsymbol{x} \in \mathcal{X}} \max_{\boldsymbol{y} \in \mathcal{Y}} J^{\boldsymbol{x}, \boldsymbol{y}}(\boldsymbol{\rho}_0), \tag{13}$$

where $J^{\boldsymbol{z}}(\rho_0) = \mathbb{E}_{s_0 \sim \boldsymbol{\rho}_0}[V^{\boldsymbol{z}}(s_0)]$. The following proposition indicates that $J^{\boldsymbol{z}}$ is general quasar convex-concave, and satisfies Assumption 4.2 and the condition of Theorem 4.4,

**Proposition 4.5.** *For any* $\mathbf{Q} = (\mathbf{Q}_1, \cdots, \mathbf{Q}_{|\mathcal{S}|})$ *with every* $\mathbf{Q}_i \in \mathbb{R}^{|\mathcal{A}| \times |\mathcal{B}|}$, *define function* $f_i(\mathbf{Q}, \boldsymbol{z}_i) := \boldsymbol{x}_i^\top \mathbf{Q}_i \boldsymbol{y}_i$ *for any* $i \in [1 : |\mathcal{S}|]$. *There exists a tensor-valued function* $\mathbf{P}$ *such that* $J^{\boldsymbol{z}}(\boldsymbol{\rho}_0)$ *satisfies GQCC condition with* $f_i(\mathbf{P}(\boldsymbol{z}), \boldsymbol{z}_i) = f_i(\mathbf{Q}^*, \boldsymbol{z}_i)$ *for any* $\boldsymbol{\rho}_0 \in \Delta_{\mathcal{S}}$, *where* $\mathbf{Q}^*$ *satisfies the conditions mentioned in Eq. (11). Moreover,* $\mathbf{P}$ *satisfies Assumption 4.2.*

According to Proposition 4.5 and Theorem 4.4, if we apply Algorithm 2 to the infinite horizon two-player Markov games basing internal operator $\boldsymbol{F}_i(\mathbf{Q}, \boldsymbol{z}) = (\boldsymbol{y}_i^\top \mathbf{Q}_i^\top, -\boldsymbol{x}_i^\top \mathbf{Q}_i)^\top$ for block $\boldsymbol{z}_i$ with parameter selection Eq. (12), which is actually a variant of optimistic gradient descent/ascent for Markov games [67], then the iterations $T$ we need to find an $\varepsilon$-approximate Nash equilibrium is upper-bounded by $\tilde{\mathcal{O}}((1-\theta)^{-2.5}\varepsilon^{-1})$. To the best of our knowledge, our iteration bound matches state-of-the-art iteration bound and is a factor of $(1-\theta)^{-1.5}|\mathcal{S}|$ better than $\tilde{\mathcal{O}}((1-\theta)^{-4}|\mathcal{S}|\varepsilon^{-1})$ bound of Cen et al. [10]. Since the upper bound of $\sum_{i=1}^{|\mathcal{S}|} \psi_i$ over feasible region $\mathcal{Z}$ in infinite horizon two-player zero-sum Markov games' setting satisfies $\sum_{i=1}^{|\mathcal{S}|} \psi_i(\boldsymbol{z}) \leq \sum_{i=1}^{|\mathcal{S}|}[\boldsymbol{d}_{\boldsymbol{\rho}_0}^{\boldsymbol{x}, \boldsymbol{y}^*(\boldsymbol{x})}(s_i) + \boldsymbol{d}_{\boldsymbol{\rho}_0}^{\boldsymbol{x}^*(\boldsymbol{y}), \boldsymbol{y}}(s_i)] \leq 2$ for any $\boldsymbol{z} \in \mathcal{Z}$, our algorithm's iteration bound does not depend on the size of states.

## 5 Conclusion

In this work, we introduce two function structures: GQC and GQCC and provide related algorithmic frameworks with convergence result. To complement our result, we also show that discounted MDP and infinite horizon two-player zero-sum Markov games admit the GQC and GQCC condition, respectively, and satisfy our mild assumptions.

## 6 Acknowledgements

C. Fang was supported by National Key R&D Program of China (2022ZD0114902) and the NSF China (No.62376008). L. Luo was supported by National Natural Science Foundation of China (No. 62206058), Shanghai Sailing Program (22YF1402900), Shanghai Basic Research Program (23JC1401000), and the Major Key Project of PCL under Grant PCL2024A06.

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

# Contents

## A   Preliminary

### A.1   Supplemental Notation

For simplicity, we denote $g(\Gamma) := \sum_{k=1}^{\infty} \Gamma^{-k}[k^7 + (k+1)\exp\{2k\}]$, the chi-squared divergence between $\boldsymbol{p}, \boldsymbol{q}$ as $\chi^2(\boldsymbol{p}\|\boldsymbol{q}) := \sum_{j=1}^{n} \frac{(\boldsymbol{p}(j)-\boldsymbol{q}(j))^2}{\boldsymbol{q}(j)}$, $\mathbb{E}_{\boldsymbol{p}}(\boldsymbol{x}) := \sum_{j=1}^{n} \boldsymbol{p}(j)\boldsymbol{x}(j)$ and $\mathrm{Var}_{\boldsymbol{p}}(\boldsymbol{x}) := \sum_{j=1}^{n} \boldsymbol{p}(j) \cdot (\boldsymbol{x}(j) - \mathbb{E}_{\boldsymbol{p}}(\boldsymbol{x}))^2$ for any $\boldsymbol{p}, \boldsymbol{q} \in \Delta_n$ and $\boldsymbol{x} \in \mathbb{R}^n$. For $\zeta > 0, n \in \mathbb{Z}_+$, we say that a sequence of distributions $\boldsymbol{p}^1, \cdots, \boldsymbol{p}^T \in \Delta_n$ is $\zeta$-consecutively close if for each $1 \le t < T$, it holds that $\max\left\{\left\|\frac{\boldsymbol{p}^t}{\boldsymbol{p}^{t+1}}\right\|, \left\|\frac{\boldsymbol{p}^{t+1}}{\boldsymbol{p}^t}\right\|\right\} \le 1 + \zeta$. For positive scalar $\theta \in [0, 1)$, non-negative integers $t$ and $T$, we define $\beta_{T,t}^{\theta} := \beta_t \prod_{j=t}^{T-1}(1 - \beta_j + \theta\beta_j)$, and $\beta_{T,T}^{\theta} = 1$.

### A.2   Finite Differences

**Definition A.1** (Finite Differences). For a sequence of vectors $\boldsymbol{L} = (\boldsymbol{L}^0, \cdots, \boldsymbol{L}^T)$ where each $\boldsymbol{L}^t \in \mathbb{R}^n$, and integers $h \in \mathbb{Z}_+$, the order-$h$ finite difference sequence for the sequence $\boldsymbol{L}$ is denoted by $D_h\boldsymbol{L} := ((D_h\boldsymbol{L})^0, \cdots, (D_h\boldsymbol{L})^{T-h})$ recursively with $(D_0\boldsymbol{L})^t := \boldsymbol{L}^t$ for all $t \in [0:T]$, and

$$(D_h\boldsymbol{L})^t := (D_{h-1}\boldsymbol{L})^{t+1} - (D_{h-1}\boldsymbol{L})^t, \tag{14}$$

for all $h \ge 1$ and $t \in [1:T-h]$.

As stated in [15, Remark 4.3], we have

$$(D_h\boldsymbol{L})^t = \sum_{s=0}^{h} \binom{h}{s}(-1)^{h-s}\boldsymbol{L}^{t+s}. \tag{15}$$

To guarantee the coherence of the analysis's structure, we introduce the definition of the shift operator $E_s$ as follows:

**Definition A.2** (Shift Operator). For a sequence of vectors $\boldsymbol{L} = (\boldsymbol{L}^0, \cdots, \boldsymbol{L}^T)$ where each $\boldsymbol{L}^t \in \mathbb{R}^n$, and integers $s \in \mathbb{Z}_+$, the $s$-shift sequence for the sequence $\boldsymbol{L}$ is denoted by $E_s\boldsymbol{L} := ((E_s\boldsymbol{L})^0, \cdots, (E_s\boldsymbol{L})^{T-h})$ with $(E_s\boldsymbol{L})^t = \boldsymbol{L}^{t+s}$ for $t \in [1:T-s]$.

### A.3   Finite Horizon Markov Decision Process

We also consider the following finite horizon Markov decision process (MDP), denoted by $\mathcal{M} := (H, \mathcal{S}_{1:H}, \mathcal{A}_{1:H}, \mathbb{P}_{2:H}, \sigma, \boldsymbol{\rho}_1)$. $H \in \mathbb{Z}_+$ denotes the number of horizon; $\mathcal{S}_{1:H} = (\mathcal{S}_1, \cdots, \mathcal{S}_H)$ is a sequence of $H$ finite state spaces; $\mathcal{A}_{1:H} = (\mathcal{A}_1, \cdots, \mathcal{A}_H)$ is a sequence of $H$ finite action spaces; $\mathbb{P}_h(s_h|s_{h-1}, a_{h-1})$ denotes the probability of transitioning from $s_{h-1}$ to $s_h$ under playing action $a_{h-1}$ at horizon $h - 1$; $\sigma : \mathcal{S}_{1:H} \times \mathcal{A}_{1:H} \to [0, 1]$ is a cost function; $\boldsymbol{\rho}_1$ is a initial state distribution over $\mathcal{S}_1$.

$\boldsymbol{\pi} = (\boldsymbol{\pi}_1, \cdots, \boldsymbol{\pi}_H) : \mathcal{S}_{1:H} \to \Delta_{\mathcal{A}_1} \times \cdots \times \Delta_{\mathcal{A}_H}$ denotes a stochastic policy. Similarly, we use $\mathbf{Pr}_h^{\boldsymbol{\pi}_{1:h-1}}(s'|s) = \mathbf{Pr}_h^{\boldsymbol{\pi}_{1:h-1}}(s_h = s'|s_1 = s)$ to denote the probability of visiting the state $s'$ from the state $s$ at horizon $h$ according to policy $\boldsymbol{\pi}_{1:h-1}$. Let trajectory $\tau = (s_h, a_h)_{h=1}^H$, where $s_1 \sim \boldsymbol{\rho}_1$, and, for all subsequent horizon $h$, $a_h \sim \boldsymbol{\pi}_h(\cdot|s_h)$ and $s_{h+1} \sim \mathbb{P}_{h+1}(\cdot|s_h, a_h)$. The value function $V_h^{\boldsymbol{\pi}_{h:H}} : \mathcal{S}_h \to \mathbb{R}$ is defined as the sum of future cost starting at state $s_h$ and executing

**Algorithm 3** Optimistic Mirror Descent for Multi-Variables

---

**Input:** $\{\boldsymbol{g}_i^0 = \boldsymbol{x}_i^0\}_{i=1}^d$, $\eta$ and $T$.
**Output:** Randomly pick up $t \in \{1, \cdots, T\}$ following the probability $\mathbb{P}[t] = 1/T$ and return $\boldsymbol{x}^t$.

1: **while** $t \le T$ **do**
2:    **for all** $i \in [1:d]$ **do**
3:       $\boldsymbol{x}_i^t = \underset{\boldsymbol{x}_i \in \mathcal{X}_i}{\arg\min} \; \eta \left\langle \boldsymbol{F}_i(\boldsymbol{x}^{t-1}), \boldsymbol{x}_i \right\rangle + V\left(\boldsymbol{x}_i, \boldsymbol{g}_i^{t-1}\right),$
4:       $\boldsymbol{g}_i^t = \underset{\boldsymbol{g}_i \in \mathcal{X}_i}{\arg\min} \; \eta \left\langle \boldsymbol{F}_i(\boldsymbol{x}^t), \boldsymbol{g}_i \right\rangle + V\left(\boldsymbol{g}_i, \boldsymbol{g}_i^{t-1}\right).$
5:    **end for**
6:    $t \leftarrow t + 1$.
7: **end while**

---

$\boldsymbol{\pi}_{h:H} = (\boldsymbol{\pi}_h, \cdots, \boldsymbol{\pi}_H)$, i.e.,

$$V_h^{\boldsymbol{\pi}_{h:H}}(s_h) = \mathbb{E}\left[ \sum_{h'=h}^H \sigma(s_{h'}, a_{h'}) \middle| \boldsymbol{\pi}_{h:H}, s_h \right].$$

For convenience, we define $V_1^{\boldsymbol{\pi}}(s_1) = V_1^{\boldsymbol{\pi}_{1:H}}(s_1)$. Moreover, we define the action-value function $Q_h^{\boldsymbol{\pi}_{h+1:H}} : \mathcal{S}_h \times \mathcal{A}_h \to [0, 1 + H - h]$ as follows:

$$Q_h^{\boldsymbol{\pi}_{h+1:H}}(s_h, a_h) = \sigma(s_h, a_h) + \mathbb{E}\left[ \sum_{h'=h+1}^H \sigma(s_{h'}, a_{h'}) \middle| \boldsymbol{\pi}_{h+1:H}, s_h, a_h \right].$$

## B   Minimization Optimization

We begin with a general version of Theorem 3.2 basing Algorithm 3 in this part.

**Theorem B.1.** *[General Version of Theorem 3.2] We consider the divergence-generating function $v$ with Bregman's divergence $V(\boldsymbol{x}_i, \boldsymbol{u}_i) = v(\boldsymbol{x}_i) - v(\boldsymbol{u}_i) - \langle \nabla v(\boldsymbol{u}_i), \boldsymbol{x}_i - \boldsymbol{u}_i \rangle$ for any block $\mathcal{X}_i$ and any $\boldsymbol{x}_i, \boldsymbol{u}_i \in \mathcal{X}_i$. Assuming that $\boldsymbol{F}$ is $L$-Lipschitz continuous with respect to $\|\cdot\|_*$ under $\|\cdot\|$, $V(\boldsymbol{x}_i, \boldsymbol{u}_i) \ge \|\boldsymbol{x}_i - \boldsymbol{u}_i\|^2$ for any $\boldsymbol{x}_i, \boldsymbol{u}_i \in \mathcal{X}_i$ and $\gamma_{max} = \max_{i \in [1:d]} \gamma_i < \infty$, we have*

$$\frac{1}{T} \sum_{t=1}^T (f(\boldsymbol{x}^t) - f(\boldsymbol{x}^*)) \le \frac{2L \left(d\gamma_{max}\right)^{1/2} \left(\sum_{i=1}^d \gamma_i^{-1}\right)^{3/2}}{T} \max_{i \in [1:d]} \left[ \max_{\boldsymbol{x}_i \in \mathcal{X}_i} V(\boldsymbol{x}_i, \boldsymbol{g}_i^0) \right], \quad (16)$$

*with setting $\eta = (L^2 d\gamma_{max} \sum_{i=1}^d \gamma_i^{-1})^{-1/2}/2$.*

*Proof.* According to GQC condition (Definition 3.1), we have the following estimation

$$\sum_{t=1}^T (f(\boldsymbol{x}^t) - f(\boldsymbol{x}^*)) \le \sum_{i=1}^d \frac{1}{\gamma_i} \sum_{t=1}^T \left\langle \boldsymbol{F}_i(\boldsymbol{x}^t), \boldsymbol{x}_i^t - \boldsymbol{x}_i^* \right\rangle. \quad (17)$$

For any fixed $i \in [1:d]$, we obtain that

$$\left\langle \boldsymbol{F}_i(\boldsymbol{x}^t), \boldsymbol{x}_i^t - \boldsymbol{x}_i^* \right\rangle = \underbrace{\left\langle \boldsymbol{F}_i(\boldsymbol{x}^t) - \boldsymbol{F}_i(\boldsymbol{x}^{t-1}), \boldsymbol{x}_i^t - \boldsymbol{g}_i^t \right\rangle}_{\mathcal{I}} + \underbrace{\left\langle \boldsymbol{F}_i(\boldsymbol{x}^{t-1}), \boldsymbol{x}_i^t - \boldsymbol{g}_i^t \right\rangle}_{\mathcal{II}}$$
$$+ \underbrace{\left\langle \boldsymbol{F}_i(\boldsymbol{x}^t), \boldsymbol{g}_i^t - \boldsymbol{x}_i^* \right\rangle}_{\mathcal{III}} \quad (18)$$

Since $\boldsymbol{F}$ is $L$-Lipschitz continuous with respect to $\|\cdot\|_*$ under $\|\cdot\|$, we have following estimation of $\mathcal{I}$ by using Cauchy-Schwarz inequality

$$\mathcal{I} \le \frac{L^2\eta}{2} \left\| \boldsymbol{x}^t - \boldsymbol{x}^{t-1} \right\|^2 + \frac{1}{2\eta} \left\| \boldsymbol{x}_i^t - \boldsymbol{g}_i^t \right\|^2. \quad (19)$$

In addition, utilizing the result of [Lemma 3.4, [34]] on step-3 and step-4 of Algorithm 3, we have

$$\mathcal{II} \leq \frac{1}{\eta} \left[ V\left(\boldsymbol{g}_i^t, \boldsymbol{g}_i^{t-1}\right) - V\left(\boldsymbol{g}_i^t, \boldsymbol{x}_i^t\right) - V\left(\boldsymbol{x}_i^t, \boldsymbol{g}_i^{t-1}\right) \right], \tag{20}$$

$$\mathcal{III} \leq \frac{1}{\eta} \left[ V\left(\boldsymbol{x}_i^*, \boldsymbol{g}_i^{t-1}\right) - V\left(\boldsymbol{x}_i^*, \boldsymbol{g}_i^t\right) - V\left(\boldsymbol{g}_i^t, \boldsymbol{g}_i^{t-1}\right) \right]. \tag{21}$$

Therefore, by applying Eq. (19), (20) and (21) into Eq. (18), we obtain

$$\sum_{t=1}^T \langle \boldsymbol{F}_i(\boldsymbol{x}^t), \boldsymbol{x}_i^t - \boldsymbol{x}_i^* \rangle \leq \frac{1}{\eta} V(\boldsymbol{x}_i^*, \boldsymbol{g}_i^0) + \sum_{t=1}^T \left[ \frac{L^2\eta}{2} \left\| \boldsymbol{x}^t - \boldsymbol{x}^{t-1} \right\|^2 + \frac{1}{2\eta} \left\| \boldsymbol{x}_i^t - \boldsymbol{g}_i^t \right\|^2 \right]$$

$$- \frac{1}{\eta} \sum_{t=1}^T V(\boldsymbol{g}_i^t, \boldsymbol{x}_i^t) - \frac{1}{\eta} \sum_{t=1}^T V(\boldsymbol{x}_i^t, \boldsymbol{g}_i^{t-1})$$

$$\underset{(a)}{\leq} \frac{1}{\eta} V(\boldsymbol{x}_i^*, \boldsymbol{g}_i^0) + \frac{L^2\eta}{2} \sum_{t=1}^T \left\| \boldsymbol{x}^t - \boldsymbol{x}^{t-1} \right\|^2$$

$$- \frac{1}{2\eta} \sum_{t=1}^T \left\| \boldsymbol{g}_i^t - \boldsymbol{x}_i^t \right\|^2 - \frac{1}{2\eta} \sum_{t=1}^T \left\| \boldsymbol{x}_i^t - \boldsymbol{g}_i^{t-1} \right\|^2$$

$$\underset{(b)}{\leq} \frac{1}{\eta} V(\boldsymbol{x}_i^*, \boldsymbol{g}_i^0) + \frac{1}{2\eta} \left\| \boldsymbol{g}_i^0 - \boldsymbol{x}_i^0 \right\|^2 + \frac{L^2\eta}{2} \sum_{t=1}^T \left\| \boldsymbol{x}^t - \boldsymbol{x}^{t-1} \right\|^2$$

$$- \frac{1}{4\eta} \sum_{t=1}^T \left\| \boldsymbol{x}_i^t - \boldsymbol{x}_i^{t-1} \right\|^2, \tag{22}$$

where (a) is derived from the assumption that $V(\boldsymbol{x}_i, \boldsymbol{u}_i) \geq \|\boldsymbol{x}_i - \boldsymbol{u}_i\|^2$ for any $\boldsymbol{x}_i, \boldsymbol{u}_i \in \mathcal{X}_i$ and (b) follows from the convexity of $\|\cdot\|$. Applying Eq. (22) to Eq. (17), we have

$$\sum_{t=1}^T (f(\boldsymbol{x}^t) - f(\boldsymbol{x}^*)) \underset{(c)}{\leq} \frac{1}{\eta} \sum_{i=1}^d \frac{V(\boldsymbol{x}_i^*, \boldsymbol{g}_i^0)}{\gamma_i} + \frac{L^2\eta}{2} \left( \sum_{i=1}^d \gamma_i^{-1} \right) \sum_{t=1}^T \left\| \boldsymbol{x}^t - \boldsymbol{x}^{t-1} \right\|^2$$

$$- \frac{1}{4\eta} \sum_{t=1}^T \left[ \sum_{i=1}^d \frac{\left\| \boldsymbol{x}_i^t - \boldsymbol{x}_i^{t-1} \right\|^2}{\gamma_i} \right]$$

$$\underset{(d)}{\leq} \frac{\sum_{i=1}^d \gamma_i^{-1}}{\eta} \max_{i \in [1:d]} \left[ \max_{\boldsymbol{x}_i \in \mathcal{X}_i} V(\boldsymbol{x}_i, \boldsymbol{g}_i^0) \right]$$

$$- \left( \frac{1}{4d\eta\gamma_{\max}} - \frac{L^2\eta}{2} \sum_{i=1}^d \gamma_i^{-1} \right) \sum_{t=1}^T \left\| \boldsymbol{x}^t - \boldsymbol{x}^{t-1} \right\|^2, \tag{23}$$

where (c) is derived from the fact that $\boldsymbol{g}_i^0 = \boldsymbol{x}_i^0$ for any $i \in [1:d]$ and (d) follows from the convexity of $\|\cdot\|$ $(\frac{1}{d}\sum_{i=1}^d \|\boldsymbol{x}_i\|^2 \leq \|\frac{1}{d}\sum_{i=1}^d \boldsymbol{x}_i\|^2)$. $\qquad\square$

Since KL divergence satisfies $\text{KL}(\boldsymbol{x}_i \| \boldsymbol{u}_i) \geq \|\boldsymbol{x}_i - \boldsymbol{u}_i\|_1^2$ (Pinsker's inequality), Theorem 3.2 can be directly derived from Theorem B.1. Next, we propose Proposition B.2 and provide related proof.

**Proposition B.2.** *We denote $N = \sum_{i=1}^d n_i$ and let a smooth vector-valued function $\boldsymbol{F} : \mathbb{R}^N \to \mathbb{R}^\ell$ satisfies:*

    *1. There is a point $\boldsymbol{y} \in \mathbb{R}^N$ such that $\|D^{\boldsymbol{\alpha}} \boldsymbol{F}(\boldsymbol{y})\|_\infty \leq \gamma^k$ with $|\boldsymbol{\alpha}| = k$ for all $k \in [0:K]$,*

    *2. For any positive integer $k$ greater than $K$, $\|D^{\boldsymbol{\alpha}} \boldsymbol{F}\|_\infty \leq \gamma^k$ with $|\boldsymbol{\alpha}| = k$ uniformly over $\mathcal{X}$,*

*with a positive constant $\gamma$ and a positive integer $K$, then $\boldsymbol{F}$ satisfies Assumption 3.3.*

*Proof of Proposition B.2.* For any $k \in \mathbb{Z}_+$ and $j \in [1:l]$, we have

$$P_{k,\boldsymbol{y}}^{\boldsymbol{F}^{(j)}}(\boldsymbol{x}) \leq \sum_{i=0}^k \sum_{|\boldsymbol{\alpha}|=i} \frac{\gamma^i}{\boldsymbol{\alpha}!} \cdot (|\boldsymbol{x}| + |\boldsymbol{y}|)^{\boldsymbol{\alpha}} = \sum_{i=0}^k \frac{[\gamma(d + \|\boldsymbol{y}\|_1)]^i}{i!} \leq \exp\{\gamma(d + \|\boldsymbol{y}\|_1)\}, \tag{24}$$

using the fact that $\|D^{\alpha}\boldsymbol{F}(\boldsymbol{y})\|_{\infty} \leq \gamma^k$ for any $k \in \mathbb{Z}_+$ and $|\boldsymbol{\alpha}| = k$. In addition, by the Taylor expansion of $\boldsymbol{F}(j)$ with Lagrange remainder formula for any $j \in [1:l]$ and $k > 1$, we can obtain

$$\left|R_{k,\boldsymbol{y}}^{\boldsymbol{F}(j)}(\boldsymbol{x})\right| = \left|\sum_{|\boldsymbol{\alpha}|=k} \frac{D^{\alpha}\boldsymbol{F}(j)(\boldsymbol{y}+t(\boldsymbol{x}-\boldsymbol{y}))}{\boldsymbol{\alpha}!}(\boldsymbol{x}-\boldsymbol{y})^{\alpha}\right| \leq \frac{[\gamma(d+\|\boldsymbol{y}\|_1)]^k}{k!}, \qquad (25)$$

where $t \in [0,1]$ depends on $\boldsymbol{F}(j), \boldsymbol{x}$ and $\boldsymbol{y}$. Letting $k_0 = \lceil 3\gamma(d+\|\boldsymbol{y}\|_1)\rceil$ and supposing $k \geq k_0\left(1 + \frac{\log(1+\gamma(d+\|\boldsymbol{y}\|_1))}{\log(3/2)}\right)$, we derive that

$$\frac{[\gamma(d+\|\boldsymbol{y}\|_1)]^k}{k!} \leq 3^{k_0-k}. \qquad (26)$$

Therefore, in the light of Eq. (24), Eq. (25) and Eq. (26), it's direct to derive that $\boldsymbol{F}$ statisfies Assumption 3.3 with $K_0 = k_0\left(1 + \frac{\log(1+\gamma(d+\|\boldsymbol{y}\|_1))}{\log(3/2)}\right)$, $\theta = \frac{1}{3}$, $\Theta_1 = 3^{k_0}$ and $\Theta_2 = \exp\{\gamma(d+\|\boldsymbol{y}\|_1)\}$. $\hfill\square$

The following remark discusses the reasonability of Proposition B.2 conditions, which supports the reasonability of Assumption 3.3.

*Remark* B.3. Since region $\mathcal{X} = \prod_{i=1}^d \Delta_{n_i}$ is bounded, it's reasonable to assume that the growth rate of the upper bound of internal function's high-order derivatives is not faster than linear growth rate. For example, the upper bounds of high-order derivatives of $\sin(C\boldsymbol{x})$, $\cos(C\boldsymbol{x})$ and $\exp\{C\boldsymbol{x}\}$ have linear growth rate over $\mathcal{X}$ for fixed constant $C$. Therefore, if the internal function $\boldsymbol{F}$ can be generated by the linear combination of $\{\sin(C_k\boldsymbol{x})\}_{k=1}^K$ and $\{\cos(C_k\boldsymbol{x})\}_{k=1}^K$ (or $\{\exp\{C_k\boldsymbol{x}\}\}_{k=1}^K$) with finite $K$, $\boldsymbol{F}$ satisfies Assumption 3.3 by using Proposition B.2.

## B.1 Proof of Theorem 3.5

We briefly introduce our techniques to make the proof of Theorem 3.5 more comprehensible in this part. Our proof consists of two ingredients. The first is applying Lemma B.4 to construct a variant upper bound of average function error $\frac{1}{T}\sum_{t=1}^T (f(\boldsymbol{x}^t) - f(\boldsymbol{x}^*))$ that is different from the upper bound derived from the classical OMD algorithm. This bound is composed of a) $\mathcal{O}\left(\frac{1}{\eta T}\right)$ invariant error and b) weighted sum of the variance for finite difference sequence $\{(D_1\boldsymbol{F}_i(\boldsymbol{x}^{t-1})\}_{t=1}^T$ and $\{(D_0\boldsymbol{F}_i(\boldsymbol{x}^{t-1})\}_{t=1}^T$ over $i \in [1:d]$, which has the form of $\sum_{i=1}^d \frac{1}{\gamma_i}[\frac{\mathcal{O}(1)}{T}\sum_{t=1}^T \text{Var}_{\boldsymbol{x}_i^t}(D_1\boldsymbol{F}_i(\boldsymbol{x}^{t-1})) - \frac{\mathcal{O}(1)}{T}\sum_{t=1}^T \text{Var}_{\boldsymbol{x}_i^t}(\boldsymbol{F}_i(\boldsymbol{x}^{t-1}))]$. The second is applying Lemma D.7 (refer to it as control lemma) on each $\{\boldsymbol{F}_i(\boldsymbol{x}^t)\}_{t=1}^T$ to bound (b) by a quantity that grows poly-logarithmically in $T$. Therefore, it's necessary to leverage Theorem B.5 and Lemma B.7 to show that every sequence $\{\boldsymbol{F}_i(\boldsymbol{x}^t)\}_{t=0}^T$ outputted by Algorithm 1 satisfies the preconditions of Lemma D.7.

### B.1.1 Part I

The next Lemma B.4 provides a variant convergence proof of the OMD algorithm. In this Lemma, basing on KL divergence, an explicit expression for the optimal solution of the OMD sub-problem is utilized to provide an upper bound of $\sum_{t=1}^T (f(\boldsymbol{x}^t) - f(\boldsymbol{x}^*))$.

**Lemma B.4.** *Suppose* $\|\boldsymbol{F}(\boldsymbol{x})\|_{\infty} \leq \Theta$ *($\Theta \geq 1$) for any* $x \in \mathcal{X}$ *and policy set* $\{\boldsymbol{x}^t\}_{t=1}^T$ *follows the iteration of Algorithm 1 with step size* $\eta \in (0, \frac{1}{32\Theta})$. *Then, it holds that*

$$\sum_{t=1}^T \left(f(\boldsymbol{x}^t) - f(\boldsymbol{x}^*)\right) \leq \sum_{i=1}^d \frac{1}{\gamma_i}\left[\frac{\log(n_i)}{\eta} + \hat{g}_1(\eta\Theta)\eta\Theta^2\sum_{t=1}^T \text{Var}_{\boldsymbol{x}_i^t}\left(\boldsymbol{F}_i(\boldsymbol{x}^t) - \boldsymbol{F}_i(\boldsymbol{x}^{t-1})\right)\right.$$

$$\left. - \hat{g}_2(\eta\Theta)\eta\Theta^2\sum_{t=1}^T \text{Var}_{\boldsymbol{x}_i^t}(\boldsymbol{F}_i(\boldsymbol{x}^{t-1}))\right], \qquad (27)$$

*where* $\hat{g}_1(\eta) := \frac{1}{2} + 64\left(\frac{1}{3(1-16\eta)} + 2\right)\eta$ *and* $\hat{g}_2(\eta) := \frac{1}{2} - 16\left(\frac{1}{3(1-16\eta)} + 2\right)\eta$.

*Proof.* As claimed by Definition 3.1, we have the following estimation

$$\sum_{t=1}^{T}(f(\boldsymbol{x}^t) - f(\boldsymbol{x}^*)) \leq \sum_{i=1}^{d}\frac{1}{\gamma_i}\sum_{t=1}^{T}\langle\boldsymbol{F}_i(\boldsymbol{x}^t), \boldsymbol{x}_i^t - \boldsymbol{x}_i^*\rangle. \tag{28}$$

In the following, considering a fixed $i \in [1:d]$, it's easy to obtain that

$$\langle\boldsymbol{F}_i(\boldsymbol{x}^t), \boldsymbol{x}_i^t - \boldsymbol{x}_i^*\rangle = \underbrace{\langle\boldsymbol{F}_i(\boldsymbol{x}^t) - \boldsymbol{F}_i(\boldsymbol{x}^{t-1}), \boldsymbol{x}_i^t - \boldsymbol{g}_i^t\rangle}_{\mathcal{I}} + \underbrace{\langle\boldsymbol{F}_i(\boldsymbol{x}^{t-1}), \boldsymbol{x}_i^t - \boldsymbol{g}_i^t\rangle}_{\mathcal{II}}$$

$$+ \underbrace{\langle\boldsymbol{F}_i(\boldsymbol{x}^t), \boldsymbol{g}_i^t - \boldsymbol{x}_i^*\rangle}_{\mathcal{III}} \tag{29}$$

Recall the update of Algorithm 1 can be devided into two parts:

$$\boldsymbol{g}_i^t = \arg\min_{\boldsymbol{g}_i\in\Delta_{n_i}} \eta\langle\boldsymbol{F}_i(\boldsymbol{x}^t), \boldsymbol{g}_i\rangle + \text{KL}(\boldsymbol{g}_i\|\boldsymbol{g}_i^{t-1}), \tag{30}$$

$$\boldsymbol{x}_i^{t+1} = \arg\min_{\boldsymbol{x}_i\in\Delta_{n_i}} \eta\langle\boldsymbol{F}_i(\boldsymbol{x}^t), \boldsymbol{x}_i\rangle + \text{KL}(\boldsymbol{x}_i\|\boldsymbol{g}_i^t), \tag{31}$$

for any $i \in [1:d]$ where $\boldsymbol{g}_i^0 \propto \boldsymbol{x}_i^0 \cdot \exp\{\eta(\boldsymbol{F}_i(\boldsymbol{x}^0) - \boldsymbol{F}_i(\boldsymbol{x}^{-1}))\}$ and $\boldsymbol{x}_i^{-1} = \boldsymbol{x}_i^0 = \left(\frac{1}{n_i}, \cdots, \frac{1}{n_i}\right)^{\top}$.
According to Cauchy-Schwarz inequality, we can evaluate $\mathcal{I}$ as follows

$$\mathcal{I} \leq \left\|\boldsymbol{g}_i^t - \boldsymbol{x}_i^t\right\|_{\boldsymbol{x}_i^t}^* \cdot \sqrt{\text{Var}_{\boldsymbol{x}_i^t}(\boldsymbol{F}_i(\boldsymbol{x}^t) - \boldsymbol{F}_i(\boldsymbol{x}^{t-1}))}. \tag{32}$$

In addition, utilizing the result of Lemma D.2, we have

$$\mathcal{II} = \frac{1}{\eta}\left[\text{KL}\left(\boldsymbol{g}_i^t\|\boldsymbol{g}_i^{t-1}\right) - \text{KL}\left(\boldsymbol{g}_i^t\|\boldsymbol{x}_i^t\right) - \text{KL}\left(\boldsymbol{x}_i^t\|\boldsymbol{g}_i^{t-1}\right)\right], \tag{33}$$

$$\mathcal{III} = \frac{1}{\eta}\left[\text{KL}\left(\boldsymbol{x}_i^*\|\boldsymbol{g}_i^{t-1}\right) - \text{KL}\left(\boldsymbol{x}_i^*\|\boldsymbol{g}_i^t\right) - \text{KL}\left(\boldsymbol{g}_i^t\|\boldsymbol{g}_i^{t-1}\right)\right]. \tag{34}$$

Therefore, by applying Eq. (32), (33) and (34) into Eq. (29), we obtain

$$\sum_{t=1}^{T}\langle\boldsymbol{F}_i(\boldsymbol{x}^t), \boldsymbol{x}_i^t - \boldsymbol{x}_i^*\rangle \leq \frac{1}{\eta}\text{KL}(\boldsymbol{x}_i^*\|\boldsymbol{g}_i^0) + \sum_{t=1}^{T}\|\boldsymbol{g}_i^t - \boldsymbol{x}_i^t\|_{\boldsymbol{x}_i^t}^* \cdot \sqrt{\text{Var}_{\boldsymbol{x}_i^t}(\boldsymbol{F}_i(\boldsymbol{x}^t) - \boldsymbol{F}_i(\boldsymbol{x}^{t-1}))}$$

$$- \frac{1}{\eta}\sum_{t=1}^{T}\text{KL}(\boldsymbol{g}_i^t\|\boldsymbol{x}_i^t) - \frac{1}{\eta}\sum_{t=1}^{T}\text{KL}(\boldsymbol{x}_i^t\|\boldsymbol{g}_i^{t-1}). \tag{35}$$

Since there is a vector $\boldsymbol{F}_i(\boldsymbol{x}^t) - \boldsymbol{F}_i(\boldsymbol{x}^{t-1})$ such that for any $j \in [1:n_i]$

$$\boldsymbol{g}_i^t(j) = \frac{\boldsymbol{x}_i^t(j)\exp\left\{\eta\left(\boldsymbol{F}_i(j)(\boldsymbol{x}^t) - \boldsymbol{F}_i(j)(\boldsymbol{x}^{t-1})\right)\right\}}{\sum_{j'=1}^{n_i}\boldsymbol{x}_i^t(j')\exp\left\{\eta\left(\boldsymbol{F}_i(j')(\boldsymbol{x}^t) - \boldsymbol{F}_i(j')(\boldsymbol{x}^{t-1})\right)\right\}}, \tag{36}$$

we have that

$$\max_{i\in[1:d]}\left\|\frac{\boldsymbol{g}_i^t}{\boldsymbol{x}_i^t}\right\|_{\infty} \leq \exp\{2\eta\left\|\boldsymbol{F}_i(\boldsymbol{x}^t) - \boldsymbol{F}_i(\boldsymbol{x}^{t-1})\right\|_{\infty}\} \leq \exp\{4\eta\Theta\} \leq 1 + 8\eta\Theta, \tag{37}$$

and

$$\max_{i\in[1:d]}\left\|\frac{\boldsymbol{x}_i^t}{\boldsymbol{g}_i^{t-1}}\right\|_{\infty} \leq \exp\{2\eta\|\boldsymbol{F}_i(\boldsymbol{x}^{t-1})\|_{\infty}\} \leq \exp\{2\eta\Theta\} \leq 1 + 4\eta\Theta,$$

with combining Eq. (31) and choosing proper $\eta$ such that $\eta\Theta \leq \frac{1}{4}$. According to Lemma D.3, we have

$$\text{KL}(\boldsymbol{g}_i^t\|\boldsymbol{x}_i^t) \geq \left(\frac{1 - 8\eta\Theta}{2} - \frac{16\eta\Theta}{3(1 - 8\eta\Theta)}\right)\mathcal{X}_i^2(\boldsymbol{g}_i^t, \boldsymbol{x}_i^t),$$

$$\text{KL}(\boldsymbol{x}_i^t\|\boldsymbol{g}_i^{t-1}) \geq \left(\frac{1 - 4\eta\Theta}{2} - \frac{8\eta\Theta}{3(1 - 4\eta\Theta)}\right)\mathcal{X}^2(\boldsymbol{x}_i^t, \boldsymbol{g}_i^{t-1}), \tag{38}$$

for any $i \in [1:d]$. Noting that $\mathcal{X}^2(\rho, \mu) = \left( \|\rho - \mu\|_\mu^* \right)^2$, in the light of Lemma D.4, we derive that

$$\mathcal{X}^2(\boldsymbol{g}_i^t, \boldsymbol{x}_i^t) \leq \left( 1 + 32 \left( \frac{1}{3(1 - 16\eta\Theta)} + 2 \right) \eta\Theta \right) (\eta\Theta)^2 \mathrm{Var}_{\boldsymbol{x}_i^t} \left( \boldsymbol{F}_i(\boldsymbol{x}^t) - \boldsymbol{F}_i(\boldsymbol{x}^{t-1}) \right),$$
$$\mathcal{X}^2(\boldsymbol{g}_i^t, \boldsymbol{x}_i^t) \geq \left( 1 - 32 \left( \frac{1}{3(1 - 16\eta\Theta)} + 2 \right) \eta\Theta \right) (\eta\Theta)^2 \mathrm{Var}_{\boldsymbol{x}_i^t} \left( \boldsymbol{F}_i(\boldsymbol{x}^t) - \boldsymbol{F}_i(\boldsymbol{x}^{t-1}) \right),$$
$$(39)$$

as long as $\eta\Theta \leq \frac{1}{32}$. There exists a similar lower bound with respect to $\mathcal{X}^2(\boldsymbol{x}_i^t, \boldsymbol{g}_i^{t-1})$

$$\begin{aligned}
\mathcal{X}^2(\boldsymbol{x}_i^t, \boldsymbol{g}_i^{t-1}) &\geq \left( 1 - 16 \left( \frac{1}{3(1 - 8\eta\Theta)} + 2 \right) \eta\Theta \right) (\eta\Theta)^2 \mathrm{Var}_{\boldsymbol{g}_i^{t-1}}(\boldsymbol{F}_i(\boldsymbol{x}^{t-1})) \\
&\geq \left( 1 - 16 \left( \frac{1}{3(1 - 8\eta\Theta)} + 2 \right) \eta\Theta \right) (\eta\Theta)^2 \exp\{-2\eta\Theta\} \mathrm{Var}_{\boldsymbol{x}_i^t}(\boldsymbol{F}_i(\boldsymbol{x}^{t-1})) \\
&\underset{(a)}{\geq} \left( 1 - 16 \left( \frac{1}{3(1 - 8\eta\Theta)} + 3 \right) \eta\Theta \right) (\eta\Theta)^2 \mathrm{Var}_{\boldsymbol{x}_i^t}(\boldsymbol{F}_i(\boldsymbol{x}^{t-1})),
\end{aligned}$$
$$(40)$$

where (a) is derived from $\exp\{-2\eta\Theta\} \geq 1 - 4\eta\Theta$ for any $\eta\Theta \leq \frac{1}{32}$. Relying on Eq. (35), Eq. (38)-(40), we conclude that

$$\begin{aligned}
&\sum_{t=1}^{T} \left\langle \boldsymbol{F}_i(\boldsymbol{x}^t), \boldsymbol{x}_i^t - \boldsymbol{x}_i^* \right\rangle \\
&\leq \frac{\log(n_i)}{\eta} + \left( 1 + 32 \left( \frac{1}{3(1 - 16\eta\Theta)} + 2 \right) \eta\Theta \right) \eta\Theta^2 \sum_{t=1}^{T} \mathrm{Var}_{\boldsymbol{x}_i^t} \left( \boldsymbol{F}_i(\boldsymbol{x}^t) - \boldsymbol{F}_i(\boldsymbol{x}^{t-1}) \right) \\
&\quad - \left( \frac{1}{2} - \left( \frac{32}{3(1 - 16\eta\Theta)} + 36 \right) \eta\Theta \right) \eta\Theta^2 \sum_{t=1}^{T} \mathrm{Var}_{\boldsymbol{x}_i^t} \left( \boldsymbol{F}_i(\boldsymbol{x}^t) - \boldsymbol{F}_i(\boldsymbol{x}^{t-1}) \right) \\
&\quad - \left( \frac{1}{2} - \left( \frac{16}{3(1 - 8\eta\Theta)} + 27 \right) \eta\Theta \right) \eta\Theta^2 \sum_{t=1}^{T} \mathrm{Var}_{\boldsymbol{x}_i^t} (\boldsymbol{F}_i(\boldsymbol{x}^{t-1})) \\
&\leq \frac{\log(n_i)}{\eta} + \left( \frac{1}{2} + 64 \left( \frac{1}{3(1 - 16\eta\Theta)} + 2 \right) \eta\Theta \right) \eta\Theta^2 \sum_{t=1}^{T} \mathrm{Var}_{\boldsymbol{x}_i^t} \left( \boldsymbol{F}_i(\boldsymbol{x}^t) - \boldsymbol{F}_i(\boldsymbol{x}^{t-1}) \right) \\
&\quad - \left( \frac{1}{2} - 16 \left( \frac{1}{3(1 - 16\eta\Theta)} + 2 \right) \eta\Theta \right) \eta\Theta^2 \sum_{t=1}^{T} \mathrm{Var}_{\boldsymbol{x}_i^t} (\boldsymbol{F}_i(\boldsymbol{x}^{t-1})).
\end{aligned}$$
$$(41)$$

Finally, applying the estimation Eq. (41) to Eq. (28), we complete the proof. $\qquad \square$

### B.1.2  Part II

Basing on the conclusion of Lemma B.4, if the finite sum of $\mathrm{Var}_{\boldsymbol{x}_i^t}(\boldsymbol{F}_i(\boldsymbol{x}^t) - \boldsymbol{F}_i(\boldsymbol{x}^{t-1}))$ can be controlled by the finite sum of $\mathrm{Var}_{\boldsymbol{x}_i^t}(\boldsymbol{F}_i(\boldsymbol{x}^{t-1}))$ with a $\mathcal{O}(\mathrm{poly}(\log(T)))$ constant for each $i \in [1:d]$, the final convergence result can be obtained directly. Hence, to demonstrate this relationship, we require the assistance of auxiliary Lemma D.7. Our initial step is to prove that $\boldsymbol{F}_i(\boldsymbol{x}^t)$ satisfies the first condition in Lemma D.7 for any $i \in [1:d]$.

**Theorem B.5.** *Assuming $f$ satisfies GQC condition and Assumption 3.3 holds, $\boldsymbol{x}^t$ follows the iteration of Algorithm 1, we set $\beta \in \left( 0, \frac{1}{(\Theta_1 + \Theta_2 + 1)(H+3)} \right)$, $\Gamma \geq e^2 + 322560\Theta_2$, $\hat{K} \geq \max\{K_0,$ $\frac{H \log(4\beta^{-1}) + \log(\Theta_1)}{\log(\theta^{-1})}\}$ and $\eta = \frac{\beta}{6e^3 \hat{K} \Gamma \max\{\Theta, 1\}}$. Then, the following finite difference bound with respect to $\{\boldsymbol{F}_i(\boldsymbol{x}^t)\}_{i=1}^{d}$ holds*

$$\max_{i \in [1:d]} \left\| (D_h \boldsymbol{F}_i(\boldsymbol{x}))^{t_0} \right\|_\infty \leq \beta^h h^{3h+1}, \tag{42}$$

*for all $h \in [1:H]$ and $t_0 \in [0 : T - h]$. Without loss of generality, we require that $H$ does not exceed $T$.*

*Proof of Theorem B.5.* According to the Taylor expansion of each component $k$ of $\boldsymbol{F}_i$ at $\boldsymbol{y}$, one can notice that

$$\left|(D_h\boldsymbol{F}_i(k)(\boldsymbol{x}))^{t_0}\right| \leq \sum_{j=0}^{\hat{K}}\sum_{|\boldsymbol{\alpha}|=j}\frac{|D^{\boldsymbol{\alpha}}\boldsymbol{F}_i(k)(\boldsymbol{y})|}{\boldsymbol{\alpha}!}\left|(D_h(\boldsymbol{x}-\boldsymbol{y})^{\boldsymbol{\alpha}})^{t_0}\right| + \left|\left(D_h R^{\boldsymbol{F}_i(k)}_{\hat{K},\boldsymbol{y}}(\boldsymbol{x})\right)^{t_0}\right|, \quad (43)$$

for any $\hat{K} \in \mathbb{Z}_+$. Therefore, setting $\hat{K} \geq \max\left\{\frac{H\log(4\beta^{-1})+\log(\Theta_1)}{\log(\theta^{-1})}, K_0\right\}$ and combining the remark Eq. (15) of operator $D_h$ in Appendix A.2, we can guarantee the validity of the following estimation

$$\left|\left(D_h R^{\boldsymbol{F}_i(k)}_{\hat{K},\boldsymbol{y}}(\boldsymbol{x})\right)^{t_0}\right| \leq 2^h\max_{\boldsymbol{x}\in\mathcal{X}}\left|R^{\boldsymbol{F}_i(k)}_{\hat{K},\boldsymbol{y}}(\boldsymbol{x})\right| \leq \Theta_1 2^h\theta^{\hat{K}} \leq \frac{1}{2}\beta^H \leq \frac{1}{2}\beta^H h^{Bh+1}, \quad (44)$$

for any $h \in [1:H]$. Moreover, as stated by Assumption 3.3, we obtain $\max_{i\in[1:d]}\|\boldsymbol{F}_i(\boldsymbol{x})\|_\infty \leq \Theta_1 + \Theta_2$ for any $\boldsymbol{x} \in \mathcal{X}$. Suppose that $\max_{i\in[1:d]}\|(D_{h'}\boldsymbol{F}_i(\boldsymbol{x}))^{t_0}\|_\infty \leq \beta^{h'}h'^{Bh'+1}$ holds for any $h' \in [1:h]$ and $t_0 \in [0:T-h']$, we deduce

$$\begin{aligned}\left|(D_{h+1}\boldsymbol{F}_i(k)(\boldsymbol{x}))^{t_0}\right| &\leq g(\Gamma)\beta^{h+1}(h+1)^{B(h+1)+1}P^{\boldsymbol{F}_i(k)}_{\hat{K},\boldsymbol{y}}(\boldsymbol{x}^{t_0}) + \frac{1}{2}\beta^{h+1}(h+1)^{B(h+1)+1}\\ &\leq \left(\frac{1}{2}+g(\Gamma)\Theta_2\right)\beta^{h+1}(h+1)^{B(h+1)+1} \leq \beta^{h+1}(h+1)^{B(h+1)+1},\end{aligned} \quad (45)$$

by using Lemma B.6 with $p(\boldsymbol{x}) := \frac{|D^{\boldsymbol{\alpha}}\boldsymbol{F}_i(k)(\boldsymbol{y})|}{\boldsymbol{\alpha}!}(\boldsymbol{x}-\boldsymbol{y})^{\boldsymbol{\alpha}}$ and the fact that $g(\Gamma)\Theta_2 \leq \frac{1}{2}$ (which can be derived from Lemma D.1). Therefore, to apply mathematical induction, it suffices to prove that $\max_{i\in[1:d]}\|(D_{h'}\boldsymbol{F}_i(\boldsymbol{x}))^{t_0}\|_\infty \leq \beta^{h'}h'^{Bh'+1}$ holds when $h' = 1$. Observe that Lemma B.6 holds in the case $h = 0$. Thus, we can obtain Eq. (45) for $h = 0$ as well. Hence, we have $\max_{i\in[1:d]}\|(D_1\boldsymbol{F}_i(\boldsymbol{x}))^{t_0}\|_\infty \leq \beta$. $\square$

The proof of Theorem B.5 relies on the next Lemma B.6.

**Lemma B.6.** *Assume* $\max_{i\in[1:d]}\|\boldsymbol{F}_i(\boldsymbol{x})\|_\infty \leq \Theta$ *for any* $\boldsymbol{x} \in \mathcal{X}$ *and each element in* $\boldsymbol{u}_t$ *belongs to one of the d probability distributions generated by Algorithm 1 with* $\eta \leq \frac{\beta}{6e^3\Gamma\hat{K}\max\{\Theta,1\}}$ *for some* $\Gamma > 1, \hat{K} \geq K$ *in iteration t, and consider positive constants* $B \geq 3$, $\beta \in \left(0, \frac{1}{(\Theta+1)(H+3)}\right)$ *and polynomial function* $p(\boldsymbol{u}) := C\prod_{k=1}^K(\boldsymbol{u}(k)-\boldsymbol{y}(k))$ *where* $\boldsymbol{u} := (\boldsymbol{u}(1),\cdots,\boldsymbol{u}(K))^\top$ *and* $\boldsymbol{y} := (\boldsymbol{y}(1),\cdots,\boldsymbol{y}(K))^\top \in \mathbb{R}^K$ *is a fixed point. Given* $h \in [1:H-1]$, *we derive that*

$$\left|(D_{h+1}p(\boldsymbol{u}))^{t_0}\right| \leq g(\Gamma)C\prod_{k=1}^K(\boldsymbol{u}^{t_0}(k)+|\boldsymbol{y}(k)|)\beta^{h+1}(h+1)^{B(h+1)+1}, \quad (46)$$

*if the condition* $\max_{i\in[1:d]}\|(D_{h'}\boldsymbol{F}_i(\boldsymbol{x}))^{t_0}\|_\infty \leq \beta^{h'}h'^{Bh'+1}$ *holds for any* $h' \in [1:h]$ *and* $t_0 \in [0:T-h']$.

*Proof.* Drawing on the premises outlined in the lemma, we assume that each $\boldsymbol{u}(k)$ corresponds to a unique $\boldsymbol{x}^{i(k)}(j(k))$. According to the iteration of Algorithm 1, we can obtain

$$\begin{aligned}\boldsymbol{u}^{t+1}(k) &= \frac{\boldsymbol{x}^t_{i(k)}(j(k))\cdot\exp\left\{\eta\cdot\left(2\boldsymbol{F}_{i(k)}(j(k))(\boldsymbol{x}^t)-\boldsymbol{F}_{i(k)}(j(k))(\boldsymbol{x}^{t-1})\right)\right\}}{\sum_{j=1}^{n_{i(k)}}\boldsymbol{x}^t_{i(k)}(j)\cdot\exp\left\{\eta\cdot\left(2\boldsymbol{F}_{i(k)}(j)(\boldsymbol{x}^t)-\boldsymbol{F}_{i(k)}(j)(\boldsymbol{x}^{t-1})\right)\right\}},\\ &= \frac{\boldsymbol{u}^t(k)\cdot\exp\left\{\eta\cdot\left(2\boldsymbol{F}_{i(k)}(j(k))(\boldsymbol{x}^t)-\boldsymbol{F}_{i(k)}(j(k))(\boldsymbol{x}^{t-1})\right)\right\}}{\sum_{j=1}^{n_{i(k)}}\boldsymbol{x}^t_{i(k)}(j)\cdot\exp\left\{\eta\cdot\left(2\boldsymbol{F}_{i(k)}(j)(\boldsymbol{x}^t)-\boldsymbol{F}_{i(k)}(j)(\boldsymbol{x}^{t-1})\right)\right\}},\end{aligned} \quad (47)$$

for any $k \in [1:K]$ and $t \in [1:T-1]$. Given the sequence $\boldsymbol{x}^1,\cdots,\boldsymbol{x}^{t_0+h}$ generated by Algorithm 1, it is straightforward to derive that

$$\begin{aligned}\boldsymbol{u}^{t_0+t+1}(k) = (N^k_{\boldsymbol{u}})^{-1}\boldsymbol{u}^{t_0}(k)\cdot\exp\{\eta\cdot(\boldsymbol{F}_{i(k)}(j(k))(\boldsymbol{x}^{t_0+t})+\sum_{t'=0}^t\boldsymbol{F}_{i(k)}(j(k))(\boldsymbol{x}^{t_0+t'})\\ -\boldsymbol{F}_{i(k)}(j(k))(\boldsymbol{x}^{t_0-1}))\},\end{aligned} \quad (48)$$

for any $k \in [1:K]$, $t_0 \in [1:T-h-1]$ and $t \in [1:h]$, where $N_{\boldsymbol{u}}^k = \sum_{j=1}^{n_{i(k)}} \boldsymbol{x}_{i(k)}^{t_0}(j) \cdot \exp\{\eta \cdot$
$(\boldsymbol{F}_{i(k)}(j)(\boldsymbol{x}^{t_0+t}) + \sum_{t'=0}^{t} \boldsymbol{F}_{i(k)}(j)(\boldsymbol{x}^{t_0+t'}) - \boldsymbol{F}_{i(k)}(j)(\boldsymbol{x}^{t_0-1}))\}$. We write

$$\boldsymbol{r}_{t_0,k}^t := \boldsymbol{F}_{i(k)}(\boldsymbol{x}^{t_0+t-1}) + \sum_{t'=0}^{t-1} \boldsymbol{F}_{i(k)}(\boldsymbol{x}^{t_0+t'}) - \boldsymbol{F}_{i(k)}(\boldsymbol{x}^{t_0-1}). \tag{49}$$

Also, for a vector $\boldsymbol{z} \in \mathbb{R}^{n_{i(k)}}$ and an index $j \in [1:n_{i(k)}]$, define

$$\boldsymbol{\psi}_{t_0,k}^j(\boldsymbol{z}) = \frac{\exp\{\boldsymbol{z}(j)\}}{\sum_{j'=1}^{n_{i(k)}} \boldsymbol{x}_{t_0}^{i(k)}(j') \cdot \exp\{\boldsymbol{z}(j')\}}, \tag{50}$$

so that $\boldsymbol{u}^{t_0+t}(k) = \boldsymbol{x}_{i(k)}^{t_0}(j(k)) \cdot \boldsymbol{\psi}_{t_0,k}^{j(k)}\left(\eta \boldsymbol{r}_{t_0,k}^t\right) = \boldsymbol{u}^{t_0}(k) \cdot \boldsymbol{\psi}_{t_0,k}^{j(k)}\left(\eta \boldsymbol{r}_{t_0,k}^t\right)$ for $t \geq 1$. For convenience, we denote that $\mathcal{D} := \{\boldsymbol{\alpha} \in \mathbb{N}^K | \boldsymbol{\alpha}(i) \in \{0,1\}, \forall i \in [1:K]\}$ and $\boldsymbol{e} := (1,\cdots,1) \in \mathbb{N}^K$. In particular, for any $\boldsymbol{\alpha} \in \mathcal{D}$, we have

$$\left|\left(D_{h'}(\boldsymbol{u}^{\boldsymbol{e}-\boldsymbol{\alpha}})\right)^{t_0}\right| \leq (\boldsymbol{u}^{t_0})^{\boldsymbol{e}-\boldsymbol{\alpha}} \underbrace{\left|\left(D_{h'}(\boldsymbol{\psi}_{t_0}(\eta \boldsymbol{r}_{t_0}))^{\boldsymbol{e}-\boldsymbol{\alpha}}\right)^0\right|}_{\mathcal{I}(\boldsymbol{\alpha},h',t_0)}, \tag{51}$$

where $\boldsymbol{\psi}_{t_0}(\eta \boldsymbol{r}_{t_0}^t) := \left(\boldsymbol{\psi}_{t_0,1}^{j(1)}(\eta \boldsymbol{r}_{t_0,1}^t), \cdots, \boldsymbol{\psi}_{t_0,K}^{j(K)}(\eta \boldsymbol{r}_{t_0,K}^t)\right)$, $h' \in [1:h+1]$ and $t_0 \in [1:T-h-1]$. It is important to observe that the finite difference in Eq. (51) pertains specifically to $\boldsymbol{\psi}_{t_0,k}^{j(k)}(\eta \boldsymbol{r}_{t_0,k}^t)$. Notice that

$$(D_1 \boldsymbol{r}_{t_0,k})^t = 2(E_1 \boldsymbol{F}_{i(k)}(\boldsymbol{x}))^{t_0+t-1} - \boldsymbol{F}_{i(k)}(\boldsymbol{x}^{t_0+t-1}), \tag{52}$$

for any $t \in [0:h]$. Therefore, for any $h' \in [1:h+1]$, we obtain

$$(D_{h'} \boldsymbol{r}_{t_0,k})^t = 2\left(E_1 D_{h'-1}\left(\boldsymbol{F}_{i(k)}(\boldsymbol{x})\right)\right)^{t_0+t-1} - \left(D_{h'-1}\left(\boldsymbol{F}_{i(k)}(\boldsymbol{x})\right)\right)^{t_0+t-1}, \tag{53}$$

for any $t \in [0:h+1-h']$. Because the step size $\eta$ satisfies $\eta \leq \frac{\beta}{6e^3 \Gamma \hat{K} \max\{\Theta,1\}}$, $\left\|(D_0 \eta \boldsymbol{r}_{t_0})^t\right\|_\infty \leq \eta H\Theta \leq \frac{1}{6e^3 \Gamma \hat{K}}$ and $\max_{i\in[1:d]} \left\|(D_{h'} \boldsymbol{F}_i(\boldsymbol{x}))^{t_0}\right\|_\infty \leq \beta^{h'} h'^{Bh'+1}$ for all $h' \in [1:h]$, the following estimation holds

$$\left\|(D_{h'+1} \eta \boldsymbol{r}_{t_0})^0\right\|_\infty \leq \frac{1}{2e^2 \Gamma \hat{K}} \beta^{h'+1}(h'+1)^{B(h'+1)}, \tag{54}$$

for any $h' \in [0:h]$ by using Eq. (53) where $\boldsymbol{r}_{t_0}^t := \left(\boldsymbol{r}_{t_0,1}^t(j(1)), \cdots, \boldsymbol{r}_{t_0,K}^t(j(K))\right)$. By Lemma D.5 and Lemma D.6, we have

$$\mathcal{I}(\boldsymbol{\alpha}, h+1, t_0) \leq g(\Gamma)\beta^{h+1}(h+1)^{B(h+1)+1}. \tag{55}$$

Noting that

$$p(\boldsymbol{u}) = C \sum_{i=0}^{K} (-1)^i \left(\sum_{\boldsymbol{\alpha}\in\mathcal{D}:|\boldsymbol{\alpha}|=i} \boldsymbol{y}^{\boldsymbol{\alpha}} \boldsymbol{u}^{\boldsymbol{e}-\boldsymbol{\alpha}}\right), \tag{56}$$

and applying bound Eq. (55) to Eq. (50), we can derive

$$\left|(D_{h+1}p(\boldsymbol{u}))^{t_0}\right| \underset{(a)}{=} |C| \left|\sum_{i=0}^{K}(-1)^i \left(\sum_{\boldsymbol{\alpha}\in\mathcal{D}:|\boldsymbol{\alpha}|=i} \boldsymbol{y}^{\boldsymbol{\alpha}} \sum_{h'=1}^{h+1} \binom{h+1}{h'}(-1)^{h'}(\boldsymbol{u}^{t_0+h'})^{\boldsymbol{e}-\boldsymbol{\alpha}}\right)\right|$$

$$\leq |C| \sum_{i=0}^{K} \sum_{\boldsymbol{\alpha}\in\mathcal{D}:|\boldsymbol{\alpha}|=i} |\boldsymbol{y}|^{\boldsymbol{\alpha}} \left|\left(D_{h+1}\left(\boldsymbol{u}^{\boldsymbol{e}-\boldsymbol{\alpha}}\right)\right)^{t_0}\right|$$

$$\leq |C| \sum_{i=0}^{K} \sum_{\boldsymbol{\alpha}\in\mathcal{D}:|\boldsymbol{\alpha}|=i} |\boldsymbol{y}|^{\boldsymbol{\alpha}}(\boldsymbol{u}^{t_0})^{\boldsymbol{e}-\boldsymbol{\alpha}} g(\Gamma)\beta^{h+1}(h+1)^{B(h+1)+1}$$

$$\leq g(\Gamma)\beta^{h+1}(h+1)^{B(h+1)+1} C \prod_{k=1}^{K} (\boldsymbol{u}^{t_0}(k) + |\boldsymbol{y}(k)|), \tag{57}$$

for any $t_0 \in [1:T-h-1]$ where (a) is derived from the equivalent expression Eq. (56) of the polynomial $p(\boldsymbol{u})$ and Eq. (15) of the finite difference $(D_h f(\boldsymbol{x}))^{t_0}$ w.r.t function $f$ respectively. $\quad\square$

Recalling that we set parameters as follows

$$T \geq 4, H := \lceil \log(T) \rceil, \beta = \frac{1}{8(\Theta_1 + \Theta_2 + 1)H^{7/2}}, \Gamma = e^2 + 322560\Theta_2,$$

$$\hat{K} = \max\left\{ \frac{H \log(4\beta^{-1}) + \log(\Theta_1)}{\log(\theta^{-1})}, K_0 \right\}, \eta = \frac{\beta}{6e^2\hat{K}\Gamma}, B \geq 3, \tag{58}$$

According to Theorem B.5, we have $\max_{i \in [1:d]} \left\| (D_h \boldsymbol{F}_i(\boldsymbol{x}))^t \right\|_\infty \leq \beta^h H^{3h+1}$ for each $h \in [0:H]$ and $t \in [1:T-h]$. We are now prepared to prove that $\boldsymbol{x}_i^t$ satisfies the second condition of Lemma D.7.

**Lemma B.7.** *The sequence $\{\boldsymbol{x}_i^t\}_{t=1}^T$ which has been generated from Algorithm 1 is $7\eta(\Theta_1 + \Theta_2)-$ consecutively close when $H \geq 1$, $\beta_0 = (4H)^{-1}$ and $\eta \in (0, \beta_0^4(\Theta_1 + \Theta_2 + 1)^{-1}/57792]$.*

*Proof.* According to the iteration of Algorithm 1, we have

$$\boldsymbol{x}_i^{t+1}(k) = \frac{\boldsymbol{x}_i^t(k) \cdot \exp\{\eta \cdot (2\boldsymbol{F}_i(k)(\boldsymbol{x}^t) - \boldsymbol{F}_i(k)(\boldsymbol{x}^{t-1}))\}}{\sum_{k'=1}^{n_i} \boldsymbol{x}_i^t(k') \cdot \exp\{\eta \cdot (2\boldsymbol{F}_i(k')(\boldsymbol{x}^t) - 2\boldsymbol{F}_i(k')(\boldsymbol{x}^{t-1}))\}}, \tag{59}$$

for any $i \in [1:d]$ and $k \in [1:n_i]$. Therefore, for any $i \in [1:d]$ and $t \in [1:T-1]$, we obtain

$$\max\left\{ \left\| \frac{\boldsymbol{x}_i^t}{\boldsymbol{x}_i^{t+1}} \right\|_\infty, \left\| \frac{\boldsymbol{x}_i^{t+1}}{\boldsymbol{x}_i^t} \right\|_\infty \right\} \leq \exp\{6\eta(\Theta_1 + \Theta_2)\} \underset{(a)}{=} (1 + 7\eta(\Theta_1 + \Theta_2)), \tag{60}$$

where (a) is derived from the fact that $\exp(x) \leq 1 + \frac{7}{6}x$ for $x \in [0, 1/24]$. $\square$

### B.1.3 The Last Step

With the preparatory work for proving Theorem 3.5 is completed, we now turn to providing the final proof:

*Proof of Theorem 3.5.* Applying Theorem B.5 and Lemma B.7 to Lemma D.7, we have

$$\sum_{t=1}^T \mathrm{Var}_{\boldsymbol{x}_i^t}(\boldsymbol{F}_i(\boldsymbol{x}^t) - \boldsymbol{F}_i(\boldsymbol{x}^{t-1})) \leq 2\beta_0 \sum_{t=1}^T \mathrm{Var}_{\boldsymbol{x}_i^t}(\boldsymbol{F}_i(\boldsymbol{x}^{t-1})) + 165120\Theta^2(1 + 7\eta\Theta)H^5 + 2. \tag{61}$$

According to the result of Lemma B.4, we obtain

$$\sum_{t=1}^T (f(\boldsymbol{x}^t) - f(\boldsymbol{x}^*)) \leq \sum_{i=1}^d \frac{1}{\gamma_i} \left[ \frac{\log(n_i)}{\eta} - \left(\hat{g}_2(\eta\Theta)\eta\Theta^2 - 2\beta_0\hat{g}_1(\eta\Theta)\eta\Theta^2\right) \sum_{t=1}^T \mathrm{Var}_{\boldsymbol{x}_i^t}(\boldsymbol{F}_i(\boldsymbol{x}^t)) \right]$$

$$+ \hat{g}_1(\eta\Theta)\eta\Theta^2 \sum_{i=1}^d \frac{1}{\gamma_i}[8\beta_0\Theta^2 + 165120\Theta^2(1 + 7\eta\Theta)H^5 + 2]. \tag{62}$$

Combining Assumptions 3.3 and parameters selection Eq. (7), we complete the proof. $\square$

### B.2 Simple Example

In this section, we provide the proof of Example 3.4 which satisfies GQC condition and Assumption 3.3.

*Proof of Example 3.4.* Recalling the objective function $f(\boldsymbol{p}, \mathbf{P}) = \frac{1}{2}\mathbb{E}_{\boldsymbol{x},y}(\sum_{i=1}^m \boldsymbol{p}_i \sigma(\boldsymbol{x}^\top \mathbf{P}_i) - y)^2$, we have

$$f(\boldsymbol{p}^*, \mathbf{P}) - f(\boldsymbol{p}, \mathbf{P}) \geq \langle \boldsymbol{F}_{\boldsymbol{p}}(\boldsymbol{p}, \mathbf{P}), \boldsymbol{p}^* - \boldsymbol{p} \rangle, \tag{63}$$

since $f(\cdot, \mathbf{P})$ is convex for any fixed $\mathbf{P}$. In addition, we obtain

$$
\begin{aligned}
f(\boldsymbol{p}^*, \mathbf{P}^*) - f(\boldsymbol{p}^*, \mathbf{P}) &= -\frac{1}{2}\mathbb{E}\left[(\sigma(\boldsymbol{x}^\top \mathbf{P}_1) - \sigma(\boldsymbol{x}^\top \mathbf{P}_1^*))^2\right] \\
&\overset{(a)}{\geq} \frac{B_C}{2}\mathbb{E}\left[\langle \sigma(\boldsymbol{x}^\top \mathbf{P}_1) - \sigma(\boldsymbol{x}^\top \mathbf{P}_1^*), \boldsymbol{x}^\top(\mathbf{P}_1^* - \mathbf{P}_1)\rangle\right] \\
&= \frac{B_C}{2}\mathbb{E}\left[(\sigma(\boldsymbol{x}^\top \mathbf{P}_1^*) - y)\boldsymbol{x}, \mathbf{P}_1^* - \mathbf{P}_1\right],
\end{aligned}
\tag{64}
$$

where $B_C$ is a constant depends on $C$ and (a) is derived from the fact that $B_C \langle \exp\{x_1\} - \exp\{x_2\}, x_1 - x_2\rangle \geq |\exp\{x_1\} - \exp\{x_2\}|^2$ for any $x_1, x_2 \in [-C, C]$. Therefore, summing up Eq. (63) and Eq. (64), we have that $f$ satisfies GQC condition with the internal functions $\boldsymbol{F}_{\boldsymbol{p}} = \{\mathbb{E}[(\sum_{j=1}^m \boldsymbol{p}_j \sigma(\boldsymbol{x}^\top \mathbf{P}_j) - y)]\sigma(\boldsymbol{x}^\top \mathbf{P}_i)\}_{i=1}^m$ for block $\boldsymbol{p}$ and $\boldsymbol{F}_{\mathbf{P}_i} = \mathbb{E}\left[(\sigma(\boldsymbol{x}^\top \mathbf{P}_i) - y)\boldsymbol{x}\right]$ for block $\mathbf{P}_i$. Notice that $\gamma_{\boldsymbol{p}} = 1$, $\gamma_{\mathbf{P}_1} = \frac{B_C}{2}$ and $\gamma_{\mathbf{P}_i} = 0$ for any $i \neq 1$. Furthermore, we have $\|D^{\boldsymbol{\alpha}} \boldsymbol{F}_{\boldsymbol{p}}(\cdot)\|_\infty \leq 2\exp\{C\}(2C)^{|\boldsymbol{\alpha}|}$ and $\|D^{\boldsymbol{\alpha}} \boldsymbol{F}_{\mathbf{P}_i}(\cdot)\|_\infty \leq \exp\{C\}C^{|\boldsymbol{\alpha}|+1}$ by using and $\boldsymbol{x} \in [-C, C]^d$. According to Proposition B.2, we complete the proof. $\qquad \square$

There is also a toy example satisfying GQC condition and Assumption 3.3.

*Example* B.8. Assuming $(\boldsymbol{p}_1, \boldsymbol{p}_2) \in \Delta_m \times \Delta_n$, the function $f(\boldsymbol{p}_1, \boldsymbol{p}_2) = \frac{1}{2}\|\boldsymbol{p}_1 \boldsymbol{p}_2^\top\|_{\mathrm{F}}^2$ satisfies GQC condition and Assumption 3.3 with the internal functions $\boldsymbol{F}_{\boldsymbol{p}_1} = \|\boldsymbol{p}_2\|^2 \boldsymbol{p}_1$ for block $\boldsymbol{p}_1$ and $\boldsymbol{F}_{\boldsymbol{p}_2} = \boldsymbol{p}_2$ for block $\boldsymbol{p}_2$.

*Proof.* We have $\frac{1}{2}\|(\boldsymbol{p}_1^*)^\top \boldsymbol{p}_2\|_F - \frac{1}{2}\|\boldsymbol{p}_1^\top \boldsymbol{p}_2\|_F \geq \|\boldsymbol{p}_2\|^2 \boldsymbol{p}_1^\top(\boldsymbol{p}_1^* - \boldsymbol{p}_1)$ and $\frac{1}{2}\|(\boldsymbol{p}_1^*)^\top \boldsymbol{p}_2^*\|_F - \frac{1}{2}\|(\boldsymbol{p}_1^*)^\top \boldsymbol{p}_2\|_F \geq \|\boldsymbol{p}_1^*\|^2 \boldsymbol{p}_2^\top(\boldsymbol{p}_2^* - \boldsymbol{p}_2)$. Therefore, we have that $f$ satisfies GQC condition with the internal functions $\boldsymbol{F}_{\boldsymbol{p}_1} = \|\boldsymbol{p}_2\|^2 \boldsymbol{p}_1$ for block $\boldsymbol{p}_1$ and $\boldsymbol{F}_{\boldsymbol{p}_2} = \boldsymbol{p}_2$ for block $\boldsymbol{p}_2$. Notice that $\gamma_{\boldsymbol{p}_1} = 1$ and $\gamma_{\boldsymbol{p}_2} = \|\boldsymbol{p}_1^*\|^2$. Since both $\|\boldsymbol{p}_2\|^2 \boldsymbol{p}_1$ and $\boldsymbol{p}_2$ are polynomials with respect to $(\boldsymbol{p}_1, \boldsymbol{p}_2)$, we derive that the internal function of $f$ satisfies Assumption 3.3. $\qquad \square$

### B.3 Application to Reinforcement Learning

#### B.3.1 Analysis of Infinite Horizon Reinforcement Learning

*Proof of Proposition 3.6.* The following performance difference lemma [30, 13, 2, 35] plays an important role in the policy gradient based model of infinite horizon reinforcement learning problems,

$$
V^{\boldsymbol{\pi}^*}(\boldsymbol{\rho}_0) - V^{\boldsymbol{\pi}}(\boldsymbol{\rho}_0) = \mathbb{E}_{s \sim \boldsymbol{d}_{\boldsymbol{\rho}_0}^{\boldsymbol{\pi}^*}} \langle A^{\boldsymbol{\pi}}(s, \cdot), \boldsymbol{\pi}^*(\cdot|s) - \boldsymbol{\pi}(\cdot|s)\rangle.
\tag{65}
$$

Let $d = |\mathcal{S}|, \mathcal{S} = \{s_i\}_{i=1}^d$ and write $1/\gamma_i = \boldsymbol{d}_{\boldsymbol{\rho}_0}^{\boldsymbol{\pi}^*}(s_i)$, $\boldsymbol{F}_i(\boldsymbol{\pi}) = \boldsymbol{Q}^{\boldsymbol{\pi}}(s_i, \cdot)$. According to Eq. (65) whose proof is given in Cheng et al. [13] and $\langle A^{\boldsymbol{\pi}}(s_i, \cdot), \boldsymbol{\pi}'(\cdot|s_i) - \boldsymbol{\pi}(\cdot|s)\rangle = \langle \boldsymbol{Q}^{\boldsymbol{\pi}}(s_i, \cdot), \boldsymbol{\pi}'(\cdot|s_i) - \boldsymbol{\pi}(\cdot|s)\rangle$ for any policy $\boldsymbol{\pi}$ and $\boldsymbol{\pi}'$, we obtain that

$$
V^{\boldsymbol{\pi}^*}(\boldsymbol{\rho}_0) - V^{\boldsymbol{\pi}}(\boldsymbol{\rho}_0) = \sum_{i=1}^d \frac{1}{\gamma_i} \langle \boldsymbol{F}_i(\boldsymbol{\pi}), \boldsymbol{\pi}^*(\cdot|s_i) - \boldsymbol{\pi}(\cdot|s_i)\rangle.
\tag{66}
$$

Eq. (66) implies that $V^{\boldsymbol{\pi}}(\boldsymbol{\rho}_0)$ satisfies GQC condition. For every $a \in \mathcal{A}$, the Taylor expansion of $Q^{\boldsymbol{\pi}}(s_i, a)$ up to $K$-th order at origin is the same as its truncation at horizon $K$, which indicates

$$
R_{K,0}^{Q^{\boldsymbol{\pi}}(s_i, a)}(\boldsymbol{\pi}) = \theta^{K+1}\mathbb{E}_{s_{K+1}}[V^{\boldsymbol{\pi}}(s_{K+1})|s_0 = s_i, a_0 = a] \leq \theta^{K+1}.
$$

Therefore, according to the fact that

$$
P_{K,0}^{Q^{\boldsymbol{\pi}}(s_i, a)}(\boldsymbol{\pi}) \leq Q^{\boldsymbol{\pi}}(s_i, a) \leq 1,
$$

we have that $\boldsymbol{Q}^{\boldsymbol{\pi}}(s_i, \cdot)$ satisfies Assumption 3.3 with $\Theta_1 = \theta$, $\Theta_2 = 1$ and $K_0 = 1$. $\qquad \square$

### B.3.2 Analysis of Finite Horizon Reinforcement Learning

The function structure of finite horizon reinforcement learning on policy is strictly polynomial. Moreover, since the action-value functions on horizon $h$ is only dependent of policy $\boldsymbol{\pi}_{h+1:H}$, we may therefore verify that the objective function of finite horizon reinforcement learning satisfies GQC condition by utilizing finite difference expansion on function error $J_1^{\boldsymbol{\pi}}(\boldsymbol{\rho}_1) - J_1^{\boldsymbol{\pi}^*}(\boldsymbol{\rho}_1)$.

The finite horizon reinforcement learning considers the following policy optimization problem:

$$\min_{\boldsymbol{\pi} \in \mathcal{X}} J_1^{\boldsymbol{\pi}}(\boldsymbol{\rho}_1), \tag{67}$$

where $J_1^{\boldsymbol{\pi}}(\boldsymbol{\rho}_1) = \mathbb{E}_{s_1 \sim \boldsymbol{\rho}_1}[V_1^{\boldsymbol{\pi}}(s_1)]$, and $\mathcal{X} = \mathcal{X}_1 \times \cdots \times \mathcal{X}_H$, and each $\mathcal{X}_h$ denotes $|\mathcal{S}_h|$ probability simplexes. We write $\mathcal{S}_h = \{s_{h,i_h}\}_{i_h=1}^{|\mathcal{S}_h|}$ for any $h \in [1:H]$ and denote the action-value vector on state $s_{h,i_h}$ at horizon $h$ by $\boldsymbol{Q}_h^{\boldsymbol{\pi}_{h+1:H}}(s_{h,i_h}, \cdot)$. According to the definition of finite horizon value function $V_h^{\boldsymbol{\pi}_{h:H}}$, we obtain the observation as Eq. (68).

$$
\begin{aligned}
J_1^{\boldsymbol{\pi}^*}(\boldsymbol{\rho}_1) - J_1^{\boldsymbol{\pi}}(\boldsymbol{\rho}_1) &= \sum_{h=1}^{H} \left[ J_1^{\boldsymbol{\pi}^*_{1:h}, \boldsymbol{\pi}_{h+1:H}}(\boldsymbol{\rho}_1) - J_1^{\boldsymbol{\pi}^*_{1:h-1}, \boldsymbol{\pi}_{h:H}}(\boldsymbol{\rho}_1) \right] \\
&= \sum_{h=1}^{H} \mathbb{E}_{s_h \sim \mathbb{E}_{s_1 \sim \boldsymbol{\rho}_1} \mathbf{Pr}_h^{\boldsymbol{\pi}^*_{1:h-1}}(\cdot|s_1)} \left\langle \boldsymbol{Q}_h^{\boldsymbol{\pi}_{h+1:H}}(s_h, \cdot), \boldsymbol{\pi}_h^*(\cdot|s_h) - \boldsymbol{\pi}_h(\cdot|s_h) \right\rangle,
\end{aligned}
\tag{68}
$$

Since $\boldsymbol{Q}_h^{\boldsymbol{\pi}_{h+1:H}}(s_h, a_h)$ is a polynomial with respect to policy $\boldsymbol{\pi}_{1:H}$, whose value is bounded by $1 + H - h$ for any $s_h \in \mathcal{S}_h$ and $a_h \in \mathcal{A}_h$, we derive that $J_1^{\boldsymbol{\pi}}(\boldsymbol{\rho}_1)$ satisfies GQC condition with internal function $\boldsymbol{F}_{h,i_h}(\boldsymbol{\pi}) = \boldsymbol{Q}_h^{\boldsymbol{\pi}_{h+1:H}}(s_{h,i_h}, \cdot)$ for variable block $\boldsymbol{x}_{h,i_h}$ where $i_h \in [1:|\mathcal{S}_h|]$, and $\boldsymbol{F}$ satisfies Assumption 3.3 with $\theta = 0$, $\Theta_1 = 0$, $\Theta_2 = H$ and $K_0 = H$. Therefore, for finite horizon reinforcement learning, it follows from Theorem 3.5 that Algorithm 1 with parameter selection Eq. (7) finds an $\varepsilon$-suboptimal global solution in a number of iterations that is at most $\mathcal{O}(H \max_{h \in [0:H]} \log(|\mathcal{A}_h|) \varepsilon^{-1} \log^4(\varepsilon^{-1}))$.

## C  Minimax Optimization

We begin with showing the connection between GQCC condition and GQC condition. Without loss of generality, we assume $n_i = n$ and $m_i = m$ for any $i \in [1:d]$, and let $\ell = n + m$. If $f(\cdot, \boldsymbol{y})$ and $-f(\boldsymbol{x}, \cdot)$ satisfy GQC condition with respect to a pair of minimizers $\boldsymbol{x}^*(\boldsymbol{y})$ and $\boldsymbol{y}^*(\boldsymbol{x})$, respectively, then we have the following estimations of function error

$$f(\boldsymbol{x}, \boldsymbol{y}) - f(\boldsymbol{x}^*(\boldsymbol{y}), \boldsymbol{y}) \leq \sum_{i=1}^{d} \frac{1}{\gamma_i(\boldsymbol{y})} \left( f_i(\mathbf{P}(\boldsymbol{z}), \boldsymbol{x}_i, \boldsymbol{y}_i) - f_i(\mathbf{P}(\boldsymbol{z}), \boldsymbol{x}^*(\boldsymbol{y})_i, \boldsymbol{y}_i) \right), \tag{69}$$

$$f(\boldsymbol{x}, \boldsymbol{y}^*(\boldsymbol{x})) - f(\boldsymbol{x}, \boldsymbol{y}) \leq \sum_{i=1}^{d} \frac{1}{\tau_i(\boldsymbol{x})} \left( f_i(\mathbf{P}(\boldsymbol{z}), \boldsymbol{x}_i, \boldsymbol{y}^*(\boldsymbol{x})_i) - f_i(\mathbf{P}(\boldsymbol{z}), \boldsymbol{x}_i, \boldsymbol{y}_i) \right), \tag{70}$$

where $f_i(\mathbf{Q}, \boldsymbol{z}_i) = \langle \mathbf{Q}_i, \boldsymbol{z}_i \rangle$ for any $\mathbf{Q} \in \mathbb{R}^{\ell \times d}$ and $\boldsymbol{z}_i \in \mathbb{R}^{n+m}$, and each $\mathbf{P}_i$ includes the internal function of $f(\cdot, \boldsymbol{y})$ for variable block $\boldsymbol{x}_i$ and the internal function of $-f(\boldsymbol{x}, \cdot)$ for variable block $\boldsymbol{y}_i$. It follows from Eq. (69) and Eq. (70) that Eq. (10) holds for $f$ with $\psi_i(\boldsymbol{z}) = \max\{1/\gamma_i(\boldsymbol{y}), 1/\tau_i(\boldsymbol{x})\}$.

### C.1  Preparatory Discussion

In this section, we provide the convergence analysis of general version of Algorithm 2, i.e., Algorithm 4. We consider the divergence-generating function $v$ with Bregman's divergence $V$ (i.e., $V(\boldsymbol{x}, \boldsymbol{u}) = v(\boldsymbol{x}) - v(\boldsymbol{u}) - \langle \nabla v(\boldsymbol{u}), \boldsymbol{x} - \boldsymbol{u} \rangle$ for any $\boldsymbol{x}, \boldsymbol{u}$) over general compact convex regions $\mathcal{Z} = \mathcal{X} \times \mathcal{Y} \subset \mathbb{R}^{\sum_{i=1}^{d} n_i} \times \mathbb{R}^{\sum_{i=1}^{d} m_i}$. Before we introduce the main theorem, we need the following assumptions:

**Assumption C.1.** There exists positive constants $A, D$ such that

[$\mathbf{A_1}$] $\max_{i \in [1:d]} \{\|\boldsymbol{z}_i\|\} \leq A$ uniformly on $\mathcal{Z}$.

**[A₂]** $\max\left\{\max_{\substack{z_i\in\mathcal{Z}_i\\ i\in[1:d]}}\left(V\left(x_i,(g_i^x)^0\right)+V\left(y_i,(g_i^y)^0\right)\right),\max_{\substack{z_i\in\mathcal{Z}_i\\ i\in[1:d]}}\left(v(x_i)+v(y_i)\right)\right\}\leq D.$

**[A₃]** $v$ modulus 2 with respect to $\|\cdot\|$ (i.e., $\forall i\in[1:d]$, $V(x_i,u_i)\geq\|x_i-u_i\|^2$ for any $x_i,u_i\in\mathcal{X}_i$ and $V(y_i,w_i)\geq\|y_i-w_i\|^2$ for any $y_i,w_i\in\mathcal{Y}_i$).

If we choose $v(x)=\sum_{j=1}^n x(j)\log(x(j))$ and $\|\cdot\|=\|\cdot\|_1$, then (1) in Assumption C.1 holds with $A=2$; (2) in Assumption C.1 holds with $D=2\max_{i\in[1:d]}\{\log(n_i)+\log(m_i)\}$; (3) in Assumption C.1 holds following Pinsker's inequality. According to Remark C.2, we state that there exist some compact convex regions in $\mathbb{R}^{\sum_{i=1}^d(n_i+m_i)}$ with proper divergence-generating function $v$ and proper choice of $g^0$ satisfy Assumption C.1.

*Remark* C.2. If the feasible region $\mathcal{Z}$ is a compact set of Euclidean space, then it is reasonable that assuming the divergence-generating function $v$ (i.e., $v(x)=\sum_{j=1}^n x(j)\log(x(j))$ over the probability simplex or $v(x)=\|x\|_2^2$ over the standard compact set) and the norm $\|\cdot\|$ are uniformly bounded on every $\mathcal{Z}_i$. For some Bregman divergences, if $x^0$ is a fixed point, $V(\cdot,x^0)$ can be bounded by a constant (may depend on the dimension of space) on a compact feasible region, such as $V(\cdot,x^0)=\|\cdot-x^0\|_2^2$ with $x^0=0$ on the closed ball $\mathbb{B}_R(0)$ for radius $R\in(0,\infty)$ and $V(\cdot,x^0)=\mathrm{KL}(\cdot\|x^0)$ with $x^0=(1/n,\cdots,1/n)$ on the probability simplex $\Delta_n$.

**Assumption C.3.** In Definition 4.1, let matrix-valued function $\mathbf{P}$ has the form of $\mathbf{P}(\mathbf{Q}^z,z)$ where $\mathbf{Q}^z\in\mathbb{R}^{\ell\times d}$ depends on $z$, and assume that $\mathbf{P}$ satisfies the following properties on region $\left\{\mathbf{Q}\in\mathbb{R}^{\ell\times d}\mid\|\mathbf{Q}\|_\infty\leq C\right\}\times\mathcal{Z}$ for some constant $C>0$:

**[A₄]** There exist constants $L_1,L_2\geq 0$ such that $F_i(\cdot,z_i)$ is uniformly $L_1$-Lipschitz continuous with respect to $\|\cdot\|_*$ under $\|\cdot\|_\infty$, and $F_i(\mathbf{P},\cdot)$ is uniformly $L_2$-Lipschitz continuous with respect to $\|\cdot\|_*$ under $\|\cdot\|$.

**[A₅]** There are a positive constant $\gamma>0$ and a pair of sets of matrices $\left\{\{\mathbf{B}_i\}_{i=1}^d,\{\mathbf{C}_i\}_{i=1}^d\right\}\subset\mathbb{R}_+^{\ell\times d}\cup\mathbf{0}$ satisfying $\left\|\sum_{i=1}^d(\mathbf{B}_i+\mathbf{C}_i)\right\|_\infty\leq\gamma$, such that the following bounds hold

$$\mathbf{D}_{\mathbf{P}(\mathbf{Q},\cdot,y)}(x,x')\leq\sum_{i=1}^d\mathbf{C}_i\left\langle F_i^x(\mathbf{Q},z_i),x_i-x_i'\right\rangle,$$

$$\mathbf{D}_{\mathbf{P}(\mathbf{Q},x,\cdot)}(y',y)\leq\sum_{i=1}^d\mathbf{B}_i\left\langle -F_i^y(\mathbf{Q},z_i),y_i'-y_i\right\rangle,$$

for any $y,y'\in\mathcal{Y}$ and $x,x'\in\mathcal{X}$.

**[A₆]** There exists $\theta\in[0,1)$ such that $\mathbf{P}(\cdot,z)$ is a $\theta$-contraction mapping under $\|\cdot\|_\infty$, and $\|\mathbf{P}(\mathbf{Q},z)\|_\infty\leq C$ for any $z\in\mathcal{Z}$.

**Lemma C.4** (General Version of Lemma 4.3). *Assuming that Assumption C.1 and C.3 hold, $[\mathbf{P}(\mathbf{Q},\cdot,\cdot)]_{k,j}$ is continuous, and convex with respect to $x$, and concave with respect to $y$ for any $(k,j)$, and $\min_{\substack{k,j\\ i\in[1:d]}}\frac{\min\{[\mathbf{C}_i]_{k,j},[\mathbf{B}_i]_{k,j}\}}{[\mathbf{C}_i]_{k,j}+[\mathbf{B}_i]_{k,j}}\geq C'$ for some $C'>0$, then we claim that there exist $\mathbf{Q}^*\in\mathbb{R}^{\ell\times d}$ and $z^*\in\mathcal{Z}$ such that*

$$\mathbf{Q}^*=\mathbf{P}(\mathbf{Q}^*,x^*,y^*),\tag{71}$$
$$\mathbf{Q}^*\leq\mathbf{P}(\mathbf{Q}^*,x,y^*),\tag{72}$$
$$\mathbf{Q}^*\geq\mathbf{P}(\mathbf{Q}^*,x^*,y).\tag{73}$$

*Proof.* We shall begin the proof by proving the following lemma.

**Lemma C.5.** *Under the conditions of Lemma C.4, it can be proven that for any $\mathbf{Q}\in\mathbb{R}^{\ell\times d}$, there exists a pair of $x^*,y^*$ that satisfy the following*

$$\mathbf{P}(\mathbf{Q},x^*,y)\leq\mathbf{P}(\mathbf{Q},x^*,y^*)\leq\mathbf{P}(\mathbf{Q},x,y^*).\tag{74}$$

*Proof.* Considering the following iteration

$$z_i^t = \operatorname*{argmin}_{z_i \in \mathcal{Z}_i} \eta \left\langle \boldsymbol{F}_i(\mathbf{Q}, z_i^{t-1}), z_i \right\rangle + V \left( \boldsymbol{x}_i, (\boldsymbol{g}_i^{\boldsymbol{x}})^{t-1} \right) + V \left( \boldsymbol{y}_i, (\boldsymbol{g}_i^{\boldsymbol{y}})^{t-1} \right),$$

$$g_i^t = \operatorname*{argmin}_{\boldsymbol{g}_i \in \mathcal{Z}_i} \eta \left\langle \boldsymbol{F}_i(\mathbf{Q}, z_i^t), \boldsymbol{g}_i \right\rangle + V \left( \boldsymbol{g}_i^{\boldsymbol{x}}, (\boldsymbol{g}_i^{\boldsymbol{x}})^{t-1} \right) + V \left( \boldsymbol{g}_i^{\boldsymbol{y}}, (\boldsymbol{g}_i^{\boldsymbol{y}})^{t-1} \right), \tag{75}$$

for any $i \in [1:d]$ and combining [57, Lemma 1], we have

$$[\mathbf{C}_i]_{k,j} \sum_{t=1}^{T} \left\langle \boldsymbol{F}_i^{\boldsymbol{x}}(\mathbf{Q}, z_i^t), \boldsymbol{x}_i^t - \boldsymbol{x}_i' \right\rangle + [\mathbf{B}_i]_{k,j} \sum_{t=1}^{T} \left\langle \boldsymbol{F}_i^{\boldsymbol{y}}(\mathbf{Q}, z_i^t), \boldsymbol{y}_i^t - \boldsymbol{y}_i' \right\rangle$$

$$\leq ([\mathbf{C}_i]_{k,j} + [\mathbf{B}_i]_{k,j}) \eta^{-1} D + [\mathbf{C}_i]_{k,j} \sum_{t=1}^{T} \left\| \boldsymbol{F}_i^{\boldsymbol{x}}(\mathbf{Q}, z_i^t) - \boldsymbol{F}_i^{\boldsymbol{x}}(\mathbf{Q}, z_i^{t-1}) \right\|_* \left\| \boldsymbol{x}_i^t - (\boldsymbol{g}_i^{\boldsymbol{x}})^t \right\|$$

$$+ [\mathbf{B}_i]_{k,j} \sum_{t=1}^{T} \left\| \boldsymbol{F}_i^{\boldsymbol{y}}(\mathbf{Q}, z_i^t) - \boldsymbol{F}_i^{\boldsymbol{y}}(\mathbf{Q}, z_i^{t-1}) \right\|_* \left\| \boldsymbol{y}_i^t - (\boldsymbol{g}_i^{\boldsymbol{y}})^t \right\|$$

$$- \frac{[\mathbf{C}_i]_{k,j}}{\eta} \sum_{t=1}^{T} \left( \left\| \boldsymbol{x}_i^t - (\boldsymbol{g}_i^{\boldsymbol{x}})^t \right\|^2 + \left\| (\boldsymbol{g}_i^{\boldsymbol{x}})^{t-1} - \boldsymbol{x}_i^t \right\|^2 \right)$$

$$- \frac{[\mathbf{B}_i]_{k,j}}{\eta} \sum_{t=1}^{T} \left( \left\| \boldsymbol{y}_i^t - (\boldsymbol{g}_i^{\boldsymbol{y}})^t \right\|^2 + \left\| (\boldsymbol{g}_i^{\boldsymbol{y}})^{t-1} - \boldsymbol{y}_i^t \right\|^2 \right)$$

$$\underset{(a)}{\leq} ([\mathbf{C}_i]_{k,j} + [\mathbf{B}_i]_{k,j}) \eta^{-1} D + \frac{\eta L_2^2 ([\mathbf{C}_i]_{k,j} + [\mathbf{B}_i]_{k,j})}{2} \sum_{t=1}^{T} \left\| z_i^t - z_i^{t-1} \right\|^2$$

$$- \frac{\min\{[\mathbf{C}_i]_{k,j}, [\mathbf{B}_i]_{k,j}\}}{\eta} \sum_{t=1}^{T} \left( \frac{1}{4} \left\| z_i^t - \boldsymbol{g}_i^t \right\|^2 + \frac{1}{2} \left\| \boldsymbol{g}_i^{t-1} - z_i^t \right\|^2 \right), \tag{76}$$

for any $i \in [1:d]$, $(k,j) \in [1:\ell] \times [1:d]$, and $z_i' \in \mathcal{Z}_i$, where (a) is derived from $\mathbf{A_4}$ in Assumption C.3 and Cauchy-Schwarz inequality. Therefore, by setting $\eta = \frac{\sqrt{C'}}{2L_2}$ and $\frac{1}{T} \sum_{t=1}^{T} z^t = \bar{z}_T = (\bar{\boldsymbol{x}}_T, \bar{\boldsymbol{y}}_T)$, the following estimation holds for any $(k,j)$

$$\max_{\boldsymbol{z}' = (\boldsymbol{x}', \boldsymbol{y}') \in \mathcal{Z}} [\mathbf{P}(\mathbf{Q}, \bar{\boldsymbol{x}}_T, \boldsymbol{y}') - \mathbf{P}(\mathbf{Q}, \boldsymbol{x}', \bar{\boldsymbol{y}}_T)]_{k,j}$$

$$\underset{(b)}{\leq} \frac{1}{T} \sum_{t=1}^{T} \left[ \mathbf{P}(\mathbf{Q}, \boldsymbol{x}^t, \bar{\boldsymbol{y}}_T^*) - \mathbf{P}(\mathbf{Q}, \bar{\boldsymbol{x}}_T^*, \boldsymbol{y}^t) \right]_{k,j}$$

$$\underset{(c)}{\leq} \frac{1}{T} \sum_{i=1}^{d} \sum_{t=1}^{T} \left( [\mathbf{C}_i]_{k,j} \left\langle \boldsymbol{F}_i(\mathbf{Q}, z_i^t), \boldsymbol{x}_i^t - (\bar{\boldsymbol{x}}_T^*)_i \right\rangle + [\mathbf{B}_i]_{k,j} \left\langle \boldsymbol{F}_i(\mathbf{Q}, z_i^t), \boldsymbol{y}_i^t - (\bar{\boldsymbol{y}}_T^*)_i \right\rangle \right)$$

$$\leq \frac{2\gamma \eta^{-1} D + 4\eta \gamma A^2 L_2^2}{T}, \tag{77}$$

where the convexity of function $[\mathbf{P}(\mathbf{Q}, \cdot, \boldsymbol{w}) - \mathbf{P}(\mathbf{Q}, \boldsymbol{u}, \cdot)]_{k,j}$ for fixed $\mathbf{Q}$ and $\boldsymbol{v} = (\boldsymbol{u}, \boldsymbol{w})$ implies (b), and (c) is derived from Eq. (76) and the definition that $(\bar{\boldsymbol{x}}_T^*, \bar{\boldsymbol{y}}_T^*) := \operatorname*{argmax}_{\boldsymbol{z}' \in \mathcal{Z}} [\mathbf{P}(\mathbf{Q}, \bar{\boldsymbol{x}}_T, \boldsymbol{y}') - \mathbf{P}(\mathbf{Q}, \boldsymbol{x}', \bar{\boldsymbol{y}}_T)]_{k,j}$. Since $\mathcal{Z}$ is a compact set, the sequence $\{(\bar{\boldsymbol{x}}_T, \bar{\boldsymbol{y}}_T)\}_{T=1}^{\infty}$ must have a convergent subsequence. Therefore, all accumulation points of the sequence $\{(\bar{\boldsymbol{x}}_T, \bar{\boldsymbol{y}}_T)\}_{T=1}^{\infty}$ satisfy Eq. (74) by using the continuity of $\mathbf{P}(\mathbf{Q}, \cdot, \cdot)$. □

Now, we define the iterately update as follows

$$\mathbf{Q}^{t+1} = \mathbf{P}(\mathbf{Q}^t, \boldsymbol{x}_t^*, \boldsymbol{y}_t^*), \tag{78}$$

where $(\boldsymbol{x}_t^*, \boldsymbol{y}_t^*)$ satisfies Eq. (74) in Lemma C.5 w.r.t $\mathbf{P}(\mathbf{Q}^t, \cdot, \cdot)$. It's direct to derive that

$$
\begin{aligned}
\mathbf{Q}^{t+1} - \mathbf{Q}^t &\leq \mathbf{P}(\mathbf{Q}^t, \boldsymbol{x}_{t-1}^*, \boldsymbol{y}_t^*) - \mathbf{P}(\mathbf{Q}^{t-1}, \boldsymbol{x}_{t-1}^*, \boldsymbol{y}_t^*) \\
&\leq \theta \left\| \mathbf{Q}^t - \mathbf{Q}^{t-1} \right\|_\infty,
\end{aligned}
\tag{79}
$$

$$
\begin{aligned}
\mathbf{Q}^{t+1} - \mathbf{Q}^t &\geq \mathbf{P}(\mathbf{Q}^t, \boldsymbol{x}_t^*, \boldsymbol{y}_{t-1}^*) - \mathbf{P}(\mathbf{Q}^{t-1}, \boldsymbol{x}_t^*, \boldsymbol{y}_{t-1}^*) \\
&\geq -\theta \left\| \mathbf{Q}^t - \mathbf{Q}^{t-1} \right\|_\infty.
\end{aligned}
\tag{80}
$$

Finally, according to the contraction mapping principle, we complete the proof. $\qquad\square$

**Corollary C.6.** *Assuming preconditions of Lemma C.4 hold, and letting $\{f_i(\mathbf{Q}, \cdot) : \mathbb{R}^{n_i+m_i} \to \mathbb{R}\}_{i=1}^d$ be a sequence of continuous convex-concave functions which satisfies $\nabla f_i(\mathbf{Q}, \cdot) = (\boldsymbol{F}_i^{\boldsymbol{x}}(\mathbf{Q}, \cdot), -\boldsymbol{F}_i^{\boldsymbol{y}}(\mathbf{Q}, \cdot))$ for any fixed $\mathbf{Q} \in \mathbb{R}^{\ell \times d}$ and $i \in [1 : d]$, then there exist a matrix $\mathbf{Q}^*$ and a pair of $(\boldsymbol{x}^*, \boldsymbol{y}^*)$ which satisfy Eq. (71)-Eq. (73) and*

$$
\begin{aligned}
f_i(\mathbf{Q}^*, \boldsymbol{x}_i^*, \boldsymbol{y}_i^*) &\geq f_i(\mathbf{Q}^*, \boldsymbol{x}_i^*, \boldsymbol{y}_i), \\
f_i(\mathbf{Q}^*, \boldsymbol{x}_i^*, \boldsymbol{y}_i^*) &\leq f_i(\mathbf{Q}^*, \boldsymbol{x}_i, \boldsymbol{y}_i^*),
\end{aligned}
$$

*for any $\boldsymbol{z}_i \in \mathcal{Z}_i$ and $i \in [1 : d]$.*

*Proof.* With proper selection of $\eta$, we have the following bound which is similar to that derived from Eq.(77)

$$
\begin{aligned}
&\max_{\boldsymbol{y}_i' \in \mathcal{Y}_i} f_i(\mathbf{Q}, (\bar{\boldsymbol{x}}_T)_i, \boldsymbol{y}_i') - \min_{\boldsymbol{x}_i' \in \mathcal{X}_i} f_i(\mathbf{Q}, \boldsymbol{x}_i', (\bar{\boldsymbol{y}}_T)_i) \\
&\leq \sum_{t=1}^T [f_i(\mathbf{Q}, \boldsymbol{x}_i^t, (\bar{\boldsymbol{y}}_T^*)_i) - f_i(\mathbf{Q}, (\bar{\boldsymbol{x}}_T^*)_i, \boldsymbol{y}_i^t)] \\
&\leq \sum_{t=1}^T \left[ \langle \boldsymbol{F}_i(\mathbf{Q}, \boldsymbol{z}_i^t), \boldsymbol{x}_i^t - (\bar{\boldsymbol{x}}_T^*)_i \rangle + \langle \boldsymbol{F}_i(\mathbf{Q}, \boldsymbol{z}_i^t), \boldsymbol{y}_i^t - (\bar{\boldsymbol{y}}_T^*)_i \rangle \right] \\
&\leq \frac{4\eta^{-1}D + 8\eta A^2 L_2^2}{T},
\end{aligned}
\tag{81}
$$

for every $i \in [1 : d]$, where $\{\boldsymbol{z}_t = (\boldsymbol{x}_t, \boldsymbol{y}_t)\}_{t=1}^T$ follows from the iteration (75) and $(\bar{\boldsymbol{z}}_T^*)_i = ((\bar{\boldsymbol{x}}_T^*)_i, (\bar{\boldsymbol{y}}_T^*)_i)$ denotes $\operatorname*{argmax}_{\boldsymbol{z}_i' \in \mathcal{Z}_i} [f_i(\mathbf{Q}, (\bar{\boldsymbol{x}}_T)_i, \boldsymbol{y}_i') - f_i(\mathbf{Q}, \boldsymbol{x}_i', (\bar{\boldsymbol{y}}_T)_i)]$. Hence, by directly leveraging the result of Lemma 4.3, we obtain the result. $\qquad\square$

Before stating the general version of Theorem 4.4 as follows, we define

$$
Y_T^\eta = 8(c+1) \left[ \gamma D \left( \frac{1}{\eta} + 16\eta L_2 \right) + 40\eta^3 \gamma A^2 L_2^4 + 2\eta\gamma L_1^2 (1 + 64\eta^2 L_2^2 C^2) \right] (\log(c+T) + 1).
\tag{82}
$$

## C.2 Theorem C.7 and Relate Proof

**Theorem C.7.** *[General Version of Theorem 4.4] For any generaized quasar-convex-concave function $f$ which satisfies Assumption C.1 and C.3 with constant matrix function $\mathbf{P} \equiv \mathbf{Q}^*$, where $\mathbf{Q}^*$ is unknown and satisfies Eq. (71)-Eq. (73), with parameter configuration in Eq. (12), the weighted average of Algorithm 2's outputs $\{\boldsymbol{z}_t\}_{t=1}^T$ satisfies the following inequality*

$$
\mathcal{G}_f(\bar{\boldsymbol{x}}_T, \bar{\boldsymbol{y}}_T) \leq \frac{6\left(\max_{\boldsymbol{z} \in \mathcal{Z}} \sum_{i=1}^d \psi_i(\boldsymbol{z})\right)(1-\theta)^{-1}\left(\frac{3D}{\eta} + 10\eta L_1^2 + 5AL_1 Y_T^\eta + 4\eta A^2 L_2^2\right)}{T+3}.
\tag{83}
$$

For a generalized quasar-convex-concave function satisfying smoothness and recurrence conditions, the iteration complexity of our algorithm matches the lower bound [52] for solving $\varepsilon$-approximate Nash equilibrium points in the smooth convex-concave setting, up to a logarithmic factor. Furthermore, we prove that standard smooth convex-concave functions satisfy the preconditions of Theorem C.7 (as discussed in Appendix C.3.2).

**Algorithm 4** Optimistic Mirror Descent with Regularization for Multi-Variables

---

**Input:** $\{\boldsymbol{z}_i^0\}_{i=1}^d = \{\boldsymbol{g}_i^0\}_{i=1}^d$, $\{\alpha_t \geq 0\}_{t=1}^T$ ($\sum_{t=1}^T \alpha_t = 1$), $\{\gamma_t \geq 0\}_{t=1}^T$, $\{\lambda_t \geq 0\}_{t=1}^T$, $\eta$ and $\mathbf{Q}^0 = \mathbf{0}$.

**Output:** $\bar{\boldsymbol{z}}_T = \sum_{t=1}^T \alpha_t \boldsymbol{z}^t$.

1: **while** $t \leq T$ **do**
2:      $\mathbf{Q}^t = (1 - \beta_{t-1})\mathbf{Q}^{t-1} + \beta_{t-1}\mathbf{P}(\mathbf{Q}^{t-1}, \boldsymbol{z}_{t-1})$.
3:      **for all** $i \in [1:d]$ **do**
4:          $\boldsymbol{x}_i^t = \underset{\boldsymbol{x}_i \in \mathcal{X}_i}{\operatorname{argmin}} \; \eta \left\langle \boldsymbol{F}_i^{\boldsymbol{x}}(\mathbf{Q}^{t-1}, \boldsymbol{z}_i^{t-1}), \boldsymbol{x}_i \right\rangle + \gamma_t V\left(\boldsymbol{x}_i, (\boldsymbol{g}_i^{\boldsymbol{x}})^{t-1}\right) + \lambda_t v(\boldsymbol{x}_i)$,
5:          $\boldsymbol{y}_i^t = \underset{\boldsymbol{y}_i \in \mathcal{Y}_i}{\operatorname{argmin}} \; \eta \left\langle \boldsymbol{F}_i^{\boldsymbol{y}}(\mathbf{Q}^{t-1}, \boldsymbol{z}_i^{t-1}), \boldsymbol{y}_i \right\rangle + \gamma_t V\left(\boldsymbol{y}_i, (\boldsymbol{g}_i^{\boldsymbol{y}})^{t-1}\right) + \lambda_t v(\boldsymbol{y}_i)$,
6:          $(\boldsymbol{g}_i^{\boldsymbol{x}})^t = \underset{\boldsymbol{g}_i^{\boldsymbol{x}} \in \mathcal{X}_i}{\operatorname{argmin}} \; \eta \left\langle \boldsymbol{F}_i^{\boldsymbol{x}}(\mathbf{Q}^t, \boldsymbol{z}_i^t), \boldsymbol{g}_i^{\boldsymbol{x}} \right\rangle + \gamma_t V\left(\boldsymbol{g}_i^{\boldsymbol{x}}, (\boldsymbol{g}_i^{\boldsymbol{x}})^{t-1}\right) + \lambda_t v(\boldsymbol{g}_i^{\boldsymbol{x}})$,
7:          $(\boldsymbol{g}_i^{\boldsymbol{y}})^t = \underset{\boldsymbol{g}_i^{\boldsymbol{y}} \in \mathcal{Y}_i}{\operatorname{argmin}} \; \eta \left\langle \boldsymbol{F}_i^{\boldsymbol{y}}(\mathbf{Q}^t, \boldsymbol{z}_i^t), \boldsymbol{g}_i^{\boldsymbol{y}} \right\rangle + \gamma_t V\left(\boldsymbol{g}_i^{\boldsymbol{y}}, (\boldsymbol{g}_i^{\boldsymbol{y}})^{t-1}\right) + \lambda_t v(\boldsymbol{g}_i^{\boldsymbol{y}})$.
8:      **end for**
9:      $t \leftarrow t + 1$.
10: **end while**

---

Our analysis relies on the connection between $\boldsymbol{F}_i(\mathbf{Q}^t, \boldsymbol{z}_i^t)$ and $\boldsymbol{F}_i(\mathbf{Q}^*, \boldsymbol{z}_i^t)$. Theorem C.8 combines a) classical $\mathcal{O}(\log(T)/T)$ bound derived from regularized OMD, and b) the weighted average of iteration error $\|\mathbf{Q}^t - \mathbf{Q}^{t-1}\|_\infty^2$ over $t \in [1:T]$ which has the form of $\sum_{t=1}^T \alpha_t \|\mathbf{Q}^t - \mathbf{Q}^{t-1}\|_\infty^2$, with magnitude $\mathcal{O}(T^{-1}\log(T))$, and c) weighted average of approximation error $\|\mathbf{Q}^t - \mathbf{Q}^*\|_\infty$ over $t \in [1:T]$ which has the form of $\sum_{t=1}^T \alpha_t \|\mathbf{Q}^t - \mathbf{Q}^*\|_\infty$ to bound the max-min gap of $f$ at $\bar{\boldsymbol{z}}_T$. Next, we leverage Lemma C.9 to show the decreasing trend of approximation error $\|\mathbf{Q}^t - \mathbf{Q}^*\|_\infty$ and bound $\sum_{t=1}^T \alpha_t \|\mathbf{Q}^t - \mathbf{Q}^*\|_\infty$ by a quantity that grows only logarithmically in $T$. We may therefore obtain the result of Theorem C.7 by applying the estimation of weighted average of approximation error $\sum_{t=1}^T \alpha_t \|\mathbf{Q}^t - \mathbf{Q}^*\|_\infty$ to Theorem C.8.

### C.2.1   Part I

**Theorem C.8.** *Assuming that Assumption C.1 holds, we set the hyper-parameters for Algorithm 2 carefully such that*

$$\alpha_t(\gamma_t + \lambda_t) \geq \alpha_{t+1}\gamma_{t+1}, \tag{84}$$

$$\eta \leq \min_{t \in [1:T]} \frac{\sqrt{\gamma_t(\gamma_t + \lambda_t)}}{4L_2}. \tag{85}$$

*Suppose that $v$ modulus 2 w.r.t $\|\cdot\|$, $\|\boldsymbol{z}\| \leq A$ for any $\boldsymbol{z} \in \mathcal{Z}$, and $\{\boldsymbol{z}_t\}_{t=1}^T$ follows the iterations of Algorithm 4, then we can show that*

$$
\begin{aligned}
\max_{\boldsymbol{y}' \in \mathcal{Y}} f(\bar{\boldsymbol{x}}_T, \boldsymbol{y}') - \min_{\boldsymbol{x}' \in \mathcal{X}} f(\boldsymbol{x}', \bar{\boldsymbol{y}}_T) \leq & B(\psi) \max_{i \in [1:d]} \left\{ \frac{\alpha_1 \gamma_1}{\eta} \max_{\boldsymbol{z}_i \in \mathcal{Z}_i} \left( V\left(\boldsymbol{x}_i, (\boldsymbol{g}_i^{\boldsymbol{x}})^0\right) + V\left(\boldsymbol{y}_i, (\boldsymbol{g}_i^{\boldsymbol{y}})^0\right) \right) \right. \\
& - \frac{1}{2\eta} \sum_{t=1}^T \alpha_t \left[ \gamma_t \|\boldsymbol{z}_i^t - \boldsymbol{g}_i^{t-1}\|^2 + \frac{\gamma_t + \lambda_t}{2} \|\boldsymbol{g}_i^t - \boldsymbol{z}_i^t\|^2 \right] \\
& \left. + \frac{2\sum_{t=1}^T \alpha_t \lambda_t}{\eta} \max_{\boldsymbol{z}_i \in \mathcal{Z}_i} \left( v(\boldsymbol{x}_i) + v(\boldsymbol{y}_i) \right) \right\} \\
& + 2B(\psi)\eta L_1^2 \sum_{t=1}^T \left[ \frac{\alpha_t}{\gamma_t + \lambda_t} \|\mathbf{Q}^t - \mathbf{Q}^{t-1}\|_\infty^2 \right] \\
& + 2AB(\psi)L_1 \sum_{t=1}^T \alpha_t \|\mathbf{Q}^t - \mathbf{Q}^*\|_\infty + \frac{8A^2 B(\psi)L_2^2 \alpha_1 \eta}{\gamma_1 + \lambda_1},
\end{aligned}
\tag{86}
$$

*where $B(\psi) := \max_{\boldsymbol{z} \in \mathcal{Z}} \left| \sum_{i=1}^d \psi_i(\boldsymbol{z}) \right|$ and $\bar{\boldsymbol{z}}_T := \sum_{t=1}^T \alpha_t \boldsymbol{z}^t$.*

*Proof.* Recalling the definition of GQCC, we derive that

$$\max_{\boldsymbol{y}' \in \mathcal{Y}} f(\boldsymbol{x}, \boldsymbol{y}') - \min_{\boldsymbol{x}' \in \mathcal{X}} f(\boldsymbol{x}', \boldsymbol{y}) \leq B(\psi) \max_{i \in [1:d]} \left[ \max_{\boldsymbol{w}_i \in \mathcal{Y}_i} f_i(\mathbf{Q}^*, \boldsymbol{x}_i, \boldsymbol{w}_i) - \min_{\boldsymbol{u}_i \in \mathcal{X}_i} f_i(\mathbf{Q}^*, \boldsymbol{u}_i, \boldsymbol{y}_i) \right], \quad (87)$$

for any $\boldsymbol{z} = (\boldsymbol{x}, \boldsymbol{y}), \boldsymbol{v} = (\boldsymbol{u}, \boldsymbol{w}) \in \mathcal{Z}$ and

$$f_i(\mathbf{Q}^*, (\bar{\boldsymbol{x}}_T)_i, \boldsymbol{w}_i) - f_i(\mathbf{Q}^*, \boldsymbol{u}_i, (\bar{\boldsymbol{y}}_T)_i) \leq \sum_{t=1}^T \alpha_t \left[ f_i(\mathbf{Q}^*, \boldsymbol{x}_i^t, \boldsymbol{w}_i) - f_i(\mathbf{Q}^*, \boldsymbol{u}_i, \boldsymbol{y}_i^t) \right] \qquad (88)$$

$$\leq \sum_{t=1}^T \alpha_t \left\langle \boldsymbol{F}_i(\mathbf{Q}^*, \boldsymbol{z}_i^t), \boldsymbol{z}_i^t - \boldsymbol{v}_i \right\rangle$$

$$\leq \sum_{t=1}^T \alpha_t \left\langle \boldsymbol{F}_i(\mathbf{Q}^t, \boldsymbol{z}_i^t), \boldsymbol{z}_i^t - \boldsymbol{v}_i \right\rangle + 2AL_1 \sum_{t=1}^T \alpha_t \| \mathbf{Q}^t - \mathbf{Q}^* \|_\infty,$$

for any $\boldsymbol{v}_i \in \mathcal{Z}_i$. Using the optimality condition, we obtain

$$\left\langle \boldsymbol{F}_i(\mathbf{Q}^{t-1}, \boldsymbol{z}_i^{t-1}), \boldsymbol{z}_i^t - \boldsymbol{g}_i^t \right\rangle \leq \frac{\gamma_t}{\eta} \left( V\left( (\boldsymbol{g}_i^{\boldsymbol{x}})^t, (\boldsymbol{g}_i^{\boldsymbol{x}})^{t-1} \right) + V\left( (\boldsymbol{g}_i^{\boldsymbol{y}})^t, (\boldsymbol{g}_i^{\boldsymbol{y}})^{t-1} \right) \right)$$

$$- \frac{\gamma_t}{\eta} \left( V\left( \boldsymbol{x}_i^t, (\boldsymbol{g}_i^{\boldsymbol{x}})^{t-1} \right) + V\left( \boldsymbol{y}_i^t, (\boldsymbol{g}_i^{\boldsymbol{y}})^{t-1} \right) \right)$$

$$- \frac{\gamma_t + \lambda_t}{\eta} \left( V\left( (\boldsymbol{g}_i^{\boldsymbol{x}})^t, \boldsymbol{x}_i^t \right) + V\left( (\boldsymbol{g}_i^{\boldsymbol{y}})^t, \boldsymbol{y}_i^t \right) \right)$$

$$+ \frac{\lambda_t}{\eta} \left( v\left( (\boldsymbol{g}_i^{\boldsymbol{x}})^t \right) + v\left( (\boldsymbol{g}_i^{\boldsymbol{y}})^t \right) - v\left( \boldsymbol{x}_i^t \right) - v\left( \boldsymbol{y}_i^t \right) \right), \qquad (89)$$

$$\left\langle \boldsymbol{F}_i(\mathbf{Q}^t, \boldsymbol{z}_i^t), \boldsymbol{g}_i^t - \boldsymbol{v}_i \right\rangle \leq \frac{\gamma_t}{\eta} \left( V\left( \boldsymbol{u}_i, (\boldsymbol{g}_i^{\boldsymbol{x}})^{t-1} \right) + V\left( \boldsymbol{w}_i, (\boldsymbol{g}_i^{\boldsymbol{y}})^{t-1} \right) \right)$$

$$- \frac{\gamma_t}{\eta} \left( V\left( (\boldsymbol{g}_i^{\boldsymbol{x}})^t, (\boldsymbol{g}_i^{\boldsymbol{x}})^{t-1} \right) + V\left( (\boldsymbol{g}_i^{\boldsymbol{y}})^t, (\boldsymbol{g}_i^{\boldsymbol{y}})^{t-1} \right) \right)$$

$$- \frac{\gamma_t + \lambda_t}{\eta} \left( V\left( \boldsymbol{u}_i, (\boldsymbol{g}_i^{\boldsymbol{x}})^t \right) + V\left( \boldsymbol{w}_i, (\boldsymbol{g}_i^{\boldsymbol{y}})^t \right) \right)$$

$$+ \frac{\lambda_t}{\eta} \left( v(\boldsymbol{u}_i) + v(\boldsymbol{w}_i) - v\left( (\boldsymbol{g}_i^{\boldsymbol{x}})^t \right) - v\left( (\boldsymbol{g}_i^{\boldsymbol{y}})^t \right) \right). \qquad (90)$$

For each $t \in [1 : T]$, we can apply Eq. (89) and Eq. (90) to the following equation

$$\alpha_t \left\langle \boldsymbol{F}_i(\mathbf{Q}^t, \boldsymbol{z}_i^t), \boldsymbol{z}_i^t - \boldsymbol{v}_i \right\rangle = \alpha_t \left[ \left\langle \boldsymbol{F}_i(\mathbf{Q}^t, \boldsymbol{z}_i^t), \boldsymbol{g}_i^t - \boldsymbol{v}_i \right\rangle + \left\langle \boldsymbol{F}_i(\mathbf{Q}^{t-1}, \boldsymbol{z}_i^{t-1}), \boldsymbol{z}_i^t - \boldsymbol{g}_i^t \right\rangle \right.$$

$$+ \left\langle \boldsymbol{F}_i(\mathbf{Q}^t, \boldsymbol{z}_i^t) - \boldsymbol{F}_i(\mathbf{Q}^{t-1}, \boldsymbol{z}_i^{t-1}), \boldsymbol{z}_i^t - \boldsymbol{g}_i^t \right\rangle \Big],$$

$$\leq \alpha_t \left[ \frac{\gamma_t}{\eta} \left( V\left( \boldsymbol{u}_i, (\boldsymbol{g}_i^{\boldsymbol{x}})^{t-1} \right) + V\left( \boldsymbol{w}_i, (\boldsymbol{g}_i^{\boldsymbol{y}})^{t-1} \right) \right) - \frac{\gamma_t + \lambda_t}{\eta} \left( V\left( \boldsymbol{u}_i, (\boldsymbol{g}_i^{\boldsymbol{x}})^t \right) \right. \right.$$

$$+ V\left( \boldsymbol{w}_i, (\boldsymbol{g}_i^{\boldsymbol{y}})^t \right) \Big) - \frac{\gamma_t}{\eta} \left( V\left( \boldsymbol{x}_i^t, (\boldsymbol{g}_i^{\boldsymbol{x}})^{t-1} \right) + V\left( \boldsymbol{y}_i^t, (\boldsymbol{g}_i^{\boldsymbol{y}})^{t-1} \right) \right)$$

$$\left. - \frac{\gamma_t + \lambda_t}{\eta} \left( V\left( (\boldsymbol{g}_i^{\boldsymbol{x}})^t, \boldsymbol{x}_i^t \right) + V\left( (\boldsymbol{g}_i^{\boldsymbol{y}})^t, \boldsymbol{y}_i^t \right) \right) \right] + \frac{\alpha_t \lambda_t}{\eta} \left( v(\boldsymbol{u}_i) + v(\boldsymbol{w}_i) \right.$$

$$\left. - v\left( (\boldsymbol{g}_i^{\boldsymbol{x}})^t \right) - v\left( (\boldsymbol{g}_i^{\boldsymbol{y}})^t \right) \right) + \alpha_t \left\langle \boldsymbol{F}_i(\mathbf{Q}^t, \boldsymbol{z}_i^t) - \boldsymbol{F}_i(\mathbf{Q}^{t-1}, \boldsymbol{z}_i^{t-1}), \boldsymbol{z}_i^t - \boldsymbol{g}_i^t \right\rangle.$$
$$(91)$$

Therefore, by summing Eq.(91) from $t = 1$ to $t = T$ and utilizing Eq. (84), we have

$$\sum_{t=1}^{T} \alpha_t \left\langle \boldsymbol{F}_i(\mathbf{Q}^t, \boldsymbol{z}_i^t), \boldsymbol{z}_i^t - \boldsymbol{v}_i \right\rangle \leq \frac{\alpha_1 \gamma_1}{\eta} \left( V \left( \boldsymbol{u}_i, (\boldsymbol{g}_i^{\boldsymbol{x}})^0 \right) + V \left( \boldsymbol{w}_i, (\boldsymbol{g}_i^{\boldsymbol{y}})^0 \right) \right)$$

$$+ \frac{2 \sum_{t=1}^{T} \alpha_t \lambda_t}{\eta} \max_{\boldsymbol{z}_i \in \mathcal{Z}_i} (v(\boldsymbol{x}_i) + v(\boldsymbol{y}_i))$$

$$- \frac{1}{\eta} \sum_{t=1}^{T} \alpha_t \left[ \gamma_t \| \boldsymbol{z}_i^t - \boldsymbol{g}_i^{t-1} \|^2 + \frac{\gamma_t + \lambda_t}{2} \| \boldsymbol{g}_i^t - \boldsymbol{z}_i^t \|^2 \right]$$

$$+ \eta \sum_{t=1}^{T} \left[ \frac{\alpha_t}{\gamma_t + \lambda_t} \left\| \boldsymbol{F}_i(\mathbf{Q}^t, \boldsymbol{z}_i^t) - \boldsymbol{F}_i(\mathbf{Q}^{t-1}, \boldsymbol{z}_i^{t-1}) \right\|_*^2 \right]. \quad (92)$$

According to the Lipschitz continuity of $\boldsymbol{F}_i$, we derive that

$$\eta \sum_{t=1}^{T} \left[ \frac{\alpha_t}{\gamma_t + \lambda_t} \left\| \boldsymbol{F}_i(\mathbf{Q}^t, \boldsymbol{z}_i^t) - \boldsymbol{F}_i(\mathbf{Q}^{t-1}, \boldsymbol{z}_i^{t-1}) \right\|_*^2 \right]$$

$$\leq 2\eta L_2^2 \sum_{t=1}^{T} \left[ \frac{\alpha_t}{\gamma_t + \lambda_t} \| \boldsymbol{z}_i^t - \boldsymbol{z}_i^{t-1} \|^2 \right] + 2\eta L_1^2 \sum_{t=1}^{T} \left[ \frac{\alpha_t}{\gamma_t + \lambda_t} \| \mathbf{Q}^t - \mathbf{Q}^{t-1} \|_\infty^2 \right]. \quad (93)$$

It follows from parameter setting Eq. (84) and Cauchy-Schwarz inequality that

$$\frac{\alpha_t \gamma_t}{2} \| \boldsymbol{z}_i^t - \boldsymbol{g}_i^{t-1} \|^2 + \frac{\alpha_{t-1}(\gamma_{t-1} + \lambda_{t-1})}{2} \| \boldsymbol{g}_i^{t-1} - \boldsymbol{z}_i^{t-1} \|^2 \geq \frac{\alpha_t \gamma_t}{4} \| \boldsymbol{z}_i^t - \boldsymbol{z}_i^{t-1} \|^2. \quad (94)$$

Combining Eq. (85) and Eq. (94), we may therefore obtain

$$- \frac{1}{\eta} \sum_{t=1}^{T} \alpha_t \left[ \frac{\gamma_t}{2} \| \boldsymbol{z}_i^t - \boldsymbol{g}_i^{t-1} \|^2 + \frac{\gamma_t + \lambda_t}{4} \| \boldsymbol{g}_i^t - \boldsymbol{z}_i^t \|^2 \right] + 2\eta L_2^2 \sum_{t=1}^{T} \left[ \frac{\alpha_t}{\gamma_t + \lambda_t} \| \boldsymbol{z}_i^t - \boldsymbol{z}_i^{t-1} \|^2 \right]$$

$$\leq \frac{2\eta L_2^2 \alpha_1}{\gamma_1 + \lambda_1} \| \boldsymbol{z}_i^1 - \boldsymbol{z}_i^0 \|^2. \quad (95)$$

Applying Eq. (93) and Eq. (95) to Eq. (92) and utilizing Eq. (87), Eq. (88), we complete the proof. $\quad \square$

### C.2.2 Part II: Estimation of Approximation Error $\|\mathbf{Q}^t - \mathbf{Q}^*\|$

According to the iterately update of $\mathbf{Q}^t$, we can derive the upper bound of weighted average of $\|\mathbf{Q}^t - \mathbf{Q}^{t-1}\|_\infty^2$. Next, we aim to bound $\|\mathbf{Q}^t - \mathbf{Q}^*\|$ for each iteration $t$. In this section, we select the following parameter settings:

$$c = 2(1 - \theta)^{-1}, \, \eta \leq \frac{(1 - \theta)^{1/2}}{8(\gamma A L_1)^{1/2} L_2}, \, \beta_t = \frac{c}{c + t}, \, \alpha_t = \beta_{T,t}, \, \gamma_t = \frac{\alpha_{t-1}}{\alpha_t}, \, \lambda_t = 1 - \gamma_t. \quad (96)$$

**Lemma C.9.** *Consider the settings:* $\gamma_t = \frac{\alpha_{t-1}}{\alpha_t} \leq 1$, $\lambda_t = 1 - \gamma_t$, *and* $\eta \leq \frac{(1-\theta)^{1/2}}{8(\gamma A L_1 L_2)^{1/2}}$. *Then, we obtain the estimation of* $\|\mathbf{Q}^t - \mathbf{Q}^*\|$ *as follows,*

$$\|\mathbf{Q}^t - \mathbf{Q}^*\|_\infty \leq \sum_{j=2}^{t} \beta_{t,j}^{(1+\theta)/2} H_j, \quad (97)$$

*for any* $t \geq 2$ *where*

$$H_j := \gamma D \left( \frac{1}{\eta} + 16 \eta L_2 \right) \left( \beta_{j-1,1} \gamma_1 + 2 \sum_{\kappa=1}^{j-1} \beta_{j-1,\kappa} \lambda_\kappa \right)$$

$$+ 2 \eta \gamma L_1^2 (1 + 64 \eta^2 L_2^2 C^2) \sum_{\kappa=1}^{j-1} \beta_{j-1,\kappa} \beta_{\kappa-1}^2 + 128 \eta^3 \gamma A^2 L_2^4 \beta_{j-1,1}. \quad (98)$$

*Proof.* According to the fact that $\mathbf{Q}^*$ is a fixed point of function $\boldsymbol{P}$, we have

$$
\begin{aligned}
\mathbf{Q}^t - \mathbf{Q}^* &= \sum_{\kappa=1}^{t-1} \beta_{t-1,\kappa}[\boldsymbol{P}(\mathbf{Q}^\kappa, \boldsymbol{z}^\kappa) - \boldsymbol{P}(\mathbf{Q}^*, \boldsymbol{z}^*)] \\
&\underset{(a)}{\leq} \sum_{\kappa=1}^{t-1} \beta_{t-1,\kappa} \left\{ [\boldsymbol{P}(\mathbf{Q}^\kappa, \boldsymbol{x}^\kappa, \boldsymbol{y}^\kappa) - \boldsymbol{P}(\mathbf{Q}^\kappa, \boldsymbol{x}^*, \boldsymbol{y}^\kappa)] + [\boldsymbol{P}(\mathbf{Q}^\kappa, \boldsymbol{x}^*, \boldsymbol{y}^\kappa) - \boldsymbol{P}(\mathbf{Q}^*, \boldsymbol{x}^*, \boldsymbol{y}^\kappa)] \right\} \\
&\leq \sum_{i=1}^{d} \left( \sum_{\kappa=1}^{t-1} \beta_{t-1,\kappa} \left\langle \boldsymbol{F}_i^{\boldsymbol{x}}(\mathbf{Q}^\kappa, \boldsymbol{z}_i^\kappa), \boldsymbol{x}_i^\kappa - \boldsymbol{x}_i^* \right\rangle \right) \mathbf{C}_i + \theta \left( \sum_{\kappa=1}^{t-1} \beta_{t-1,\kappa} \|\mathbf{Q}^\kappa - \mathbf{Q}^*\|_\infty \right) \boldsymbol{e}_d \boldsymbol{e}_d^\top .
\end{aligned}
$$
(99)

Where (a) is derived from the maximizer's property of $\boldsymbol{y}^*$ for matrix-valued function $\boldsymbol{P}(\mathbf{Q}^*, \boldsymbol{x}^*, \cdot)$. Similarly, we can obtain

$$
\mathbf{Q}^t - \mathbf{Q}^* \geq -\sum_{i=1}^{d} \left( \sum_{\kappa=1}^{t-1} \beta_{t-1,\kappa} \left\langle \boldsymbol{F}_i^{\boldsymbol{y}}(\mathbf{Q}^\kappa, \boldsymbol{z}_i^\kappa), \boldsymbol{y}_i^\kappa - \boldsymbol{y}_i^* \right\rangle \right) \mathbf{B}_i - \theta \left( \sum_{\kappa=1}^{t-1} \beta_{t-1,\kappa} \|\mathbf{Q}^\kappa - \mathbf{Q}^*\|_\infty \right) \boldsymbol{e}_d \boldsymbol{e}_d^\top .
$$
(100)

Hence, we derive

$$
\begin{aligned}
\|\mathbf{Q}^t - \mathbf{Q}^*\|_\infty \leq & \gamma \max_{i \in [1:d]} \left\{ \frac{\beta_{t-1,1}\gamma_1}{\eta} \max_{\boldsymbol{z}_i \in \mathcal{Z}_i} \left( V\left(\boldsymbol{x}_i, (\boldsymbol{g}_i^{\boldsymbol{x}})^0\right) + V\left(\boldsymbol{y}_i, (\boldsymbol{g}_i^{\boldsymbol{y}})^0\right) \right) \right. \\
& + \frac{2 \sum_{\kappa=1}^{t-1} \beta_{t-1,\kappa} \lambda_\kappa}{\eta} \max_{\boldsymbol{z}_i \in \mathcal{Z}_i} \left( v(\boldsymbol{x}_i) + v(\boldsymbol{y}_i) \right) \\
& \left. + 2\eta L_2^2 \sum_{\kappa=1}^{t-1} \frac{\beta_{t-1,\kappa}}{\gamma_\kappa + \lambda_\kappa} \left\| \boldsymbol{z}_i^\kappa - \boldsymbol{z}_i^{\kappa-1} \right\|^2 \right\} \\
& + 2\eta \gamma L_1^2 \sum_{\kappa=1}^{t-1} \frac{\beta_{t-1,\kappa}}{\gamma_\kappa + \lambda_\kappa} \|\mathbf{Q}^\kappa - \mathbf{Q}^{\kappa-1}\|_\infty^2 + \theta \left( \sum_{\kappa=1}^{t-1} \beta_{t-1,\kappa} \|\mathbf{Q}^\kappa - \mathbf{Q}^*\|_\infty \right),
\end{aligned}
$$
(101)

by combining $\beta_{t-1,\kappa} \prod_{j=t}^{T}(1 - \beta_j) = \alpha_\kappa$ and Eq. (99), and using the proof technique of Theorem C.8. Next, for any $i \in [1 : d]$, we can obtain an upper bound estimation of $\max_{\boldsymbol{v}_i \in \mathcal{Z}_i} \sum_{\kappa=1}^{t-1} \beta_{t-1,\kappa} \langle \boldsymbol{F}_i(\mathbf{Q}^\kappa, \boldsymbol{z}_i^\kappa), \boldsymbol{z}_i^\kappa - \boldsymbol{v}_i \rangle$ as follows

$$
\begin{aligned}
\max_{\boldsymbol{v}_i \in \mathcal{Z}_i} \sum_{\kappa=1}^{t-1} \beta_{t-1,\kappa} \left\langle \boldsymbol{F}_i(\mathbf{Q}^\kappa, \boldsymbol{z}_i^\kappa), \boldsymbol{z}_i^\kappa - \boldsymbol{v}_i \right\rangle \leq & \frac{D}{\eta} \left( \beta_{t-1,1}\gamma_1 + 2 \sum_{\kappa=1}^{t-1} \beta_{t-1,\kappa}\lambda_\kappa \right) + 8\eta A^2 L_2^2 \beta_{t-1,1} \\
& + 8\eta L_1^2 C^2 \sum_{\kappa=1}^{t-1} \beta_{t-1,\kappa}\beta_{\kappa-1}^2 - \frac{1}{8\eta} \sum_{\kappa=1}^{t-1} \beta_{t-1,\kappa} \left\| z_\kappa^k - z_{\kappa-1}^k \right\|^2 .
\end{aligned}
$$
(102)

Furthermore, we also have a lower bound estimation of it

$$
\begin{aligned}
\max_{\boldsymbol{v}_i \in \mathcal{Z}_i} \sum_{\kappa=1}^{t-1} \beta_{t-1,\kappa} \left\langle \boldsymbol{F}_i(\mathbf{Q}^\kappa, \boldsymbol{z}_i^\kappa), \boldsymbol{z}_i^\kappa - \boldsymbol{v}_i \right\rangle \geq & \max_{\boldsymbol{v}_i \in \mathcal{Z}_i} \sum_{\kappa=1}^{t-1} \beta_{t-1,\kappa} \left\langle \boldsymbol{F}_i(\mathbf{Q}^\kappa, \boldsymbol{v}_i), \boldsymbol{z}_i^\kappa - \boldsymbol{v}_i \right\rangle \\
& \geq \max_{\boldsymbol{v}_i \in \mathcal{Z}_i} \sum_{\kappa=1}^{t-1} \beta_{t-1,\kappa} \left\langle \boldsymbol{F}_i(\mathbf{Q}^*, \boldsymbol{v}_i), \boldsymbol{z}_i^\kappa - \boldsymbol{v}_i \right\rangle \\
& \quad - 2AL_1 \sum_{\kappa=1}^{t-1} \beta_{t-1,\kappa} \|\mathbf{Q}^\kappa - \mathbf{Q}^*\|_\infty \\
& \geq -2AL_1 \sum_{\kappa=1}^{t-1} \beta_{t-1,\kappa} \|\mathbf{Q}^\kappa - \mathbf{Q}^*\|_\infty .
\end{aligned}
$$
(103)

Therefore, combining Eq. (102) and Eq. (103), we derive the following result

$$
\begin{aligned}
\sum_{\kappa=1}^{t-1} \beta_{t-1,\kappa} \left\| z_i^\kappa - z_i^{\kappa-1} \right\|^2 \leq & 16\eta A L_1 \sum_{\kappa=1}^{t-1} \beta_{t-1,\kappa} \|\mathbf{Q}^\kappa - \mathbf{Q}^*\|_\infty + 64\eta^2 A^2 L_2^2 \beta_{t-1,1} \\
& + 8D \left( \beta_{t-1,1}\gamma_1 + 2 \sum_{\kappa=1}^{t-1} \beta_{t-1,\kappa}\lambda_\kappa \right) + 64\eta^2 C^2 L_1^2 \sum_{\kappa=1}^{t-1} \beta_{t-1,\kappa}\beta_{\kappa-1}^2,
\end{aligned}
\tag{104}
$$

for any $i \in [1:d]$.

$$
\begin{aligned}
\|\mathbf{Q}^t - \mathbf{Q}^*\|_\infty \leq & \gamma D \left( \frac{1}{\eta} + 16\eta L_2^2 \right) \left( \beta_{t-1,1}\gamma_1 + 2 \sum_{\kappa=1}^{t-1} \beta_{t-1,\kappa}\lambda_\kappa \right) + 128\eta^3 \gamma A^2 L_2^4 \beta_{t-1,1} \\
& + 2\eta\gamma L_1^2 \left( 1 + 64\eta^2 C^2 L_2^2 \right) \sum_{\kappa=1}^{t-1} \beta_{t-1,\kappa}\beta_{\kappa-1}^2 \\
& + \left( 32\eta^2 \gamma A L_1 L_2^2 + \theta \right) \sum_{\kappa=1}^{t-1} \beta_{t-1,\kappa} \|\mathbf{Q}^\kappa - \mathbf{Q}^*\|_\infty.
\end{aligned}
\tag{105}
$$

Finally, by applying [67, Lemma 33] to Eq. (105), we complete the proof. $\qquad\square$

Under parameter settings Eq. (96), the following auxiliary Lemma C.10 provides both lower bound and upper bound of $\beta_{T,t}$.

**Lemma C.10.** *Assuming that $\beta_t = \frac{c'}{c+t}$ and $c \geq c'$, we can obtain the following result:*

$$
\exp\left\{ -\frac{(c'+c'c)^2}{2c} \right\} \frac{(c+t)^{c'-1}}{(c+T)^{c'}} \leq \beta_{T,t} \leq \frac{(1+c)(c+t+1)^{c'-1}}{(c+T+1)^{c'}},
\tag{106}
$$

*for any $T \geq t \geq 1$.*

*Proof.* Recalling that

$$
\beta_{T,t} = \frac{c'}{c+t} \prod_{k=t+1}^{T} \left( 1 - \frac{c'}{c+k} \right) = \frac{c'}{c+t} \exp\left\{ \sum_{k=t+1}^{T} \log\left( 1 - \frac{c'}{c+k} \right) \right\},
\tag{107}
$$

we have

$$
\beta_{T,t} \leq \frac{c'}{c+t} \left( \frac{c+t+1}{c+T+1} \right)^{c'} \leq \frac{(1+c)(c+t+1)^{c'-1}}{(c+T+1)^{c'}},
\tag{108}
$$

and

$$
\beta_{T,t} \geq \exp\left\{ -\frac{(c'+c'c)^2}{2c} \right\} \frac{(c+t)^{c'-1}}{(c+T)^{c'}},
\tag{109}
$$

by combining the result of Lemma D.9. $\qquad\square$

**Corollary C.11.** *Assuming that $\beta_t = \frac{c'}{c+t}$, $c \geq 1$ and $c'(1-\theta) \geq 1$, we can obtain*

$$
\beta_{T,t}^\theta \leq \frac{c'}{c+T},
\tag{110}
$$

*for any $T \geq t \geq 1$.*

By utilizing the result of Lemma C.10, we notice that

$$
\begin{aligned}
\sum_{\kappa=1}^{j-1} \beta_{j-1,\kappa}\lambda_\kappa \leq & \sum_{\kappa=1}^{j-1} \frac{(1+c)^2(c+\kappa+1)^{c-2}}{(c+j)^c} \\
\underset{(a)}{\leq} & \frac{(1+c)^2}{(c+j)^c} \int_1^{j-1} (c+x+1)^{c-2}\mathrm{d}x + \frac{(1+c)^2}{(c+j)^2} \\
\leq & \frac{(1+c)^2}{(c-1)(c+j)} + \frac{(1+c)^2}{(c+j)^2},
\end{aligned}
\tag{111}
$$

and

$$\sum_{\kappa=1}^{j-1} \beta_{j-1,\kappa}\beta_{\kappa-1}^2 \leq \sum_{\kappa=1}^{j-1} \frac{(1+c)^4(c+\kappa+1)^{c-3}}{c(c+j)^c}$$

$$\underset{\text{(b)}}{\leq} \frac{(1+c)^3}{c(c+j)^c}\int_1^{j-1}(c+x+1)^{c-2}\mathrm{d}x + \frac{(1+c)^4}{c(c+j)^3}$$

$$\leq \frac{(1+c)^3}{c(c-1)(c+j)} + \frac{(1+c)^4}{c(c+j)^3}, \tag{112}$$

where (a) and (b) are derived from the fact that $\sum_{\kappa=1}^{j-2}(c+\kappa+1)^{c-2} \leq \int_1^{j-1}(c+x+1)^{c-2}\mathrm{d}x$.
Next, we have

$$H_j \leq \gamma D\left(\frac{1}{\eta} + 16\eta L_2\right)\left(\frac{2(c+2)^{c-1}}{(c+j)^c} + \frac{2(1+c)^2}{(c-1)(c+j)} + \frac{2(1+c)^2}{(c+j)^2}\right)$$

$$+ 2\eta\gamma L_1^2(1 + 64\eta^2 L_2^2 C^2)\left(\frac{(1+c)^3}{c(c-1)(c+j)} + \frac{(1+c)^4}{c(c+j)^3}\right)$$

$$+ 128\eta^3\gamma A^2 L_2^4\frac{(c+2)^c}{(c+j)^c}, \tag{113}$$

and

$$\sum_{j=2}^t \beta_{t,j}^{(1+\theta)/2}H_j \underset{\text{(c)}}{\leq}\frac{c}{c+t}\left\{\gamma D\left(\frac{1}{\eta} + 16\eta L_2\right)\left[\int_2^t\frac{2(c+2)^{c-1}}{(c+x)^c}\mathrm{d}x\right.\right.$$

$$+ \int_1^t\left(\frac{2(1+c)^2}{(c-1)(c+x)} + \frac{2(1+c)^2}{(c+x)^2}\right)\mathrm{d}x + 1\Big]$$

$$+ 2\eta\gamma L_1^2(1 + 64\eta^2 L_2^2 C^2)\int_1^t\left(\frac{(1+c)^3}{c(c-1)(c+x)} + \frac{(1+c)^4}{c(c+x)^3}\right)\mathrm{d}x\Bigg\}$$

$$+ 128\eta^3\gamma A^2 L_2^4\left[1 + \int_2^t\frac{(c+2)^c}{(c+x)^c}\mathrm{d}x\right]$$

$$\leq \frac{c}{c+t}\left\{\gamma D\left(\frac{1}{\eta} + 16\eta L_2\right)\left[\frac{2(c+1)^2}{c-1}\log(c+t) + 5 + 2c\right] + 640\eta^3\gamma A^2 L_2^4\right.$$

$$+ 2\eta\gamma L_1^2(1 + 64\eta^2 L_2^2 C^2)\left[\frac{2(c+1)^2}{c-1}\log(c+t) + \frac{(c+1)^2}{2c}\right]\Bigg\}, \tag{114}$$

where (c) follows from Corollary C.11. For simplicity, we denote

$$Y_T^\eta := 8(c+1)\left[\gamma D\left(\frac{1}{\eta} + 16\eta L_2\right) + 40\eta^3\gamma A^2 L_2^4 + 2\eta\gamma L_1^2(1 + 64\eta^2 L_2^2 C^2)\right](\log(c+T) + 1). \tag{115}$$

Therefore, it follows from Eq. (114) and Lemma C.9 that

$$\|\mathbf{Q}^t - \mathbf{Q}^*\|_\infty \leq \sum_{j=2}^t \beta_{t,j}^{(1+\theta)/2}H_j \leq \frac{c}{c+t}Y_t^\eta. \tag{116}$$

### C.2.3 The Last Step

According to Eq. (116) and the initial $\mathbf{Q}^0$ satisfies $\|\mathbf{Q}^0\|_\infty \leq C$, we have $\|\mathbf{Q}^*\| \leq C$. We are ready to complete the proof of Theorem C.7.

*Proof of Theorem C.7.* It is noteworthy that the hyper-parameters selected in Eq. (96) satisfies the preconditions of Theorem C.8. Combining the conclusion of Theorem C.8, Eq. (116) and the

estimation of $\alpha_t$ (i.e. $\beta_{T,t}$) in Lemma C.10, we obtain:

$$
\begin{aligned}
\mathcal{G}_f(\boldsymbol{x}, \boldsymbol{y}) \leq & B \max_{i \in [1:d]} \left\{ \frac{\alpha_1 \gamma_1}{\eta} \max_{\boldsymbol{z}_i \in \mathcal{Z}_i} \left( V\left(\boldsymbol{x}_i, (\boldsymbol{g}_i^{\boldsymbol{x}})^0\right) + V\left(\boldsymbol{y}_i, (\boldsymbol{g}_i^{\boldsymbol{y}})^0\right) \right) \right. \\
& \left. + \frac{2\sum_{t=1}^T \alpha_t \lambda_t}{\eta} \max_{\boldsymbol{z}_i \in \mathcal{Z}_i} \left( v(\boldsymbol{x}_i) + v(\boldsymbol{y}_i) \right) \right\} + 2\eta B L_1^2 \sum_{t=1}^T \alpha_t \beta_{t-1}^2 \\
& + 2ABcL_1 Y_T^\eta \sum_{t=1}^T \frac{\alpha_t}{c+t} + 8\eta A^2 B L_2^2 \alpha_1 \\
\underset{\mathrm{a}}{\leq} & \frac{2BD}{\eta} \left( \frac{(c+2)^{c-1}}{(c+T+1)^c} + \frac{c(c+2)}{(c+T+1)^2} + \frac{2(c+1)}{c+T+1} \right) \\
& + 2\eta B L_1^2 \left( \frac{(c+1)^3}{c(c-1)(c+T+1)} + \frac{(c+1)^4}{c(c+T+1)^3} \right) \\
& + 2AB L_1 Y_T^\eta \left( \frac{4c}{c+T+1} + \frac{c(c+2)}{(c+T+1)^2} \right) + 8\eta A^2 B L_2^2 \frac{(1+c)(c+2)^{c-1}}{(c+T+1)^c} \\
\leq & 2B \left( \frac{3D}{\eta} + 10\eta L_1^2 + 5AL_1 Y_T^\eta + 4\eta A^2 L_2^2 \right) \frac{c+1}{c+T+1},
\end{aligned}
\tag{117}
$$

when $T \geq 1$, where $B$ denotes $\max_{\boldsymbol{z} \in \mathcal{Z}} \sum_{i=1}^d \psi_i(\boldsymbol{z})$ and (a) is derived from parameter settings Eq. (96) and the result of Lemma C.10. $\qquad\square$

## C.3  Application to Minimax Problems

### C.3.1  Infinite Horizon Two-Player Zero-Sum Markov Games

To simplify notations, in the following discussion, we write $\mathcal{S} = \{s_i\}_{i=1}^{|\mathcal{S}|}$, and denote by $\mathbf{Q}^{\boldsymbol{z}} = \left( \mathbf{Q}^{\boldsymbol{z}}(s_1, \cdot, \cdot), \cdots, \mathbf{Q}^{\boldsymbol{z}}(s_{|\mathcal{S}|}, \cdot, \cdot) \right)$ the joint action-value matrix where $\mathbf{Q}^{\boldsymbol{z}}(s_i, \cdot, \cdot) \in \mathbb{R}^{|\mathcal{A}| \times |\mathcal{B}|}$ is an action-value matrix on state $s_i$. According to the connection between value function and action-value function

$$
V^{\boldsymbol{z}}(s_i) = \mathbb{E}_{\substack{a \sim \boldsymbol{x}(\cdot|s_i) \\ b \sim \boldsymbol{y}(\cdot|s_i)}} [Q^{\boldsymbol{z}}(s_i, a, b)], \ Q^{\boldsymbol{z}}(s_i, a, b) = (1-\theta)\sigma(s_i, a, b) + \theta \mathbb{E}_{s_{i'} \sim \mathbb{P}(\cdot|s_i, a, b)} [V^{\boldsymbol{z}}(s_{i'})],
$$

we provide the following proof for Proposition 4.5.

*Proof of Proposition 4.5.* By defining

$$
[\mathbf{P}_i(\mathbf{Q}, \boldsymbol{z})]_{a,b} := (1-\theta)\sigma(s_i, a, b) + \theta \mathbb{E}_{s_{i'} \sim \mathbb{P}(\cdot|s_i, a, b)} \left[ \langle \mathbf{Q}_{i'} \boldsymbol{y}_{i'}, \boldsymbol{x}_{i'} \rangle \right],
\tag{118}
$$

we derive that $\mathbf{Q}^{\boldsymbol{z}} = \mathbf{P}(\mathbf{Q}^{\boldsymbol{z}}, \boldsymbol{z})$. We can notice that

$$
\begin{aligned}
[\mathbf{P}_i(\mathbf{Q}, \boldsymbol{x}, \boldsymbol{y}) - \mathbf{P}_i(\mathbf{Q}, \boldsymbol{x}', \boldsymbol{y})]_{a,b} &= \theta \mathbb{E}_{s_{i'} \sim \mathbb{P}(\cdot|s_i, a, b)} \left[ \langle \mathbf{Q}_{i'} \boldsymbol{y}_{i'}, \boldsymbol{x}_{i'} - (\boldsymbol{x}')_{i'} \rangle \right], \\
[\mathbf{P}_i(\mathbf{Q}, \boldsymbol{x}, \boldsymbol{y}') - \mathbf{P}_i(\mathbf{Q}, \boldsymbol{x}, \boldsymbol{y})]_{a,b} &= \theta \mathbb{E}_{s_{i'} \sim \mathbb{P}(\cdot|s_i, a, b)} \left[ \langle -\mathbf{Q}_{i'}^\top \boldsymbol{x}_{i'}, \boldsymbol{y}_{i'} - (\boldsymbol{y}')_{i'} \rangle \right].
\end{aligned}
\tag{119}
$$

Therefore, for any $\mathbf{Q}$ satisfies $\|\mathbf{Q}\|_\infty \leq 1$, it's easy to verify that

1. $\boldsymbol{F}_i(\cdot, \boldsymbol{z}_i)$ is uniformly 2-Lipschitz continuous with respect to $\|\cdot\|_\infty$ under $\|\cdot\|_\infty$ for any $\boldsymbol{z}_i \in \mathcal{Z}_i$, and $\boldsymbol{F}_i(\mathbf{Q}, \cdot)$ is uniformly 1-Lipschitz continuous with respect to $\|\cdot\|_\infty$ under $\|\cdot\|_1$, since $\boldsymbol{F}(\mathbf{Q}, \boldsymbol{z}_i) = \left( \boldsymbol{y}_i^\top \mathbf{Q}_i^\top, -\boldsymbol{x}_i^\top \mathbf{Q}_i \right)^\top$,

2. $\mathbf{P}$ satisfies $[\mathbf{A_2}]$ in Assumptions 4.2 with $[\mathbf{B}_i]_{s,a,b} = [\mathbf{C}_i]_{s,a,b} = \theta \mathbb{P}(s_i|s, a, b)$ and $\gamma = 2\theta$, since Eq. (119),

3. $\mathbf{P}(\cdot, \boldsymbol{z})$ is a $\theta$-contraction mapping under $\|\cdot\|_\infty$, and $\|\mathbf{P}(\cdot, \cdot)\|_\infty \leq 1$, since the definition of $\mathbf{P}$,

4. $[\mathbf{P}_i(\mathbf{Q}, \cdot, \cdot)]_{a,b}$ is bi-linear with respect to $\boldsymbol{x}$ and $\boldsymbol{y}$, and $\frac{\min\{[\mathbf{C}_i]_{s,a,b}, [\mathbf{B}_i]_{s,a,b}\}}{[\mathbf{C}_i]_{s,a,b} + [\mathbf{B}_i]_{s,a,b}} \equiv 1/2$ for any $i$ and $s, a, b$.

Therefore, according to Lemma 4.3, there exist a tensor $\mathbf{Q}^*$ and a pair of $(\boldsymbol{x}^*, \boldsymbol{y}^*)$ satisfy Eq. (11). Furthermore, the $(\boldsymbol{x}^*, \boldsymbol{y}^*)$ mentioned above is a Nash equilibrium of $J^{\boldsymbol{x},\boldsymbol{y}}(\boldsymbol{\rho}_0)$ by utilizing Corollary C.6. We may therefore derive that $\mathbf{Q}^* \equiv \mathbf{Q}^{\boldsymbol{z}^*}$. Leveraging Eq. (65) for any Nash equilibrium $(\boldsymbol{x}^*, \boldsymbol{y}^*) \in \mathcal{Z}$ and denoting $\mathbf{Q}_i^* = \mathbf{Q}^{\boldsymbol{x}^*,\boldsymbol{y}^*}(s_i, \cdot, \cdot)$, we have

$$J^{\boldsymbol{x}^*,\boldsymbol{y}^*}(\boldsymbol{\rho}_0) - J^{\boldsymbol{x}^*(\boldsymbol{y}),\boldsymbol{y}}(\boldsymbol{\rho}_0) = \sum_{s \in \mathcal{S}} \boldsymbol{d}_{\boldsymbol{\rho}_0}^{\boldsymbol{x}^*(\boldsymbol{y}),\boldsymbol{y}}(s) \Big[ \langle \mathbf{Q}^{\boldsymbol{x}^*,\boldsymbol{y}^*}(s,\cdot,\cdot)\boldsymbol{y}^*(\cdot|s), \boldsymbol{x}^*(\cdot|s) \rangle \tag{120}$$

$$- \langle \mathbf{Q}^{\boldsymbol{x}^*,\boldsymbol{y}^*}(s,\cdot,\cdot)\boldsymbol{y}(\cdot|s), \boldsymbol{x}^*(\boldsymbol{y})(\cdot|s) \rangle \Big]$$

$$\leq \sum_{i=1}^{|\mathcal{S}|} \boldsymbol{d}_{\boldsymbol{\rho}_0}^{\boldsymbol{x}^*(\boldsymbol{y}),\boldsymbol{y}}(s_i) \Big[ \langle \mathbf{Q}_i^* \boldsymbol{y}_i^*, \boldsymbol{x}_i^* \rangle - \min_{\boldsymbol{u}_i \in \mathcal{X}_i} \langle \mathbf{Q}_i^* \boldsymbol{y}_i, \boldsymbol{u}_i \rangle \Big], \tag{121}$$

where $\boldsymbol{x}^*(\boldsymbol{y}) = \operatorname*{argmin}_{\boldsymbol{u} \in \mathcal{X}} J^{\boldsymbol{u},\boldsymbol{y}}(\boldsymbol{\rho}_0)$ and $\boldsymbol{y}^*(\boldsymbol{x}) = \operatorname*{argmax}_{\boldsymbol{w} \in \mathcal{Y}} J^{\boldsymbol{x},\boldsymbol{w}}(\boldsymbol{\rho}_0)$. Similarly, we have

$$J^{\boldsymbol{x},\boldsymbol{y}^*(\boldsymbol{x})}(\boldsymbol{\rho}_0) - J^{\boldsymbol{x}^*,\boldsymbol{y}^*}(\boldsymbol{\rho}_0) \leq \sum_{i=1}^{|\mathcal{S}|} \boldsymbol{d}_{\boldsymbol{\rho}_0}^{\boldsymbol{x},\boldsymbol{y}^*(\boldsymbol{x})}(s_i) \Big[ \max_{\boldsymbol{w}_i \in \mathcal{Y}_i} \langle (\mathbf{Q}_i^*)^\top \boldsymbol{x}_i, \boldsymbol{w}_i \rangle - \langle (\mathbf{Q}_i^*)^\top \boldsymbol{x}_i^*, \boldsymbol{y}_i^* \rangle \Big]. \tag{122}$$

By setting $\psi_i(\boldsymbol{z}) := \max\{\boldsymbol{d}_{\boldsymbol{\rho}_0}^{\boldsymbol{x},\boldsymbol{y}^*(\boldsymbol{x})}(s_i), \boldsymbol{d}_{\boldsymbol{\rho}_0}^{\boldsymbol{x}^*(\boldsymbol{y}),\boldsymbol{y}}(s_i)\}$ and combining the facts that $f_i(\mathbf{Q}^*, \boldsymbol{x}_i^*, \boldsymbol{y}_i^*) - \min_{\boldsymbol{u}_i \in \mathcal{X}_i} f_i(\mathbf{Q}^*, \boldsymbol{u}_i, \boldsymbol{y}_i) \geq 0$ and $\max_{\boldsymbol{w}_i \in \mathcal{Y}_i} f_i(\mathbf{Q}^*, \boldsymbol{x}_i, \boldsymbol{w}_i) - f_i(\mathbf{Q}^*, \boldsymbol{x}_i^*, \boldsymbol{y}_i^*) \geq 0$ derived from Corollary C.6, we have

$$J^{\boldsymbol{x},\boldsymbol{y}^*(\boldsymbol{x})}(\boldsymbol{\rho}_0) - J^{\boldsymbol{x}^*(\boldsymbol{y}),\boldsymbol{y}}(\boldsymbol{\rho}_0) \leq \sum_{i=1}^{d} \psi_i(\boldsymbol{z}) \Big[ \max_{\boldsymbol{w}_i \in \mathcal{Y}_i} f_i(\mathbf{Q}^*, \boldsymbol{x}_i, \boldsymbol{w}_i) - \min_{\boldsymbol{u}_i \in \mathcal{X}_i} f_i(\mathbf{Q}^*, \boldsymbol{u}_i, \boldsymbol{y}_i) \Big]. \tag{123}$$

$\square$

### C.3.2 Convex-Concave Minimax Problems

In this section, we consider convex-concave minimax problem over compact concave region $\mathcal{Z} = \mathcal{X} \times \mathcal{Y} \subset \mathbb{R}^{\sum_{i=1}^{d} n_i} \times \mathbb{R}^{\sum_{i=1}^{d} m_i}$ which satisfies Assumption C.1 with divergence-generating function $v$. The standard convex-concave minimax problem is formulated as follows:

$$\min_{\boldsymbol{x} \in \mathcal{X}} \max_{\boldsymbol{y} \in \mathcal{Y}} f(\boldsymbol{x}, \boldsymbol{y}), \tag{124}$$

where $f$ is convex with respect to $\boldsymbol{x}$ and concave with respect to $\boldsymbol{y}$. Therefore, we obtain that

$$f(\boldsymbol{x}, \boldsymbol{y}^*(\boldsymbol{x})) - f(\boldsymbol{x}^*(\boldsymbol{y}), \boldsymbol{y}) \leq \langle \nabla_{\boldsymbol{x}} f(\boldsymbol{z}), \boldsymbol{x} - \boldsymbol{x}^*(\boldsymbol{y}) \rangle + \langle -\nabla_{\boldsymbol{y}} f(\boldsymbol{z}), \boldsymbol{y} - \boldsymbol{y}^*(\boldsymbol{x}) \rangle, \tag{125}$$

for any $\boldsymbol{z} = (\boldsymbol{x}, \boldsymbol{y}) \in \mathcal{Z}$. We may therefore derive that $f$ satisfies GQCC condition with $g(\boldsymbol{z}) \equiv 1$ and $f(\mathbf{P}(\boldsymbol{z}), \boldsymbol{z}) = f(\boldsymbol{z})$. Furthermore, assuming $\nabla f$ is $L$-lipschitz continuous (i.e., $\|\nabla f(\boldsymbol{z}) - \nabla f(\boldsymbol{v})\|_* \leq L\|\boldsymbol{z} - \boldsymbol{v}\|$ for any $\boldsymbol{z}, \boldsymbol{v} \in \mathcal{Z}$) and choosing $\mathbf{P} \equiv \mathbf{0}$, then verifying that $f$ satisfies the preconditions of general version of Theorem C.7 is reduced to verifying that $f$ satisfies (1) in Assumption C.3. Since $\boldsymbol{F} = \nabla f$ only depends on variable $\boldsymbol{z}$, it is evident that $f$ satisfies (1) in Assumption C.3 when $\nabla f$ is L-Lipschitz. Therefore, under the smoothness condition of $f$, Theorem C.7 implies that $\mathcal{O}(\varepsilon^{-1})$ iterations Algorithm 4 needs to find an $\varepsilon$-approximate Nash equilibrium of $f$ matches the lower bounds of $\Omega(\varepsilon^{-1})$ [52] for the number of iterations that any deterministic first-order method requires to find an $\varepsilon$-approximate Nash equilibrium of a smooth convex-concave function.

## D  Auxiliary Lemma

**Lemma D.1.** *For $\Gamma \geq 17$, the function $g(\Gamma)$ can be bounded by $\frac{80640}{\Gamma - 1} + \frac{2}{\Gamma e^{-2} - 1}$. Let $g(\Gamma)$ be defined as $\sum_{k=1}^{\infty} \Gamma^{-k}[k^7 + (k+1)\exp\{2k\}]$.*

*Proof.*

$$g(\Gamma) \leq \sum_{k=1}^{\infty} \left[ \Gamma^{-k} \frac{(k+7)!}{k!} + \left( \frac{e^2}{\Gamma} \right)^k (k+1) \right]$$

$$= \frac{d^7}{d\alpha^7} \left( \frac{\alpha^8}{1-\alpha} \right) \Big|_{\alpha=\Gamma^{-1}} + \frac{d}{d\alpha} \left( \frac{\alpha^2}{1-\alpha} \right) \Big|_{\alpha=e^2\Gamma^{-1}}$$

$$\underset{(a)}{\leq} \frac{80640}{\Gamma - 1} + \frac{2}{\Gamma e^{-2} - 1}, \tag{126}$$

(a) can be deduced based on the following inequality

$$\frac{d^7}{d\alpha^7} \left( \frac{\alpha^8}{1-\alpha} \right) = \sum_{k=0}^{7} (-1)^k \binom{7}{k} \frac{8!k!}{(k+1)!} \left( \frac{\alpha}{1-\alpha} \right)^{k+1}$$

$$\underset{(b)}{\leq} 7! \sum_{k=1}^{8} \binom{8}{k} \left( \frac{\alpha}{1-\alpha} \right)^k$$

$$= 7! \left[ \left( 1 + \frac{\alpha}{1-\alpha} \right)^8 - 1 \right]$$

$$\leq 7! \left[ \exp\left\{ \frac{8\alpha}{1-\alpha} \right\} - 1 \right]$$

$$\underset{(c)}{\leq} \frac{80640\alpha}{1-\alpha}, \tag{127}$$

where (b) and (c) are derived from Leibniz equation, and the inequality $e^x - 1 \leq 2x$ holds for $0 \leq x \leq 1/2$ respectively. $\qquad\square$

**Lemma D.2.** *For any $n \in \mathbb{N}, \boldsymbol{r} \in \mathbb{R}^n, \boldsymbol{p} \in \Delta^n$, if it holds that $\boldsymbol{p}^* = \arg\min_{\boldsymbol{p} \in \Delta_n} \eta \langle \boldsymbol{p}, \boldsymbol{r} \rangle + \mathrm{KL}(\boldsymbol{p} \| \boldsymbol{q})$, then we have*

$$\langle \boldsymbol{p}^* - \boldsymbol{p}, \boldsymbol{r} \rangle = \frac{1}{\eta} \left( \mathrm{KL}(\boldsymbol{p} \| \boldsymbol{q}) - \mathrm{KL}(\boldsymbol{p} \| \boldsymbol{p}^*) - \mathrm{KL}(\boldsymbol{p}^* \| \boldsymbol{q}) \right). \tag{128}$$

*Proof.* We just need to prove $\boldsymbol{p}^*(i) \equiv \boldsymbol{p}'(i) := \frac{\boldsymbol{q}(i) \exp\{-\eta \boldsymbol{r}(i)\}}{\sum_{j=1}^{n} \boldsymbol{q}(j) \exp\{-\eta \boldsymbol{r}(j)\}}$ for any $i \in [n]$ which satisfies

$$\langle \boldsymbol{p} - \boldsymbol{p}', \eta \boldsymbol{r} + \log(\boldsymbol{p}') - \log(\boldsymbol{q}) \rangle = 0, \tag{129}$$

for any $\boldsymbol{p} \in \Delta_n$. Assume that $F(\boldsymbol{p}) := \eta \langle \boldsymbol{p}, \boldsymbol{r} \rangle + \mathrm{KL}(\boldsymbol{p} \| \boldsymbol{q})$ and define $\mathcal{E}(\boldsymbol{p}) = \sum_{i=1}^{n} \boldsymbol{p}(i) \log(\boldsymbol{p}(i))$ for any $\boldsymbol{p} \in \Delta_n$. Clearly, $\boldsymbol{p}' \in \Delta_n$. Hence, for all $\boldsymbol{p} \in \Delta_n$,

$$F(\boldsymbol{p}) = \eta \langle \boldsymbol{p}, \boldsymbol{r} \rangle + \mathrm{KL}(\boldsymbol{p} \| \boldsymbol{q})$$

$$= \eta \langle \boldsymbol{p}', \boldsymbol{r} \rangle + \mathrm{KL}(\boldsymbol{p}' \| \boldsymbol{q}) + \langle \boldsymbol{p} - \boldsymbol{p}', \eta \boldsymbol{r} - \log(\boldsymbol{q}) \rangle + \mathcal{E}(\boldsymbol{p}) - \mathcal{E}(\boldsymbol{p}')$$

$$\underset{(a)}{=} \eta \langle \boldsymbol{p}', \boldsymbol{r} \rangle + \mathrm{KL}(\boldsymbol{p}' \| \boldsymbol{q}) + \mathcal{E}(\boldsymbol{p}) - \mathcal{E}(\boldsymbol{p}') + \langle \boldsymbol{p} - \boldsymbol{p}', -\log(\boldsymbol{p}') \rangle$$

$$\underset{(b)}{=} F(\boldsymbol{p}') - \mathrm{KL}(\boldsymbol{p} \| \boldsymbol{p}'), \tag{130}$$

where (a) is derived from Eq. (129). Therefore, we obtain that $\boldsymbol{p}^* \equiv \boldsymbol{p}'$. By using equality (b), we finish the proof. $\qquad\square$

**Lemma D.3.** *Suppose that for $\tau \in (0, 1)$, we have $\left\| \frac{\boldsymbol{p}}{\boldsymbol{q}} \right\|_{\infty} \leq 1 + \tau$. Then*

$$\left( \frac{1-\tau}{2} - \frac{2\tau}{3(1-\tau)} \right) \mathcal{X}^2(\boldsymbol{p}, \boldsymbol{q}) \leq \mathrm{KL}(\boldsymbol{p} \| \boldsymbol{q}).$$

*Proof.* We consider the Taylor expansion of the function $\log(1+x) = \sum_{k=1}^{\infty} \frac{(-1)^{k-1}}{k} x^k$ and define $Q_{\tau,D}(x) := x - \left(\frac{1}{2} + D\tau\right) x^2$. According to

$$\log(1+x) - Q_{\tau,D}(x) \geq D\tau x^2 - \frac{|x|^3}{3(1-\tau)}, \tag{131}$$

for any $x \in [-\tau, \tau]$, we have $\log(1+x) \geq Q_{\tau,D}(x)$ when $D \geq \frac{1}{3(1-\tau)}$ and $x \in [-\tau, \tau]$. Therefore, we obtain

$$
\begin{aligned}
\mathrm{KL}(\boldsymbol{p}\|\boldsymbol{q}) &= \sum_{j=1}^{n} \boldsymbol{p}(j) \log\left(\frac{\boldsymbol{p}(j)}{\boldsymbol{q}(j)}\right) \\
&\geq \sum_{j=1}^{n} \boldsymbol{p}(j) \left[\left(\frac{\boldsymbol{p}(j)}{\boldsymbol{q}(j)} - 1\right) - \left(\frac{1}{2} + D\tau\right)\left(\frac{\boldsymbol{p}(j)}{\boldsymbol{q}(j)} - 1\right)^2\right] \\
&= \mathcal{X}^2(\boldsymbol{p}, \boldsymbol{q}) - \left(\frac{1}{2} + D\tau\right) \sum_{j=1}^{n} \frac{\boldsymbol{p}(j)}{\boldsymbol{q}(j)} \boldsymbol{q}(j) \left(\frac{\boldsymbol{p}(j)}{\boldsymbol{q}(j)} - 1\right)^2 \\
&\geq \mathcal{X}^2(\boldsymbol{p}, \boldsymbol{q}) - \left(\frac{1+\tau}{2} + D\tau(1+\tau)\right) \mathcal{X}^2(\boldsymbol{p}, \boldsymbol{q}) \\
&= \left(\frac{1-\tau}{2} - D\tau(1+\tau)\right) \mathcal{X}^2(\boldsymbol{p}, \boldsymbol{q}).
\end{aligned}
\tag{132}
$$

We complete the proof if $D = \frac{1}{3(1-\tau)}$. $\square$

**Lemma D.4.** *Suppose that $\boldsymbol{r} \in \mathbb{R}^n, \tau \in (0, 1/2), \|\boldsymbol{r}\|_{\infty} \leq \frac{\tau}{2}$, and $\boldsymbol{p}, \tilde{\boldsymbol{p}} \in \Delta_n$ satisfy, for each $j \in [n]$,*

$$\tilde{\boldsymbol{p}}(j) = \frac{\boldsymbol{p}(j) \cdot \exp\{\boldsymbol{r}(j)\}}{\sum_{j' \in [n]} \boldsymbol{p}(j') \cdot \exp\{\boldsymbol{r}(j')\}}. \tag{133}$$

*Then*

$$\left(1 - \left(\frac{2}{3(1-\tau)} + 4\right)\tau\right) \mathrm{Var}_{\boldsymbol{p}}(\boldsymbol{r}) \leq \mathcal{X}^2(\tilde{\boldsymbol{p}}, \boldsymbol{p}) \leq \left(1 + \left(\frac{2}{3(1-\tau)} + 4\right)\tau\right) \mathrm{Var}_{\boldsymbol{p}}(\boldsymbol{r}). \tag{134}$$

*Proof.* Without loss of generality, we consider the case $\langle \boldsymbol{p}, \boldsymbol{r}\rangle = 0$. If not, redefine $\tilde{\boldsymbol{r}} := \boldsymbol{r} - \langle \boldsymbol{p}, \boldsymbol{r}\rangle \cdot \boldsymbol{e} (\|\tilde{\boldsymbol{r}}\|_{\infty} \leq \tau)$ and analyze $\tilde{\boldsymbol{r}}$ where $\boldsymbol{e} \in \mathbb{R}^n$ is an all 1 vector. It's clear that

$$\mathcal{X}^2(\tilde{\boldsymbol{p}}, \boldsymbol{p}) = -1 + \sum_{j=1}^{n} \boldsymbol{p}(j) \left(\frac{\tilde{\boldsymbol{p}}(j)}{\boldsymbol{p}(j)}\right)^2 = -1 + \mathbb{E}_{\boldsymbol{p}}\left[\frac{\exp\{\boldsymbol{r}\}}{\mathbb{E}_{\boldsymbol{p}}[\exp\{\boldsymbol{r}\}]}\right]^2. \tag{135}$$

We define $F_D^1(x) := 1 + x + \frac{1-D\tau}{2}x^2, F_D^2(x) := 1 + x + \frac{1+D\tau}{2}x^2$ and note that for any $x \in [-\tau, \tau]$

$$\exp\{x\} - F_D^1(x) \geq \frac{D\tau}{2}x^2 - \frac{x^3}{6}, \tag{136}$$

$$F_D^2(x) - \exp\{x\} \geq \frac{D\tau}{2}x^2 - \frac{|x|^3}{6(1-\tau)}, \tag{137}$$

where Eq. (136) is derived from the summation of the $2k$-th and $2k+1$-th ($k \geq 2$) terms in the Taylor expansion of $\exp\{x\}$ is always non-negative, Eq. (137) is derived from $\sum_{k=3}^{\infty} \frac{x^k}{k!} \leq \frac{|x|^3}{6(1-x)} \leq \frac{|x|^3}{6(1-\tau)}$ for any $x \in [-\tau, \tau]$. Therefore, we have $\exp\{x\} - F_D^1(x) \geq 0$ and $F_D^2(x) - \exp\{x\} \geq 0$ for all $x \in [-\tau, \tau]$ if $D \geq \frac{1}{3(1-\tau)}$. Then, we have

$$1 + 2x + (2 - (D+2)\tau)x^2 \leq (\exp\{x\})^2 \leq 1 + 2x + (2 + (D+2)\tau)x^2, \tag{138}$$

when $D\tau \leq \frac{1}{2}$. In addition, by $\langle \boldsymbol{p}, \boldsymbol{r}\rangle = 0$, it's obvious that

$$1 + \frac{1 - D\tau}{2} \mathbb{E}_{\boldsymbol{p}}[\boldsymbol{r}^2] \leq \mathbb{E}_{\boldsymbol{p}}[\exp\{\boldsymbol{r}\}] \leq 1 + \frac{1 + D\tau}{2} \mathbb{E}_{\boldsymbol{p}}[\boldsymbol{r}^2]. \tag{139}$$

Combining Eq. (138) and (139), we derived that

$$1 + (1 - (D+1)\tau)\mathbb{E}_{\boldsymbol{p}}[\boldsymbol{r}^2] \leq (\mathbb{E}_{\boldsymbol{p}}[\exp\{\boldsymbol{r}\}])^2 \leq 1 + (1 + (D+1)\tau)\mathbb{E}_{\boldsymbol{p}}[\boldsymbol{r}^2], \tag{140}$$

$$1 + (2 - (D+2)\tau)\mathbb{E}_{\boldsymbol{p}}[\boldsymbol{r}^2] \leq \mathbb{E}_{\boldsymbol{p}}\left[(\exp\{\boldsymbol{r}\})^2\right] \leq 1 + (2 + (D+2)\tau)\mathbb{E}_{\boldsymbol{p}}[\boldsymbol{r}^2], \tag{141}$$

for $D\tau \leq \frac{1}{2}$. According to Eq. (135) ,(140) and (141), we have

$$-1 + \mathbb{E}_{\boldsymbol{p}}\left[\frac{\exp\{\boldsymbol{r}\}}{\mathbb{E}_{\boldsymbol{p}}[\exp\{\boldsymbol{r}\}]}\right]^2 \geq \frac{(1 - (2D+3)\tau)\mathbb{E}_{\boldsymbol{p}}[\boldsymbol{r}^2]}{1 + (1 + (D+1)\tau)\mathbb{E}_{\boldsymbol{p}}[\boldsymbol{r}^2]} \geq (1 - (2D+4)\tau)\mathbb{E}_{\boldsymbol{p}}[\boldsymbol{r}^2],$$

$$-1 + \mathbb{E}_{\boldsymbol{p}}\left[\frac{\exp\{\boldsymbol{r}\}}{\mathbb{E}_{\boldsymbol{p}}[\exp\{\boldsymbol{r}\}]}\right]^2 \leq \frac{(1 + (2D+3)\tau)\mathbb{E}_{\boldsymbol{p}}[\boldsymbol{r}^2]}{1 + (1 - (D+1)\tau)\mathbb{E}_{\boldsymbol{p}}[\boldsymbol{r}^2]} \leq (1 - (2D+4)\tau)\mathbb{E}_{\boldsymbol{p}}[\boldsymbol{r}^2].$$

We derive Eq. (134) by setting $D = \frac{1}{3(1-\tau)}$. $\qquad\square$

**Lemma D.5** (Lemma B.6, [15]). *Let $\phi_1, \cdots, \phi_l$ be softmax-type functions.*

$$\phi_i(\boldsymbol{x}) = \frac{\exp\{\boldsymbol{x}(j_i)\}}{\sum_{k=1}^n \tau_{ik}\exp\{\boldsymbol{x}(k)\}}, \tag{142}$$

*where $j_i \in [1, \cdots, n], \sum_{k=1}^n \tau_{ik} = 1$ for any $i \in [1, \cdots, l]$. Let $P(\boldsymbol{x}) = \sum_{k=0}\sum_{|\boldsymbol{\alpha}|=k}\frac{D^{\boldsymbol{\alpha}}P(\boldsymbol{0})}{\boldsymbol{\alpha}!}\boldsymbol{x}^{\boldsymbol{\alpha}}$ denote the Taylor series of $\prod_{i=1}^l \phi_i$. Then for any integer $k$,*

$$\sum_{|\boldsymbol{\alpha}|=k}\frac{|D^{\boldsymbol{\alpha}}P(\boldsymbol{0})|}{\boldsymbol{\alpha}!} \leq (e^3 l)^k. \tag{143}$$

We introduce the conception of $(Q, R)$-bounded function briefly. Suppose $\phi : \mathbb{R}^n \to \mathbb{R}$ is real-analytic in a neighborhood of the origin. For real numbers $Q, R > 0$, we say that $\phi$ is $(Q, R)$-bounded if the Taylor expansion of $\phi$ at $\boldsymbol{0}$, denoted $P_\phi(\boldsymbol{x}) = \sum_{k=0}^\infty \sum_{|\boldsymbol{\alpha}|=k}\frac{D^{\boldsymbol{\alpha}}f(\boldsymbol{0})}{\boldsymbol{\alpha}!}\boldsymbol{x}^{\boldsymbol{\alpha}}$, satisfies, for each integer $i \geq 0$, $\sum_{|\boldsymbol{\alpha}|=k}\frac{|D^{\boldsymbol{\alpha}}\phi(\boldsymbol{0})|}{\boldsymbol{\alpha}!} \leq Q \cdot R^k$.

**Lemma D.6** (Detailed version of Lemma 4.5, [15]). *Suppose that $h, n \in \mathbb{N}, \phi : \mathbb{R}^n \to \mathbb{R}$ is a $(Q, R)$-bounded function such that the radius of convergence of its power series at $\boldsymbol{0}$ is at least $\nu > 0$, and $\boldsymbol{Z} = \{\boldsymbol{Z}^0, \cdots, \boldsymbol{Z}^T\} \subset \mathbb{R}^n$ is a sequence of vectors satisfying $\left\|\boldsymbol{Z}^t\right\|_\infty \leq \nu$ for $t \in [0, \cdots, T]$. Suppose for some $\beta \in (0, 1)$, for each $0 \leq h' \leq h$ and $t \in [0, \cdots, T - h']$, it holds that $\left\|D_{h'}\boldsymbol{Z}^t\right\|_\infty \leq \frac{1}{\Gamma R}\beta^{h'}(h')^{Bh'}$ for some $B \geq 3, \Gamma \geq e^3$. Then for all $t \in [0, \cdots, T - h]$,*

$$\left|(D_h(\phi \circ \boldsymbol{Z}))^t\right| \leq Q \cdot g(\Gamma) \cdot \beta^h h^{Bh+1}, \tag{144}$$

*where $g(\Gamma)$ is a bounded function with respect to $\Gamma$.*

*Proof.* Without loss of generality, we assume $\phi(\boldsymbol{0}) = 0$. We define $(\phi \circ Z)^t = \sum_{\boldsymbol{\gamma}\in\mathbb{Z}_{\geq 0}^n:|\boldsymbol{\gamma}|=k} a_{\boldsymbol{\gamma}}\left(\boldsymbol{Z}^t\right)^{\boldsymbol{\gamma}}$ and obtain

$$\left|(D_h(\phi \circ \boldsymbol{Z}))^t\right| = \left|\sum_{k=1}^\infty \sum_{\boldsymbol{\gamma}\in\mathbb{Z}_{\geq 0}^n:|\boldsymbol{\gamma}|=k} a_{\boldsymbol{\gamma}}(D_h\boldsymbol{Z}^{\boldsymbol{\gamma}})^t\right|$$

$$\leq \sum_{k=1}^\infty \sum_{\boldsymbol{\gamma}\in\mathbb{Z}_{\geq 0}^n:|\boldsymbol{\gamma}|=k} |a_{\boldsymbol{\gamma}}|\left(\sum_{x:[h]\to[k]}\prod_{j=1}^k\left|\left(E_{t'_{x,j}}D_{h'_{x,j}}\boldsymbol{Z}(l'_{x,j})\right)^t\right|\right)$$

$$\leq \sum_{k=1}^\infty \sum_{\boldsymbol{\gamma}\in\mathbb{Z}_{\geq 0}^n:|\boldsymbol{\gamma}|=k} |a_{\boldsymbol{\gamma}}| \cdot \frac{\beta^h}{(\Gamma R)^k}\cdot\left(\sum_{x:[h]\to[k]}\prod_{j=1}^k (h'_{x,j})^{Bh'_{x,j}}\right)$$

$$\leq \sum_{k=1}^\infty \sum_{\boldsymbol{\gamma}\in\mathbb{Z}_{\geq 0}^n:|\boldsymbol{\gamma}|=k} |a_{\boldsymbol{\gamma}}| \cdot \frac{\beta^h}{(\Gamma R)^k}h^{Bh}\max\left\{k^7, (hk+1)\exp\left\{\frac{2k}{h^{B-1}}\right\}\right\}$$

$$\underset{(c)}{\leq} \sum_{k=1}^\infty Q\left(\frac{R}{\Gamma R}\right)^k\cdot\max\left\{k^7, (k+1)\exp\{2k\}\right\}\cdot\beta^h h^{Bh+1}$$

$$\leq Q \cdot g(\Gamma) \cdot \beta^h h^{Bh+1}, \tag{145}$$

where (c) is derived from $(Q, R)$-bounded condition. $\qquad\qquad\qquad\qquad\qquad\qquad\square$

**Lemma D.7** (Lemma C.4, [15]). *Let $\{n, T\} \subset \mathbb{Z}_+$ with $n \geq 2$ and $T \geq 4$, we select $H :=$ $\lceil \log(T) \rceil, \beta_0 = \frac{1}{4H}$, and $\beta = \frac{\sqrt{\beta_0/8}}{H^3}$. Assume that $\{z^t\}_{t=1}^{T} \subset [0, 1]^n$ and $\{p^t\}_{t=1}^{T} \subset \Delta_n$ satisfy the following condition*

1. *For each $0 \leq h \leq H$ and $1 \leq t \leq T - h$, it holds that $\|(D_h z)^t\|_\infty \leq \beta^h H^{3h+1}$.*

2. *The sequence $\{p^t\}_{t=1}^{T}$ is $\zeta$−consecutively close for some $\zeta \in \left[ (2T)^{-1}, \beta_0^4/8256 \right]$.*

*Then, we have*

$$\sum_{t=1}^{T} \mathrm{Var}_{p^t}(z^t - z^{t-1}) \leq 2\beta_0 \sum_{t=1}^{T} \mathrm{Var}_{p^t}(z^{t-1}) + 165120(1 + \zeta)H^5 + 2. \tag{146}$$

**Proposition D.8.** *Given a constant $c > 0$, we have*

$$\sum_{k=1}^{t} \left( \frac{c}{c+k} \right)^2 \leq c. \tag{147}$$

**Lemma D.9.** *For a constant $c \geq c' > 0$, the following inequality holds*

$$c' \log \left( \frac{c+t-1}{c+T} \right) - \frac{(c' + c'c)^2}{2c} \leq \sum_{k=t}^{T} \log \left( 1 - \frac{c'}{c+k} \right) \leq c' \log \left( \frac{c+t}{c+1+T} \right), \tag{148}$$

*when $T > t \geq 1$.*

*Proof.* Accoding to the Taylor expansion of $\log(1 - x)$ when $x < 1$, we obtain the estimation of $\log \left( 1 - \frac{c'}{c+k} \right)$ for any $k \geq 1$ as follows

$$\log \left( 1 - \frac{c'}{c+k} \right) \leq - \frac{c'}{c+k}, \tag{149}$$

$$\log \left( 1 - \frac{c'}{c+k} \right) \geq - \frac{c'}{c+k} - \frac{(c' + cc')^2}{2} \left( \frac{1}{c+k} \right)^2. \tag{150}$$

Next, we have

$$\sum_{k=t}^{T} - \frac{c'}{c+k} \leq - \int_{t}^{T+1} \frac{c'}{c+x} dx = c' \log \left( \frac{c+t}{c+1+T} \right), \tag{151}$$

$$\sum_{k=t}^{T} \left[ - \frac{c'}{c+k} - \frac{(c' + cc')^2}{2} \left( \frac{1}{c+k} \right)^2 \right] \geq - \int_{t-1}^{T} \left[ \frac{c'}{c+x} + \frac{(c' + cc')^2}{2} \left( \frac{1}{c+x} \right)^2 \right] dx$$

$$\geq c' \log \left( \frac{c+t-1}{c+T} \right) - \frac{(c' + cc')^2}{2c}. \tag{152}$$

$\qquad\qquad\qquad\qquad\qquad\qquad\qquad\qquad\qquad\qquad\qquad\qquad\qquad\qquad\qquad\square$

# E Limitation

For objectives with GQC condition (GQCC condition) and general smooth internal function (i.e. Lipschitz continuous internal function), our analytical method might not provide similar iteration complexity. We leave the related algorithmic analysis on more generalized smoothness conditions as a future work.

