# OpenReview forum: "Optimizing over Multiple Distributions under Generalized Quasar-Convexity Condition"
_NeurIPS.cc/2024/Conference — NeurIPS 2024 poster_

### Official Review · Reviewer_oxwM · 2024-06-21

**Soundness:** 3
**Presentation:** 3
**Contribution:** 3
**Rating:** 7
**Confidence:** 2

**Summary:**

The paper considers a minimization problem, where the optimizing variable is composed of $d$ probability distributions. The notion of quasar convexity is generalized by allowing for different quasar-convexity parameters $\gamma_i, i=1,...,d$ for the $d$ distributions. Instead of using the gradient function for the oracle term, more generic functions $F_i, i=1,...,d$ ("internal functions") are allowed.
Convergence guarantees are established for the situations where the interval functions are either Lipschitz continuous of "polynomial-like".
Infinite horizon reinforcement learning is shown two be a special case of this setting. Similar results are derived for the minimax setting.

**Strengths:**

* thorough theoretical analysis
* method requires strictly less iterations than mirror descent
* algorithm also works for unknown $\gamma_i$
* reinforcement learning application is an important one
* extensive overview of related work

**Weaknesses:**

* Algorithm 1 and Algorithm 2 are never actually described in the text, they are only provided as pseudocode. It think it would be nice to add 2-3 sentences to point out how exactly it differs from basic mirror descent. Example given, I don't understand why one randomly picks t following probability $\mathbb{P}[t] =1/T$.
* There is "only" one example (the reinforcement application) for the new structure the paper introduces. I am not sure if one example is enough to justify speaking of a new structure. (But I also understand that such examples are complex and it might go beyond the scope of the paper to come up with several such examples)
* the paper does not mention the computational costs of the algorithm. Is it $d$ times as much as basic mirror descent? Is this still feasible in praxis?



Typos:
* Headline Algorithm 1: Mirior -> Mirror
* line 206: depends -> depend

**Questions:**

* I am confused about the following part in the introduction (line 62-65): "We then study designing efficient algorithms to solve (1). One simple case is when $\gamma_i, i=1...m$ is pre-known by the algorithms. The possible direction is to impose a $\gamma_i$-dependent update rule, such as by non-uniform sampling. However, in general cases,
$\gamma_i, i=1...m$ is not known and determining $\gamma_i, i=1...m$ require non-negligible costs." Can you elaborate on this?

**Limitations:**

I don't see that the authors explicitly mention limitations of their work, but it is also a theoretical paper, so everyone can just read the assumptions at the beginning of the theorems to see where the results apply and where not. The assumptions made for the theoretical analysis are stated clearly.

---

> ### Author Rebuttal · Authors · 2024-08-05
>
> Thank you for your comments and suggestions. We will respond to each of your comments individually in the following.
>
> ## Algorithms Descriptions:
> Thanks for your suggestion. We will use equations and sentences to introduce the algorithm. One randomly picks $t$ following probability means we output $x^t$ by uniformly sample from $(x^t)_{t=1}^T$.
>
> ## Examples:
> For minimization problems, we provide two examples of functions that simultaneously satisfy both the GQC condition and Assumption 3.3: a toy example (Example B.8 in Appendix B) and a simple neuron network over a simplex (Example 3.4 in Section 3.2). For minimax problems, In addition to infinite horizon two-player zero-sum Markov games, we have also demonstrated that general smooth convex-concave problems satisfy the GQCC condition and Assumption 4.2 (see Appendix C.3.2). Further examples await our future exploration.
>
> ## Computational Costs:
> Since the MD method requires computing a $\sum_{i=1}^d n_i$-dimensional gradient vector at each iteration, and Algorithm 1 also requires computing a $\sum_{i=1}^d n_i$-dimensional internal function at each iteration, so Algorithm 1 does not need more additional computational costs.
>
> ## Response to confuse about line 62-65:
> Sorry for the confusion. We mean that If $\{\gamma_i\}$ is known, then the update of our algorithm on each variable block will depend on $\gamma_iF_i$ rather than solely on $F_i$. However, in many applications, $\{\gamma_i\}$ is often related to the information of the optimal point (see Appendix B.3 for reinforcement learning problem, $\gamma_i $ depends on the optimal solutions). Therefore, the computational costs of estimating $\{\gamma_i\}$ may even exceed that to solve the optimization problems. This is our motivation to design adaptive algorithms without pre-known $\{\gamma_i\}$.

---

> > ### Comment · Reviewer_oxwM · 2024-08-12
> >
> > Thank you for the clarifications. I keep my positive score.

---

### Official Review · Reviewer_e1HQ · 2024-07-11

**Soundness:** 2
**Presentation:** 2
**Contribution:** 2
**Rating:** 4
**Confidence:** 4

**Summary:**

The authors study a typical optimization model where the optimization variable is composed of multiple probability distributions. For this optimization problem, they propose a new structural condition/landscape description named generalized quasar-convexity (GQC) beyond the realms of convexity.

**Strengths:**

It looks like the authors propose a new theoretical analysis for the quasar-convex optimization.

**Weaknesses:**

It would be better if the authors added more motivation for why they focus on quasar optimization.

Another point: the theoretical analysis demonstrated is not very clear. If they can show it clearer, it will be better,

**Questions:**

Does the assumption 4.2 can be relaxed?

Can the Section 4.3 be written more?

---

> ### Author Rebuttal · Authors · 2024-08-05
>
> Thank you for your comments and suggestions. We will respond to each of your comments individually in the following.
>
> ## Motivation:
> In many non-convex optimization problems, the objective function often admits  ''convexity-like'' properties, such as matrix completion, phase retrieval, and neural network under some settings (Bartlett et al., 2019; Ge et al., 2016). Therefore, exploring the structured sets of functions for which the problem can be efficiently solved is meaningful. This aligns with the ideas proposed by Hinder et al., (2020). Quasar convexity is a widely known condition in the field of optimization with two typical examples in machine learning: 1) the loss functions of neural networks satisfy the quasar convexity condition in large neighborhoods of the minimizers (Kleinberg et al., 2018; Zhou et al., 2019); 2) Under mild assumptions, the objective function for learning linear dynamical systems also admits to the quasar convexity condition (Hardt et al., 2018). Furthermore, generalized quasar convexity (GQC), as an extension of quasar convexity, encompasses not only quasar convexity but also the function structures of reinforcement learning and certain neural networks. This implies that the objective functions of more machine learning problems may satisfy the GQC condition. Our analysis provides a theoretical guarantee for the solvability of approximate global optima of these problems. As a supplement to the GQC condition, generalized quasar-convexity-concavity (GQCC) extends the concept of convexity-concavity. Based on this landscape characterization and appropriate assumptions, we propose an algorithm for solving minimax problems that have a better convergence rate in specific machine learning problems.
>
> ## Theoretical analysis is not clear:
> Thanks for your suggestions. We will add more intuitive discussions for our analysis. For
> Thm 3.5, the key idea is to reduce the impact of each variable block on the function error, which plays a crucial role in establishing global convergence. Concretely, we separate the upper bound of $\frac{1}{T}\sum_{t=1}^T(f(x^t)-f(x^{*}))$ into: a) an invariant lower order term and b) the weighted sum of variance of finite difference term $\frac{\mathcal{O}(1)}{T} \sum_{t=1}^TVar_{x_i^t}(F_i(x^t)-F_i(x^{t-1}))-\frac{\mathcal{O}(1)}{T} \sum_{t=1}^TVar_{x_i^t}(F_i(x^{t-1}))$. Furthermore, applying the property of sequence $(F_i(x^t))_{t=1}^T$ under ''polynomial-like'' structure condition, we bound (b) by a quantity that grows poly-logarithmically in $T$. For a more detailed proof sketch, please refer to the beginning of Appendix B.1.
> We will add more explanations for both Thm 3.5 and 4.4 in the revised version.
>
> ##  Relax Assumption 4.2:
> Thanks for suggestion. Currently, to achieve the faster $\tilde{\mathcal{O}}(1/T)$ convergence rate, we do not find a clear way to have a good relaxation. The main challenges of multi-variable block optimization problems lie in its non-convex (non-convex-non-concave) structure and the coupling of multiple variable blocks. Therefore, a simple Lipschitz continuity assumption on internal function may not be helpful in obtaining a faster rate. However, for a slower rate, such as $\tilde{\mathcal{O}}(1/\sqrt{T})$, it is possible and we leave it as a future work. We will discuss it in the revised version.
>
> ## Written more Section 4.3:
> Thanks for suggestion. We will introduce more background about Markov games, including its problem formulation, relation to convex and concave optimization, and current results.

---

> ### Author Response · Authors · 2024-08-11
>
> Dear reviewer,
>
> Thank you for your efforts in the review process. We have tried our best to address your concerns and answer the insightful questions. Please let us know if you have any other questions regarding our paper or response. If we have successfully addressed your questions, we would greatly appreciate if you can reevaluate the score.
>
> Best,
>
> Authors

---

### Official Review · Reviewer_NA4U · 2024-07-13

**Soundness:** 3
**Presentation:** 3
**Contribution:** 3
**Rating:** 6
**Confidence:** 2

**Summary:**

This paper introduces a novel optimization model for addressing problems involving multiple probability distributions, a common scenario in fields such as policy optimization and reinforcement learning. The authors present a new structural condition called Generalized Quasar-Convexity (GQC), which extends the original quasar-convexity concept by allowing individual quasar-convex parameters for each variable block. This flexibility accommodates varying degrees of block convexity. The paper proposes an Optimistic Mirror Descent (OMD) algorithm tailored to this framework and demonstrates its efficiency in achieving an ε-suboptimal global solution. Furthermore, the paper extends the GQC concept to minimax optimization problems by introducing the Generalized Quasar-Convexity-Concavity (GQCC) condition. The theoretical findings are supported by applications in discounted Markov Decision Processes (MDPs) and Markov games, showing improved iteration complexity bounds over existing methods.

**Strengths:**

- The introduction of GQC and GQCC conditions provides a fresh perspective on optimization problems involving multiple distributions, extending beyond traditional convexity assumptions.

- The paper offers rigorous theoretical analysis, including complexity bounds for the proposed OMD algorithm, and demonstrates that these bounds are tighter than those for existing methods.

- The application of the proposed framework to reinforcement learning and Markov games showcases its practical relevance and potential impact on real-world problems.

**Weaknesses:**

- The assumptions made for the GQC and GQCC conditions, such as polynomial-like structures and specific Lipschitz continuity requirements, may limit the generality of the results. It would be beneficial to discuss the applicability of these assumptions in broader contexts.

- While the theoretical contributions are robust, the paper lacks empirical validation through extensive experiments. Including experimental results on synthetic and real-world datasets would strengthen the paper by demonstrating the practical performance of the proposed algorithms.

**Questions:**

Yes

**Limitations:**

- How do the assumptions made for the GQC and GQCC conditions compare to those in other optimization frameworks? Are there potential relaxations or alternative conditions that could be explored to broaden the applicability of the proposed methods?

- Can you provide more detailed insights or examples of real-world problems where the proposed GQC and GQCC conditions are particularly beneficial? How do these conditions improve the solution quality or computational efficiency in such scenarios?

---

> ### Author Rebuttal · Authors · 2024-08-05
>
> Thanks for your comments and suggestions. We will respond to each of your comments individually in the following.
> ## Discuss the applicability of these assumptions in broader contexts:
> Thanks for your suggestion. We will add more discussions on the applicability of the assumption and our framework. For minimization problems under the GQC condition, we begin with a convergence result relying on the general Lipschitz continuous condition of internal function $F$. However, under this condition, we need to additionally assume that $\max_{i\in[1:d]}\gamma_i<+\infty$. Based on the aforementioned assumption merely, according to our knowledge, it appears impossible to eliminate the coupling effects between different variable blocks without a consistent upper-bounded assumption of $\{\gamma_i\}$. In certain applications, $\max_{i\in[1:d]}\gamma_i$ can indeed approach $+\infty$. For instance, in the infinite horizon reinforcement learning problems, the discounted state visitation distribution $d_{\rho_0}^{\pi}(s)$ for a particular state $s$ may approach zero, implying that $1/\gamma_s=0$. Therefore, to address this uncertainty of unknown $\{\gamma_i\}$, we introduce the Assumption 3.3.  Assumption 3.3 is not hard to achieve. In Proposition B.2 of Appendix B, we demonstrate that if the absolute value of the higher-order derivatives of a vector-valued function $F$ grows exponentially with its order, then $F$ admits Assumption 3.3. Furthermore, since the feasible domain of the functions considered in this paper is a bounded closed region, the conditions of Proposition B.2 are easily satisfied by smooth $F$, making Assumption 3.3 hold easily. Additionally, we provide two examples of functions that simultaneously satisfy both the GQC condition and Assumption 3.3: a toy example (Example B.8 in Appendix B) and a simple neuron network over a simplex (Example 3.4 in Section 3.2). For minimax problems, most algorithms that guarantee global convergence are based on a convex-concave structure. Our assumptions not only encompass the assumption of Lipschitz continuous gradients for the objective function $f$ in the convex-concave case (refer to Section C.3.2 in Appendix C) but also provide a unified landscape description for infinite horizon two-player zero-sum Markov games. It is worth noting that Algorithm 2 in this paper achieves a better convergence rate in infinite horizon two-player zero-sum Markov games compared to the results in Wei et al., (2021).
>
> ## Experiments:
> Please refer to response Author Rebuttal:$\textbf{Experiments}$.
>
> ## Potential relaxations or alternative conditions:
> Thanks for your concerns about the generality of our conditions and the request to relax the assumptions. The GQC condition include quasar convexity (QC) condition. Actually, our algorithm can achieve $\tilde{\mathcal{O}}(1/T)$ convergence rate under QC condition with Lipschitz continuous internal function $\nabla f$. Currently, to achieve the faster $\tilde{\mathcal{O}}(1/T)$ convergence rate, we do not find a clear way to have a good relaxation. The main challenges of multi-variable block optimization problems lie in its non-convex (non-convex-non-concave) structure and the coupling of multiple variable blocks. Therefore, a simple Lipschitz continuity assumption on internal function may not be helpful in obtaining a faster rate. However, for a slower rate, such as $\tilde{\mathcal{O}}(1/\sqrt{T})$, it is possible and we leave it as a future work.
>
>
> ## Detailed insights or examples of real-world problems where the proposed GQC and GQCC conditions are particularly beneficial?
> We prove that for reinforcement learning (RL) problem, GQC can be achieved. For the infinite horizon reinforcement learning problems, the discounted state visitation distribution $d_{\rho_0}^{\pi}(s)$ satisfies $\sum_{s\in S}d_{\rho_0}^{\pi}(s)=1$, implying that $\sum_{s\in S}1/\gamma_s=1$, therefore our algorithmic complexity does not rely on $|S|$. However, if we describe the properties of the function using the worst $1/\gamma_s$ (similar to the QC condition), then the complexity might be $|S|\max_{s\in S}1/\gamma_s$. Similarly, we prove that infinite horizon two-player zero-sum Markov games satisfies GQCC condition. For infinite horizon two-player zero-sum Markov games, $\sum_{s\in S}\psi_s(z)\leq 2$ for any $z\in \mathcal{Z}$. Therefore, the convergence rate of Algorithm 2 does not depend on $|S|$.

---

### Official Review · Reviewer_hhK8 · 2024-07-14

**Soundness:** 3
**Presentation:** 3
**Contribution:** 3
**Rating:** 7
**Confidence:** 2

**Summary:**

The paper studies the optimization of generalized quasar convex (GQC) functions, a new global
structure introduced in this paper. This definition relaxes the quasar convexity (QC)
condition, in a way that different components of the optimization variable satisfy
the QC like condition with different parameters.
The key idea is that if we simply treat a GQC function as a QC function, one needs to take
d times the maximum value of these paramters which can be large. Applying GQC allows one
to deal with the sum instead which could be significantly smaller than d. Leveraging this
insight, the authors adapt the well-known optimistic mirror descent algorithm.
The main use case appears to be in policy optimization in MDPs, where the authors show
that the GQC condition is shown to be satisfied by the expected value function.
The authors also define a similar condition for minimax optimization problems, and apply
their framework to find the Nash equilibrium of a two-player zero sum Markov game.

Overall my evaluation of the paper is quite positive. Even as a non-expert, I was able to
appreciate the ideas and the contributions of the authors. The paper is also well-written
and easy to follow.

I have given a positive score to reflect my current assessment. However, as this is not
my area, I am unable to evaluate the technical merit and novelty of proot techniques in
the paper, and I did not dive into the proofs in the appendix. So I will also wait to hear
from more expert reviewers.

**Strengths:**

See above

**Weaknesses:**

See above

**Questions:**

See above

**Limitations:**

See above

---

> ### Author Rebuttal · Authors · 2024-08-05
>
> We thank you for dedicating your time and effort to reviewing our manuscript. We appreciate that you acknowledge our paper. We will also carefully revise the paper to enhance its readability.

---

### Author Rebuttal · Authors · 2024-08-05

## Experiments:
We add a simple simulation to validate our proposed algorithm. We do experiments on learning one single neuron network over
a simplex, which has been discussed in Example 3.4 in Section 3.2. The objective function is written as

\begin{align}
f(p,P)=\frac{1}{2}E_{x,y}(\sum_{i=1}^m p_i\sigma(x^\top  P_{i})-y)^2
\end{align}

where $\sigma(\cdot)=\exp(\cdot), p\in \Delta_m$ and $P = (P_1,\cdots,P_m)\in\prod_{i=1}^m\Delta_d $ and the target $y$ given $x\in[-C,C]^{d}$ admits $y = \sigma(x^\top P_1^*)$ for some $P_1^*\in \Delta_d$. We set $m=20$, $d=20$, $C=4$ and the optimal solution $\mathbf{P}_1^*\in\Delta_d$ is a random probability density function. We compare the convergence performance of OMD\_IF (Optimistic Mirror Descent Method with internal functions, Algorithm 1), OMD\_G (Optimistic Mirror Descent Method with gradients, (Rakhlin et al., 2012)), MD (Mirror Descent Method with gradients), and AMD (Accelerated Mirror Descent Method with gradients, (Lan, 2020)). The experimental result is shown in the PDF file. From the figure, we can see that Algorithm 1 consistently shows superior convergence performance under various step size choices.

---

### Decision · Program_Chairs · 2024-09-25

**Decision:**

Accept (poster)

**Comment:**

This paper introduces "generalized quasar convexity" and shows that the optimization problems of (P1) Tabular Infinite Horizon reinforcement learning problems and (P2) Two-Player Markov-Zero Sum Game satisfy the condition. Iteration complexity of Optimistic Mirror Descent for these two applications using the generalized quasar convexity is provided.

An issue raised during the discussion is that the iteration complexity is not better than the existing results in the literature, and the authors, in the current version, were not explicit about the comparison.  Specifically, for (P1), the current version does not explicitly state the best iteration complexity in the related works; for (P2), while it mentions that the result matches the state-of-the-art iteration bound, no citations are provided. Therefore, the usefulness of the proposed "generalized quasar convexity" might not be completely clear, and the authors might want to discuss more in the next version.

While the paper is recommended for acceptance, a more comprehensive discussion of the literature on (P1), (P2), and quasar convexity should be provided in the next version. It would also be illuminating to discuss how existing results on P1 and P2 were derived, such as using the PL condition (Gradient-Dominance Condition), and if the existing conditions used in the related works could be related to the proposed generalized quasar convex condition for these two problems.